# Procedural Knowledge in Pretraining Drives Reasoning in Large Language Models

**Laura Ruis***
AI Centre, UCL

**Maximilian Mozes**
Cohere

**Juhan Bae**
University of Toronto & Vector Institute

**Siddhartha Rao Kamalakara**
Cohere

**Dwarak Talupuru**
Cohere

**Acyr Locatelli**
Cohere

**Robert Kirk**
AI Centre, UCL

**Tim Rocktäschel**
AI Centre, UCL

**Edward Grefenstette**
AI Centre, UCL

**Max Bartolo**
Cohere

## Abstract

The capabilities and limitations of Large Language Models (LLMs) have been sketched out in great detail in recent years, providing an intriguing yet conflicting picture. On the one hand, LLMs demonstrate a general ability to solve problems. On the other hand, they show surprising reasoning gaps when compared to humans, casting doubt on the robustness of their generalisation strategies. The sheer volume of data used in the design of LLMs has precluded us from applying the method traditionally used to measure generalisation: train-test set separation. To overcome this, we study what kind of generalisation strategies LLMs employ when performing reasoning tasks by investigating the pretraining data they rely on. For two models of different sizes (7B and 35B) and 2.5B of their pretraining tokens, we identify what documents influence the model outputs for three simple mathematical reasoning tasks and contrast this to the data that are influential for answering factual questions. We find that, while the models rely on mostly distinct sets of data for each factual question, a document often has a similar influence across different reasoning questions within the same task, indicating the presence of procedural knowledge. We further find that the answers to factual questions often show up in the most influential data. However, for reasoning questions the answers usually do not show up as highly influential, nor do the answers to the intermediate reasoning steps. When we characterise the top ranked documents for the reasoning questions qualitatively, we confirm that the influential documents often contain procedural knowledge, like demonstrating how to obtain a solution using formulae or code. Our findings indicate that the approach to reasoning the models use is unlike retrieval, and more like a generalisable strategy that synthesises procedural knowledge from documents doing a similar form of reasoning.

## 1 Introduction

Current advancements in artificial intelligence are characterised by the increasing scale of datasets, computational power, and model size (Kaplan et al., 2020; Hoffmann et al., 2022). While one of the manifestations of this approach, Large Language Models (LLMs), is rapidly saturating benchmarks measuring reasoning capabilities (Cobbe et al., 2021; Hendrycks et al., 2021, inter alia), the debate over whether they exhibit 'genuine understanding' is ongoing (as reviewed by Mitchell & Krakauer, 2023). The well-documented robust and versatile reasoning abilities (Webb et al., 2023; 2024; McLeish et al., 2024, inter alia) sharply contrast with the line of work highlighting the brittleness of LLM reasoning (Razeghi et al., 2022; McCoy et al., 2023; Ullman, 2023; Wu et al., 2024; Mahowald et al., 2024). A finding common to these works is that LLM reasoning depends on the frequency of similar problems in the training data.

---

*Work done while at Cohere, correspondence to laura.ruis.21@ucl.ac.uk

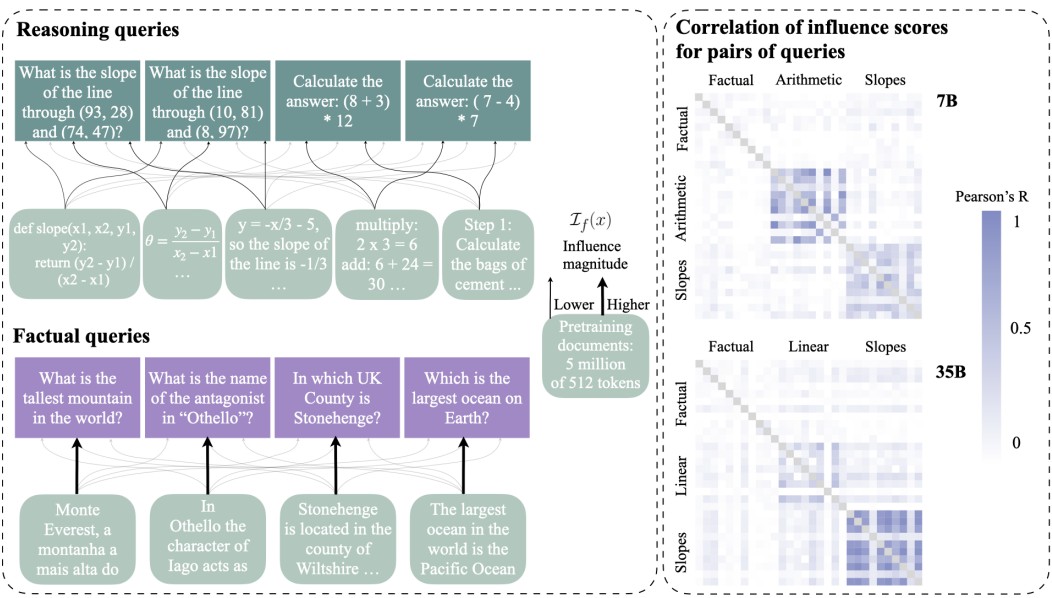

Figure 1: A summary of our most important findings towards answering the question *"how do LLMs learn to reason from pretraining data?"* We rank 5 million pretraining documents according to their influence on the likelihood of completions of two models, Cohere's Command R 7B and 35B, for 40 factual and 40 reasoning queries. We find that procedural knowledge drives influence on reasoning traces: a document's influence on the reasoning traces of one query is strongly predictive of that document's influence on another query with the same mathematical task, in 3 of the 4 cases. We show this on the left through arrows indicating influence, and on the right through correlations of all 5M document influences between a random sample of 10 queries per task (a plot with all queries can be found in Figure 12 in Appendix A.9.1). Further, we find that the answers to factual queries often show up in the top 0.01% of pretraining documents (see text in bottom row of documents), but not for the reasoning questions. Finally, individual documents influence reasoning traces much less strongly than factual answer generations, indicating models rely on documents less when reasoning. All documents and queries shown are redacted versions of real data, and the relations are based on documents found in the top 50 for the queries.

A key reason why benchmark saturation cannot be taken at face value is the issue of data contamination: benchmark data often appear in the pretraining set. Where we typically measure generalisation in machine learning by separating the test data from the training data, the trillions of tokens used in the design of current state-of-the-art models cannot reasonably be separated from benchmarks anymore. Recent works have documented the extent of the contamination issue (Brown et al., 2020; Touvron et al., 2023; Gunasekar et al., 2023; Yang et al., 2023; Deng et al., 2024), showing that many common benchmarks have a high percentage of contaminated data. Additionally, Yang et al. (2023) show that even rephrased benchmark data that elude N-gram-based detection methods can impact performance, further complicating the issue. However, it is unclear how and when state-of-the-art LLMs rely on contaminated data to perform reasoning.

This raises the question: *"how do LLMs learn to reason from pretraining data?"* In this work, we take a complementary approach to most interpretability research by focusing on the pretraining data used by language models to generalise, rather than interpreting the model weights themselves. We investigate which data influence the model's produced reasoning traces and how those data relate to the specific problems being addressed. Are models simply 'retrieving' answers from previously seen pretraining data and reassembling them, or are they employing a more robust strategy for generalisation? We use a technique from robust statistics (Hampel, 1974) adapted to large-scale Transformers (Koh & Liang, 2017; Grosse et al., 2023) to compute the influence of pretraining documents on the likelihood of prompt-completions pairs under a trained model. In the extreme case, a language model answering reasoning questions may rely heavily on retrieval from parametric knowledge influenced by a limited set of documents within its pretraining data. In this scenario,

specific documents containing the information to be retrieved (i.e. the reasoning traces) contribute significantly to the model's output, while many other documents play a minimal role. Conversely, at the other end of the spectrum, the model may draw from a broad range of documents that are more abstractly related to the question, with each document influencing many different questions similarly, but contributing a relatively small amount to the final output. We propose generalisable reasoning should look like the latter strategy.

We investigate the pretraining data (called 'documents') that are influential for a set of factual and reasoning questions (called 'queries'). The reasoning questions cover three mathematical tasks; two-step arithmetic, calculating slopes, and solving linear equations. The factual questions require retrieving from parametric knowledge. We experiment with two LLMs (7B and 35B) and 2.5B of their pretraining tokens. Our findings are as follows (summarised in Figure 1):

1. **Procedural knowledge in documents drives influence on reasoning traces**: a document's influence on the reasoning traces of a query is strongly predictive of that document's influence on another query with the same mathematical task (Figure 1 and Finding 1 in Section 5.1). By contrast, this does not hold for factual queries. This indicates that documents often contribute similarly to many questions that require applying the same procedure to different numbers. The correlation is particularly strong for queries involving calculating a slope, and for that task we find procedures for a solution in code or math in the top 0.002% of ranked pretraining data multiple times for most queries (Finding 4 in Section 5.2).

2. **The models rely less on individual documents for reasoning questions, and the set of documents they rely on is less specific**: we find that the magnitude of influence of documents per unit of query information generated by the models is usually much lower for reasoning questions than for factual questions (Finding 2 in Section 5.1). Further, the overall magnitude of influence of the set of documents is less volatile. The former indicates that when generating reasoning traces, the models rely less on each individual document per nat of query information they generate than for factual retrieval. The latter indicates that for a random subset of 2.5B pretraining tokens, it is more up to chance whether highly influential documents are part of it for factual questions than for reasoning questions. Taken together, this indicates the models likely generalise from a more general set of documents for reasoning than for factual questions, relying on each individual document less.

3. **For the factual questions, the answer often shows up as highly influential, whereas for reasoning questions it does not**: we look at the top 500 (top 0.01%) influential documents for each query, and find the answer to factual questions relatively often (55% of the queries for the 7B, and 30% for the 35B), and almost never for reasoning questions, even when we do find the answers in the larger set of 2.5B tokens (Finding 3 in Section 5.2).

4. **We find evidence for code being important for mathematical reasoning**: code data is strongly overrepresented w.r.t. the training distribution for the top portions of the positively and negatively influential rankings for reasoning queries (Finding 5 in Section 5.2).

Our findings suggest a generalisation strategy for reasoning that is unlike retrieval from the parametric knowledge formed during pretraining. Instead, the models learn to apply procedural knowledge extracted from documents involving similar reasoning processes, either in the form of general descriptions of procedures, or applications of similar procedures. This indicates that we may not need to cover every possible case in the pretraining data; focusing on high-quality data demonstrating procedures across diverse reasoning tasks could be more effective. Although our findings are limited to models learning from procedures within the same mathematical task, we observe that code plays a significant role for all tasks we look at. This raises an interesting question: is there a type of pretraining data — such as code — from which models, particularly larger ones, can learn about multiple tasks? Understanding the extent of procedural generalisation can inform future pretraining strategies and help determine where to concentrate data selection efforts.

## 2 RELATED WORK

The subfield with the aim of understanding how large language models generalise is growing rapidly. This question can be approached in different ways, and many recent works interpret weights of smaller models on synthetic tasks to explain particular phenomena that we observe in language

models at scale such as grokking (Wang et al., 2024), in-context learning (Olsson et al., 2022; Singh et al., 2024), or superposition (Elhage et al., 2022; Bricken et al., 2023). Scaling interpretability methods to modern-sized LLMs is challenging for many reasons, of which one is computational tractability. Nonetheless, there are a few works that apply techniques from interpretability to language models at scale. Templeton et al. (2024) use sparse autoencoders to extract interpretable features from Claude 3 Sonnet, and demonstrate how to use these features to control model outputs. Grosse et al. (2023) adapt EK-FAC influence functions (George et al., 2018) to large-scale Transformers, and use them to understand what kind of pretraining data influence completions of models up to 50B parameters. The authors show, among many other things, that larger models rely on pretraining data that are more abstractly related to the completion than smaller models. In this work, we build on the results of Grosse et al. (2023), leaning heavily on their efforts to make influence functions tractable at scale, but focus instead on understanding reasoning specifically.

## 3 COMPUTING THE INFLUENCE OF A DOCUMENT ON A COMPLETION

**Background on influence functions.** Given a pretrained model $\boldsymbol{\theta}^u$ that parametrises a distribution over next tokens conditioned on a prompt $p_{\boldsymbol{\theta}^u}(\mathbf{y}_c \mid \mathbf{y}_p)$ (where $\mathbf{y}_c = \{y_1, \ldots, y_m\}$ is a completion, $\mathbf{y}_p = \{y_1, \ldots, y_n\}$ a prompt, and $u$ indicates the parameters are not necessarily trained to convergence), we are interested in finding data from the pretraining set $\mathcal{D} = \{\mathbf{x}_i\}_{i=1}^N$ that influence the completion. Put differently, we want to know which examples in the pretraining set 'caused' a completion. To this end, we use EK-FAC influence functions for large-scale transformers as proposed by Grosse et al. (2023). The parameters $\boldsymbol{\theta}^u$ are typically found by performing a gradient-based iterative algorithm on an objective function and stopping based on some criterion. We want to know the influence of a training document $\mathbf{x}_j \in \mathcal{D}$ on the parameters $\boldsymbol{\theta}^u$ (which can be reformulated to influence on any continuous differentiable function of $\boldsymbol{\theta}^u$ using the chain-rule). We can calculate influence exactly by removing $\mathbf{x}_j$ from the original training set, re-training the model, and comparing the resulting set of parameters (or a function thereof) to the originally trained model. This is intractable for any interesting number of documents and parameters. Influence functions estimate this counterfactual by taking a Taylor expansion of the *response function* (shown here for optimal parameters):[1] $\boldsymbol{\theta}^\star(\epsilon) = arg\min_{\boldsymbol{\theta} \in \mathbb{R}^D} \mathcal{J}(\boldsymbol{\theta}, \mathcal{D}, \epsilon) = arg\min_{\boldsymbol{\theta} \in \mathbb{R}^D} \frac{1}{N} \sum_{i \neq j} \mathcal{L}(\mathbf{x}_i, \boldsymbol{\theta}) + \epsilon \mathcal{L}(\mathbf{x}_j, \boldsymbol{\theta})$, where $\mathcal{L}(\cdot)$ is a loss function, like the cross-entropy. The first-order Taylor approximation around $\epsilon = 0$ of the response function is used to reason about how the optimal parameters change if you change $\epsilon$, which changes the weight of the document we want to know the influence of. Using the implicit function theorem, *influence* can then be defined as follows: $\mathcal{I}_{\boldsymbol{\theta}^\star}(\mathbf{x}) = \frac{d\boldsymbol{\theta}^\star}{d\epsilon}\Big|_{\epsilon=0} = -\mathbf{H}^{-1}\nabla_{\boldsymbol{\theta}}\mathcal{L}(\mathbf{x}, \boldsymbol{\theta}^\star)$. Where $\mathbf{H} = \nabla_{\boldsymbol{\theta}}^2 \mathcal{J}(\boldsymbol{\theta}^\star, \mathcal{D})$ is the Hessian of the objective. Using the chain-rule, we can estimate influence of a training document $\mathbf{x} = \{x_1, \ldots, x_k\}$ on the completion given a prompt by approximating the following:

$$\mathcal{I}_f(\mathbf{x}) = -\nabla_{\boldsymbol{\theta}} f(\boldsymbol{\theta}^\star)^T \mathbf{H}^{-1} \nabla_{\boldsymbol{\theta}} \mathcal{L}(\mathbf{x}, \boldsymbol{\theta}^\star) \tag{1}$$

Since we are investigating models with billions of parameters $D$, the above Hessian is intractable, and we estimate it using EK-FAC estimation. For a detailed derivation, the reader is referred to Section 2 and 3 in Grosse et al. (2023). We will mention here that it involves estimating two expectations $\mathbb{E}_{p_{\boldsymbol{\theta}}}[\Delta\boldsymbol{\theta}\Delta\boldsymbol{\theta}^T]$ and $\mathbb{E}_{p_{\boldsymbol{\theta}}}[\mathbf{A}\mathbf{A}^T]$ where $\mathbf{A}$ denotes the activations of the model. To make this estimation tractable we make a number of simplifying assumptions across all our estimations, like independence between layers and we only take into account MLP parameters of the transformer layers (Grosse et al., 2023). A full list of approximations can be found in Appendix A.7.

**Adapting EK-FAC influence functions to our problem**. Prior work has shown that EK-FAC influence functions more accurately estimate the counterfactual given by the response function than other types of influence functions (Grosse et al., 2023). However, besides influence on language model completions, we are also interested in influence on the *accuracy* of a trained language model when answering questions. We can only calculate the influence on a continuous differentiable function, and to the best of our knowledge, no work has shown that influence functions also estimate effect on the underlying accuracy of text produced by next-token prediction. As a proxy for accuracy, we take as a continuous differentiable function the cross-entropy loss function ($f$ in Equation

---

[1]The actual response function to derive influence functions for non-converged parameters like $\boldsymbol{\theta}^u$ is the Proximal Bregman response function. The reader is referred to a derivation in Grosse et al. (2023).

Table 1: Example from the reasoning set that involves simple two-step arithmetic.

| Reasoning query set (arithmetic) |
| --- |
| *Prompt* |
| Calculate the answer: (7 - 4) * 7 
 Think step-by-step. |
| *Completion (by 7B)* |
| First, let's evaluate the expression inside the parentheses: 
 7 - 4 = 3 
 Now, let's multiply the result by 7: 
 3 * 7 = 21 
 Therefore, the answer to the expression is 21. |

Table 2: Example from the factual set that requires retrieving the right answer.

| Factual query set |
| --- |
| *Prompt* |
| What is the tallest mountain in the world and how tall is it? |
| *Completion (by 7B)* |
| The tallest mountain in the world is Mount Everest, which is located in the Himalayas. It is 29,029 feet tall. |

1). In Appendix A.1 we show that the influence calculated in this way surfaces documents that have a causal effect on the accuracy of a 7B model fine-tuned to do reasoning and reading comprehension tasks. Namely, if we remove documents from the fine-tuning data according to their influence and re-train the model, the accuracy drops significantly more than if we take out the same amount of documents randomly, or the same amount of documents using gradient similarity. In parallel, we motivate the use of EK-FAC estimations of the Hessian, by showing it significantly improves over a method using only first-order information.

It is only reasonably possible to loop over the pretraining data sample once, and to store more than a single query gradient in memory (which has the same memory complexity as the model itself), Grosse et al. (2023) use singular-value decomposition (SVD). Instead of SVD, we use approximate SVD with a probabilistic algorithm (Halko et al., 2011), which significantly speeds up the computation of the query gradients. We justify each approximation we do in Appendix A.2.1.

We approximate Equation 1 to get scores for documents from the pretraining data $\mathcal{D}$ that represent the influence they have on a completion $\mathbf{y}_c$ given a prompt $\mathbf{y}_p$. Given the counterfactual question approximated by the response function, an influence score of 1 implies the log-probability of the sequence $\mathbf{y}_c$ is increased by 1 (Grosse et al., 2023). To compare influence scores across different completions (and token lengths), we normalise the scores for each query by the information content of its completion $\mathbf{y}_c$, measured in nats. The information content of a query is defined as $\mathbb{I}(\mathbf{y}_c) = -\log\left(p_{\boldsymbol{\theta}^u}(\mathbf{y}_c \mid \mathbf{y}_p)\right)$. The influence scores induce a ranking over documents from most positively to most negatively influential, where a score can be interpreted as the increase (or decrease) in log-probability per nat of query information. The pipeline is shown in Figure 6 in the Appendix.

## 4 EXPERIMENTAL SETUP

**Query set**. We collect a query set with different types of questions, of which 40 are reasoning questions and 40 factual questions. Note that it is only tractable to loop over the pretraining sample we look at once, so we need to be able to store all query gradients in memory and cannot go beyond about 80 questions. For the reasoning questions, we identify two types of mathematical reasoning each model can do robustly with zero-shot chain-of-thought (Wei et al., 2022). We do this by evaluating the models on larger sets of 100 questions for each type of reasoning, and selecting tasks where it gets at least 80% correct. This surfaces simple two-step arithmetic for the 7B model (Table 1), calculating the slope between two numbers for both models (of which two redacted examples are shown in Figure 1), and solving for $x$ in linear equations for the 35B model (see Table 9 in Appendix A.3 for prompt-completion pairs of the linear equations task). We ensure no query ever requires outputting a fraction. To make the results between 7B and 35B more comparable, we use the same slope questions for both models. For the 40 factual questions, we make sure the model gets half right and half wrong, allowing us to identify failures of retrieving facts from parametric knowledge, and we also ensure 16 of 40 overlap between models. We calculate influence over the full completion, which includes the chain-of-thought in the reasoning case. The query sets are provided in the supplement.

**Documents set**. We want to compare the influence of pretraining data on reasoning by differently sized models (7B and 35B), so we select two models that are trained on the same data. The EK-FAC estimation of the Hessian only needs to be done once per model, but the other terms in Equation 1 require two forward- and backward-passes through the model per document-query pair. This means that obtaining a ranking over pretraining data for a single query has a computational complexity similar to pretraining itself. To overcome this issue, we sample a set of documents from the pretraining data that covers multiple examples from each batch seen during pretraining, giving a total of 5 million documents (approximately 2.5B tokens) distributed similary as the training distribution. We batch queries and obtain the influence scores in parallel. Each document contains 512 tokens.[2]

**EK-FAC estimation**. To estimate the Hessian for the 7B and 35B models (the expectations from Section 3), we randomly sample $100\,000$ documents equally spread-out through pretraining for both models. Details on how exactly we approximate the Hessian are in Appendix A.2. We note here that although this aspect of the pipeline requires estimating over 300B parameters representing second-order information, the bottleneck remains calculating document gradients.

**Models**. We look at two models of different sizes, 7B and 35B, which are base and supervised fine-tuned versions of Cohere's Command R series.[3] We estimate the second order information and calculate document gradients using the base models, and generate completions and calculate the query gradients using the models fine-tuned with supervised instruction-tuning. The reason for choosing this setup is that the fine-tuned models are much better at instruction following. This means we are assuming the EK-FAC for the fine-tuning phase is the identity (Bae et al., 2024), and we are focusing only on the influence of the pretraining data and ignoring the fine-tuning data.

## 5 EXPERIMENTS AND RESULTS

We compare the rankings (from most positively to most negatively influential) over pretraining data produced by influence functions for reasoning questions to the rankings for factual questions (which can only be answered by retrieving parametric knowledge). We first analyse the rankings quantitatively by looking at the influence of different documents per nat of generated query information (Section 5.1). We aim to elucidate how generalisable the information in the influential documents is, and how many documents the model is relying on when doing reasoning compared to retrieval. Then, in Section 5.2 we investigate how the documents relate to the queries qualitatively.

### 5.1 QUANTITATIVE ANALYSIS

*Finding 1: There is a significant positive correlation between the influence scores of documents for queries with the same underlying reasoning task, indicating that these documents are relevant for questions requiring the same procedure applied to different numbers.*
If models are relying on documents that contain 'general' knowledge that is applicable to any query with the same task (e.g. queries that require finding the slope between two points for many different points), we would expect there to be a significant correlation in the influence scores for these queries. We calculate the Pearson's R correlation of all 5 million document scores for all query combinations (leading to $80^2$ correlations per model). The results can be seen in the right panel of Figure 1 for a subsample of 10 queries per task, and all query correlations can be found in Figure 12 in Appendix A.9.1. We find a strongly significant (p-values all below $4e-8$) positive correlation between many queries of the same reasoning type, and a strongly significant absence of correlation (p-values all around $4e-3$) for most (but not all) factual queries or other combinations (e.g. reasoning queries of different types). This means that many documents have a similar influence on the same type of reasoning. Given that each type of reasoning query requires applying the same procedure to different numbers, the positive correlation indicates that the influence scores for reasoning queries pick up on procedural knowledge. The correlations are strongest for the slope queries by the 35B model, and this is also the type of reasoning the model can do most robustly compared to solving linear equations. For the model to be able to solve linear equations with an accuracy of more than 80%, we restrict the calculations to lead to positive $x$, whereas for the slopes questions the answers can be positive or negative. In Appendix A.9.1 we falsify the hypothesis that the correlations are

---

[2]We choose 512 tokens because qualitatively interpreting more is hard (usually spanning multiple topics).
[3]https://cohere.com/command

caused by the fact that the reasoning questions are superficially similar to each other, by using a set of control queries that are also superficially similar but do not require any reasoning and repeating the entire experiment. For the control queries we mostly do not observe a correlation. In Appendix A.9.1 we highlight examples of queries with high or low correlation for different query sets, finding that some of the correlation seems driven by formatting of reasoning steps, and most by reasoning procedure.

***Finding 2: When reasoning, the model on average relies on each individual document less per generated nat of information than when answering factual questions, and the total magnitude of influence is much less volatile, indicating it is generalising from a more general set of documents. The effect is more pronounced for the larger model.***
In Figure 2 we show the total influence for different percentiles of the positive parts of the rankings.

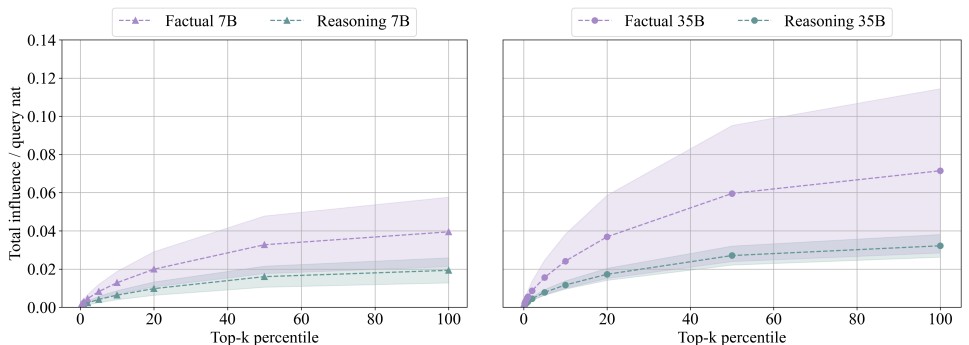

Figure 2: The total influence per nat of query completion information for different portions of the positive ranking over documents, left for the 7B model, right for the 35B. The total influence per nat is usually lower for reasoning questions than for factual questions, and the influence per document varies more for factual questions than for reasoning questions, especially for the 35B model.

The results depict the total amount of influence contained in the top-$k$ percentile of the positively ranked documents: e.g. the 20th percentile contains 20% of the positive documents for a query, and the amount of total influence shown is the sum of all document influences up to that part of the ranking. The equivalent for the negative portions looks similar (Figure 15 in Appendix A.9.2) and the discussion below applies similarly to the negative ranking. We observe two things for both models. Firstly, the amount of total influence for most factual questions at any part of the ranking is higher than for reasoning questions. Secondly, there is more variation in the influence of documents at the same rank across different factual queries (and for a few factual queries the amount of influence is actually lower than for the reasoning queries, seen more clearly in Figure 20 in Appendix A.9.3). The first result means that, on average, the models rely on individual documents within our set less for generating reasoning traces than for answering factual questions. The second result indicates that for the factual questions the model relies on more 'specific' and infrequent documents: for a factual question it is more up to chance whether relatively highly influential documents (w.r.t. influence of documents for other factual questions) are part of the pretraining sample or not.

**Influence spread.** Another way to analyse the magnitude of influence is to look at the dispersion of influence across the ranking: how much of total influence for each query is contained at the top and bottom parts of the ranking? Similarly to what Grosse et al. (2023) report, we observe that the top parts of the rankings over documents follow a power law characterised by a linear relation between rank and influence per nat in log-log space (shown in Figure 20 in Appendix A.9.3). We find that the slopes for the reasoning questions by the 35B are slightly steeper than for the factual questions, and therefore the percentage of positive influence contained in the top portions of the rankings for the 35B reasoning questions increases faster with rank than for the factual questions (shown in Figure 22 in Appendix A.9.3). For the 7B, the slopes for the reasoning questions the model gets right are on average also a bit steeper than for the factual questions, but the effect goes away when comparing slopes for all factual vs. reasoning queries. This means that the percentage of the total positive influence the top sequences cover is higher for the reasoning questions than for the factual questions for the 35B model (and similarly for the bottom sequences, see Figure 15). There

is a chance this finding is caused by noise for the 35B model and we discuss this possibility more in Appendix A.9.3, where we note that for the reasoning query with the steepest power law, the top 1 document is qualitatively entirely unrelated to the prompt.

If we compare the result between models, we find that the difference in magnitude and volatility are more pronounced for the 35B model across the full rankings. We look into this in Appendix A.9.2, and find that the effect remains even if we only look at queries that are the same for both models, which points to higher data efficiency for the larger model.

## 5.2 QUALITATIVE ANALYSIS

We perform three qualitative analyses on the top portions of the rankings for each query; we search for the answer, we characterise the documents' relation to the reasoning queries, and we investigate what source datasets they are from (for both the top and bottom parts of the ranking, e.g. code, Wikipedia, etc). To filter some of the noise, we divide the influence scores by the document gradient norm and re-rank them, which has empirically been found to help (Choe et al., 2024).

***Finding 3: The answer to the factual questions shows up relatively often in the top influential documents for the factual questions, and almost never for the reasoning questions.***

To find the answer to the questions in the queries in the top documents manually, we construct keywords for each query that should be in the document if the answer is there. For example, for the factual query in Table 2, the keywords are "tallest", "highest", "Mount Everest", "29029", "8848". For the reasoning queries, we construct many more keywords per query, but some examples for the example in Table 2 are $7 - 4$, $3$, $21$, $3*7$, as well as replacing the operations with words like 'minus' and 'times', and different ways of representing the content in this query. For details on which keywords we use for each query, see Appendix A.4. We determine the occurrence of each of these keywords independently in the top 100 documents for each query (meaning even if just the keyword '7' is present it would be a hit), resulting in many false-positives. We manually look over the hits to find the answer. On top of that, we craft a prompt for Command R+ (a more capable 100B model) to find the answer in a query-

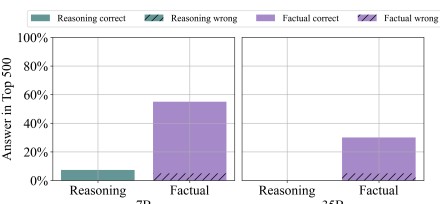

Figure 3: We search for the answer in the top 500 (top 0.01%) documents, and find it relatively frequently for the factual questions. For the reasoning questions, we find the answer twice for the 7B, and never for the 35B. Both those times, the answers to the steps occur in separate documents.

document pair, and use it to find the answer in the top 500 documents for each query independent of keyword overlap (the prompt is given in Appendix A.5). Then, we manually look over the hits and keep track of documents that have the answer to a query. We verify that Command R+ finds all, and more, of the answers we have identified manually. We look for the full answer in a single document. For the reasoning queries, we also count partial answers in separate documents if they combine to the full answer. For example, if one document contains $7 - 4 = 3$, and another $3 * 7 = 21$, we consider that an answer. Finally, we apply the keyword overlap search combined with prompting Command R+ to a subset of the broader 2.5B pretraining tokens to verify that the answer to the questions are in the entire set even if they do not show up in the top 500 documents for queries.

The results are shown in Figure 3. For the 7B model, we find the answer in the top 500 documents for 55% of the factual queries, compared to 7.4% of the reasoning queries. For the 35B model, the answer to the factual queries shows up in the top influential documents 30% of the time, and never for the reasoning set. We expect the answer shows up less frequently for the 35B model simply because the factual questions are much more 'niche'. For example, one of the questions the model gets correct is *"In which year did the Beinecke Library open?"*. Moreover, in certain cases, the answer shows up multiple times in the top 500 documents. If we count all these separately, as opposed to a binary 'yes' or 'no' per query on which the results in Figure 3 are based, answers to questions show up 30 times for the factual questions in the 7B rankings, and twice for the reasoning questions. For the 35B, the same result is 15 times for the factual questions, and never for the reasoning questions. Interestingly, the answer to the factual questions often shows up in different languages, like Spanish or Portuguese. We give two examples in Appendix A.8.2. To falsify the

hypothesis that the answers to reasoning questions are not showing up because they are not present in the set of 5M documents, we repeat the above keyword search over a random subset of the 5M documents. We identify answers to reasoning steps in documents that do not show up in the top 500 documents for 13 of 20 arithmetic queries and a full answer for 1 of 20, and expect more to be there that elude the keyword search. For the slopes and linear equation queries, we find answers to 3 reasoning steps which do not show up in the top 0.01%. In Appendix A.8.1 we show some of these documents and their ranks.

***Finding 4: We find that influential documents for the reasoning queries are often doing a similar form of step-by-step reasoning, e.g. also arithmetic. Further, we find that the influential documents often implement a solution to reasoning questions in code or general math.***

For the slope queries (of which we have 20 which are the same for both models), many different documents surface as highly influential that show how to calculate the slope between two points in code or math. For the 7B model, documents that present procedural knowledge on how to calculate the slope in either code or math show up in the top 100 documents for 16/20 queries (38 times), and for the 35B model they show up for all queries (51 times). All together, we manually find 7 unique documents that implement the slope in code in the top 100 documents, and 13 that present equations for calculating the slope. The 7B model relies on 18 of these documents for its completions (meaning 18 different ones appear in the top 100 documents for all queries), and the 35B on 8. An example of a highly influential document implementing the solution in JavaScript (left) and in maths (right):

<table>
<tr><td>Positively influential code</td><td>Positively influential math</td></tr>
<tr><td>

```
function eqOfLine(x1, y1, x2, y2) {
  if (x1 === x2) {
    // Handle a vertical line
    return `x = ${x1}`;
  } else {
    // Calculate the slope
    const m = (y2 - y1) / (x2 - x1);
    const b = y1 - m * x1;
    // Return y = mx + b
    return `y = ${m}x + ${b}`;
  }
}
```

</td><td>

If a straight line passing through the points $P(x_1, y_1), Q(x_2, y_2)$ is making an angle $\theta$ with the positive $X$-axis, then the slope of the straight line is:

(A) $\frac{y_2 + y_1}{x_2 + x_1}$

(B) $\theta$

(C) $\frac{y_2 - y_1}{x_2 - x_1}$

(D) $\sin \theta$

**Solution:**
Correct answer: (C)

</td></tr>
</table>

We prompt Command R+ to further characterise the top 500 documents for each query by choosing from a set of provided keywords, and find that often the documents are doing similar arithmetic on other numbers (e.g. much larger or smaller), doing similar arithmetic on similar numbers (for the slope questions), or similar algebraic operations on similar numbers (for solving linear equations). We present the detailed results and prompt for this analysis in Appendix A.8.3.

***Finding 5: For factual queries, the most influential data sources include Wikipedia and trivia, while for reasoning, key sources consist of maths, StackExchange, ArXiv, and code.***

We look at the type of source datasets that represent the most influential documents. Specifically, we count the source datasets of the top and bottom $k$ documents with $k \in \{50, 500, 5000, 50000, 500000\}$, and compare the count to the pretraining distribution. We present the details in Appendix A.8.4, but mention here that code data is highly influential for reasoning. StackExchange as a source has ten times more influential data in the top portions of the rankings than expected if the influential data was randomly sampled from the pretraining distribution. Other code sources are twice as influential as expected when drawing randomly from the pretraining distribution for $k = 50$ up to $k = 50000$. Similar patterns hold for the bottom portions of the rankings.

## 6 DISCUSSION, LIMITATIONS, AND FUTURE WORK

In this work, we investigate what kind of generalisation strategy two LLMs (7B and 35B respectively) employ when reasoning, and contrast it to the strategy used for a task that requires retrieving factual parametric knowledge. By creating rankings for 200 such questions over 5 million pretraining documents based on their influence on the likelihood of the completions, we conclude that the generalisation strategy for reasoning is unlike retrieval. More often than not, even if the answer is part of the set of pretraining documents we look at, it does not show up as highly influential as the

answers to factual questions do. We find that instead, the positively influential documents often contain procedural knowledge on how to get to a solution. Further, the models rely less on individual documents when reasoning than when answering factual questions, and the set of documents they rely on is more general. Finally, documents often have a similar influence on reasoning queries that require applying the same procedure to different numbers. These findings can inform pretraining data selection for more robust reasoning: we likely do not need to cover every case in pretraining but can rather focus on data describing and applying procedures to diverse reasoning problems.

We find that the distribution of influence is less spread out for reasoning than for factual questions, characterised by steeper power laws. The distribution of influence over documents tells us something about the type of generalisation strategy the model is using; the more documents that contribute to each nat of query information (i.e. the more spread out the total influence), the more documents the model is relying on to produce the completion. One would perhaps expect a steeper power law for factual questions than for reasoning (meaning more of the total positive influence contained at the top parts of the ranking), but our results show evidence for the opposite. Perhaps a model needs to generalise from a broader set of documents for factual retrieval than for reasoning because it needs to see the same information more often to memorise it. This is supported by the finding that for factual questions the answer often shows up multiple times in the top 0.01% most influential data.

There are important limitations to our approach, most notably that we do not calculate influence on the entire training set, which is intractable. An alternative explanation of our results is then the opposite conclusion: the model is relying on data for reasoning that are so infrequent that a random sample of 2.5B tokens does not surface relatively highly influential samples for any of the 60 unique reasoning queries. This would result in the conclusion that LLMs rely on sparse and infrequent documents for reasoning. That means we are effectively looking at a set of relatively uninfluential documents for reasoning, and that perhaps the answers to reasoning traces would be highly influential when looking at the entire pretraining data. We would argue that this is the more unlikely explanation for three reasons: (1) the qualitative analysis shows that the influential data for the reasoning questions are intuitively highly relevant, and that the answers to many reasoning traces *are* part of the 2.5B tokens, they are just not highly influential for reasoning, (2) the correlation of influence scores for the different reasoning tasks is highly significant, and (3) we confirm that these results do not hold for control queries that look similar to the reasoning queries superficially, but do not require step-by-step reasoning. Moreover, it seems unlikely that the model is learning to do retrieval from such infrequent data for one of the simplest forms of mathematical reasoning, namely subtraction and multiplication on small numbers. Taken together we argue the results indicate a generalisation strategy that relies on procedural knowledge. Regardless, the nature of interpretability research such as the work presented here is that all we can do is provide evidence, and not proof.

Another limitation is that we do not look at the supervised fine-tuning stage. The reason we only look at the pretraining data is because the fine-tuning stage is targeted at making the models more aligned and 'instructable', and prior work has shown that SFT serves primarily to enhance existing model capabilities (Jain et al., 2024; Kotha et al., 2024; Prakash et al., 2024). Nonetheless, an interesting direction for future work is applying the same method used here to the fine-tuning data.

This work spurs further avenues for future work. Firstly, as previously discussed, identifying data types that are similarly influential across reasoning types could provide additional insight into data selection techniques for improved reasoning. Relatedly, what properties of code data makes it influential for reasoning? What kind is positively influential, and what kind negatively? Further, since we only take into account the feed-forward layers and treat the attention as fixed, an interesting avenue for future work would be to investigate how the relatively low magnitude of influence of pretraining data on feed-forward parameters for reasoning traces interacts with attention, connecting to a finding from literature that certain forms of reasoning happen in the attention heads (Olsson et al., 2022). Finally, in this work we investigate mathematical reasoning. Future work should verify whether similar results hold for other types of reasoning, such as inductive reasoning.

With this work, we do not claim to say contamination is not an issue, or that LLM reasoning is not brittle and reliant on pretraining statistics. What we demonstrate is that, in principle, it appears possible for LLMs to produce reasoning traces using a generalisation strategy that combines information from procedurally related documents, as opposed to doing a form of retrieval. This is not to say that there are no cases of LLM reasoning where the model is in fact doing retrieval, on the contrary, models can be overfit to contaminated data if it appears often enough in the training data.

REPRODUCIBILITY STATEMENT

Although this work is based on proprietary models and pretraining data, we make the following efforts for reproducibility. We add pretraining data with answers to factual and reasoning questions to the supplement, as well as data in which procedures for calculating the slope have been identified. For one of the models we use (the 35B model), the final-stage model (further trained after SFT) is publicly available on HuggingFace.[4] We provide all queries, completions, and keywords in the supplemental material. Additionally, we verify that the influence scores generated with our internal codebase correlate with a Pearson's R of more than 0.99 with a public implementation of EK-FAC influence functions (see Appendix A.2.2). Further, we provide details on hyperparameters for every experiment we have done at the relevant sections, as well as the prompts used to find answers to the reasoning questions and characterise the relationship between the query-document pairs (Appendix A.5 and A.6 respectively).

ACKNOWLEDGEMENTS

We'd like to thank Andrew Lampinen, Stephanie Chan, Akbir Khan, and Philipp Jettkant for fruitful discussions about the work presented here. This work was supported by the EPSRC Grant EP/S021566/1 and UCL International Scholar Award for Doctoral Training Centres.

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

## A  APPENDIX

Below we outline the contents of the appendix.

**EK-FAC influence functions.** In Appendix A.1 we discuss the counterfactual re-training experiments that motivate our use of EK-FAC influence functions for estimating the effect of pretraining data on the accuracy of downstream behaviour. We describe in more detail how we use influence functions at scale in Appendix A.2, documenting how we estimate the Hessian, how we store many query gradients in memory (each having the same memory complexity as the entire model), and how we sample from the pretraining distribution.

**Query sets examples.** Then, in Appendix A.3, we show examples of the reasoning sets that we did not show examples for in the main body of this manuscript.

**Finding query answers in documents and characterising document-query relations.** In Appendix A.4 we discuss how we create keywords for each query in order to find the answer in the top documents, and in the sections directly after that, Appendix A.5 and A.6, we give the prompts we used to allow Command R+ to search for answers in the top 500 documents for each query, as well as characterise their relationship.

**Limitations.** In Appendix A.7 we discuss limitations specific to influence functions.

**Additional qualitative results.** In Appendix A.8 we provide additional qualitative results.

*Answer finding*. We show examples of answer documents in Appendix A.8.1.

*Cross-lingual transfer*. We give some examples of cross-lingual transfer in Appendix A.8.2.

*Characterise query-document relation*. We give detailed results on the characterisation of the relationship between queries and the top 500 documents in Appendix A.8.3.

*Source-dataset analysis*. We analyse which datasets the influential data comes from in Appendix A.8.4.

*Content analysis of relevant documents*. We classify data from the source dataset code for whether it actually contains code in Appendix A.8.5.

**Additional quantitative results.** In Appendix A.9 we provide additional quantitative results.

*Correlation analysis*. Further results for the correlation analysis of influence scores for documents for different queries in Appendix A.9.1.

*Magnitude of influence*. Further results for the magnitude of influence in Appendix A.9.2.

*Spread of influence*. Further results for the spread of influence over the rankings in Appendix A.9.3.

## A.1 COUNTERFACTUAL RE-TRAINING EXPERIMENTS WITH INFLUENCE FUNCTIONS

We use EK-FAC influence functions to approximate the counterfactual question: which documents from pretraining have a causal effect on the completions of a trained model. However, we are also interested in the causal effect on the *accuracy* of the completions. In this section, we aim to motivate two aspects of this choice; the fact that influence functions are designed to estimate the effect on continuous differentiable functions, like the log-likelihood, and not on the accuracy. Secondly, we motivate the need for estimating the second-order information of the pretraining objective using EK-FAC, which is very computationally expensive. We present four different experiments in this section, which show that indeed the influence of documents as determined by influence functions also estimate the effect on downstream task accuracy, as well as the benefits from estimating second order information over simply using first-order gradient information.

The pipeline for each of these experiments is similar; we take a pretrained model, we fine-tune it on some dataset, and evaluate it on 50 validation examples with a metric (perplexity or accuracy). We then use the fine-tuned weights to calculate the influence of the documents in the dataset used for fine-tuning on the set of 50 validation questions with two methods: EK-FAC influence functions and TracIn (Pruthi et al., 2020). Subsequently, we use those two methods to remove the $k$ most positively influential documents from the fine-tuning dataset, as well as randomly selecting $k$ documents as a baseline, and fine-tune the original pretrained model five times (with different seeds) on each new fine-tuning dataset created (for different values for $k$). We then calculate the perplexity or accuracy on the validation questions used to calculate the influence, and see how it changed. The more it changed, the more the documents indeed influence the relevant metric (i.e. perplexity or accuracy). Note that for $n$ different values for $k$, this requires fine-tuning $3 * 5 * n$ models: five times for each of the three methods of removing documents from the training set.

We start by motivating the use of EK-FAC influence functions over simple similarity information between document and query gradients. In our setup, where we only have access to the final checkpoint of pretraining, a dot-product between the query and document gradient effectively boils down to a method for estimating influence of documents on queries called TracIn (Pruthi et al., 2020). With access to multiple checkpoints, TracIn uses gradient information from all of them, accounting for the learning rate used at that point in training. However, we only use the final checkpoint and hence taking into account learning rate only changes scores by a constant. We take GPT-2-small (124M) from HuggingFace,[5] and fine-tune it for three epochs with next-word prediction on Wikitext-2 (Merity et al., 2016). We use Adam optimizer (Kingma & Ba, 2015) with default parameters (b1 0.9, b2 0.999, eps 1e-8, additive weight decay 0.01). The results can be found in Figure 4 and Table 3, showing that removing documents using EK-FAC influence functions has a significantly larger effect on downstream perplexity for all values of $k$. We do the exact same experiment but instead remove the most negatively influential documents, and see that instead the perplexity decreases significantly more for EK-FAC influence functions (Figure 4 and Table 4).

Table 3: Wikitext remove top influential

| k → | 50 | 100 | 150 | 200 | 250 | 300 |
|---|---|---|---|---|---|---|
| Random | $22.09 \pm 0.02$ | $22.12 \pm 0.02$ | $22.10 \pm 0.02$ | $22.20 \pm 0.06$ | $22.19 \pm 0.05$ | $22.15 \pm 0.05$ |
| TracIn | $22.16 \pm 0.02^{\star\star}$ | $22.22 \pm 0.02^{\star\star}$ | $22.25 \pm 0.01^{\star\star}$ | $22.35 \pm 0.03^{\star\star}$ | $22.42 \pm 0.01^{\star\star}$ | $22.45 \pm 0.02^{\star\star}$ |
| IF (ours) | $\underline{22.49} \pm 0.02^{\star\star}$ | $\underline{22.66} \pm 0.02^{\star\star}$ | $\underline{22.73} \pm 0.02^{\star\star}$ | $\underline{22.88} \pm 0.01^{\star\star}$ | $\underline{22.97} \pm 0.02^{\star\star}$ | $\underline{23.05} \pm 0.05^{\star\star}$ |

Table 4: Wikitext remove bottom influential

| k → | 50 | 100 | 150 | 200 | 250 | 300 |
|---|---|---|---|---|---|---|
| Random | $27.40 \pm 0.08$ | $26.24 \pm 0.10$ | $25.62 \pm 0.15$ | $25.22 \pm 0.10$ | $25.04 \pm 0.12$ | $24.85 \pm 0.10$ |
| TracIn | $26.73 \pm 0.04^{\star\star}$ | $25.48 \pm 0.05^{\star\star}$ | $24.86 \pm 0.02^{\star\star}$ | $24.36 \pm 0.04^{\star\star}$ | $24.16 \pm 0.05^{\star\star}$ | $23.94 \pm 0.03^{\star\star}$ |
| IF (ours) | $\underline{25.96} \pm 0.04^{\star\star}$ | $\underline{24.78} \pm 0.05^{\star\star}$ | $\underline{23.95} \pm 0.03^{\star\star}$ | $\underline{23.52} \pm 0.03^{\star\star}$ | $\underline{23.46} \pm 0.03^{\star\star}$ | $\underline{23.32} \pm 0.04^{\star\star}$ |

Next, we turn to motivating the use of EK-FAC influence functions in estimating the effect of documents on downstream accuracy of model generations. To this end, we look at two different datasets:

---

[5] https://huggingface.co/

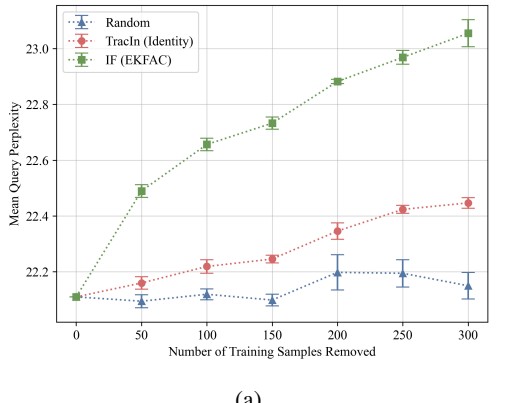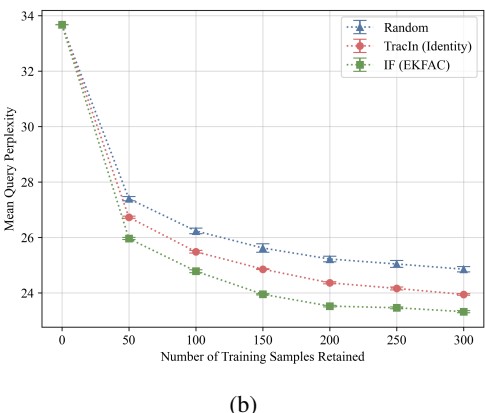

(a)                  (b)

Figure 4: (a) Counterfactual retraining experiments on Wikitext-2. We finetuned GPT-2 (124M) on Wikitext-2 and use three different methods to remove training examples from the training set: randomly, TracIn, and Influence Functions (IF). For each number of samples removed we finetune the base model five times with different training data ordering, the variance over these runs is represented by the error bars. Each point on the plot is the average perplexity achieved by the five models after fine-tuning on the augmented dataset. We find that influence functions can find examples that impact the perplexity significantly more than baselines. (b) We repeat the same experiment as in (a), but retain top influential queries instead (removing most negatively influential).

DROP (Dua et al., 2019) and RACE (Lai et al., 2017). DROP is a reading comprehension dataset requiring different skills like subtraction, addition, coreference resolution, counting, and other skills. The model needs to generate an answer that often consists of one or a few words. We allow the fine-tuned models to generate answers to the questions freely, and evaluate based on exact match. In this experiment, we use a 7B model. We randomly select a subset of 8000 examples for fine-tuning, and use the procedure described above to perform counterfactual experiments. We use Adam optimizer again, with the same hyperparameters as for the above experiment: b1 0.9, b2 0.999, eps 1e-8, additive weight decay 0.01, but only train for one epoch. The results can be found in the left panel of Figure 5 as well as in Table 5. We find that EK-FAC influence functions are succesful in selecting data points that impact downstream accuracy, much more so than randomly removing the same amount of training data. For most $k$ (all but $k = 1000$), EK-FAC influence functions also have a significantly stronger effect on accuracy than TracIn, but the difference is less large. We apply the exact same procedure to the RACE dataset, except now we keep 10k examples (empirically found to lead to the least overfitting when fine-tuning). Further, RACE is a multiple-choice dataset, so we allow the model to generate a single token indicating the choice, and calculate the accuracy. The results can be seen in Figure 5 and Table 6. Again, the finding is similar; EK-FAC influence functions surface documents that have a stronger effect on accuracy than TracIn for all but one value of $k$, and for all values of $k$ than randomly removing documents. There is a large variance in the results for all methods though, which we attribute to the fact that the model sometimes seems to overfit to the fine-tuning data. Further, the reason why the difference between TracIn and EK-FAC influence functions is much larger in the perplexity experiments than in the accuracy experiments could be attributed to the fact that we only fine-tune for one epoch in the accuracy experiments (as more cause overfitting). EK-FAC influence functions differ from TracIn in that they estimate second order information, which becomes more important with more training steps. An interesting avenue for future work is to do counterfactual re-training experiments like these on a subset of pretraining data for a 7B model, but this is incredibly computationally expensive.

Although the results of the experiments in this section are an encouraging sign for using EK-FAC influence functions in estimating causal effect of data on accuracy, it is important to note that they are limited in several ways. Accuracy is a discrete metric and it is a prior unclear how many documents need to be removed to flip its value. However, the influence functions we use estimate effect of removing a single document, and removing multiple documents can have additional effects that are unaccounted for. This makes removing multiple documents a cruder way to empirically show impact of influence functions on accuracy, but at the same time it is unavoidable. Therefore, any significant

Table 5: Counterfactual re-training accuracies on DROP (free generation of answers). We use three different methods (random, TracIn, influence functions) to remove $k$ datapoints, and re-train a model on the resulting dataset. Each number is the mean over five re-training runs with different data ordering. $\star$ indicates significantly lower than random with a p-value below 0.1 and $\star\star$ with a p-value below 0.05. The underlined means are the lowest.

| $k \rightarrow$ | 500 | 1000 | 1500 | 2000 |
|---|---|---|---|---|
| Random | $0.61 \pm 0.05$ | $0.60 \pm 0.03$ | $0.56 \pm 0.05$ | $0.57 \pm 0.06$ |
| TracIn | $0.55 \pm 0.03^\star$ | $\underline{0.49} \pm 0.02^{\star\star}$ | $0.44 \pm 0.04^{\star\star}$ | $0.43 \pm 0.06^{\star\star}$ |
| IF (ours) | $\underline{0.51} \pm 0.03^{\star\star}$ | $0.50 \pm 0.04^{\star\star}$ | $\underline{0.40} \pm 0.05^{\star\star}$ | $\underline{0.38} \pm 0.05^{\star\star}$ |

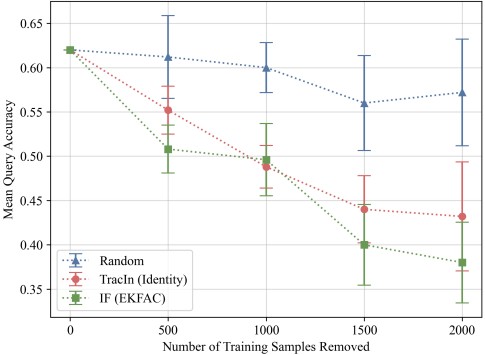
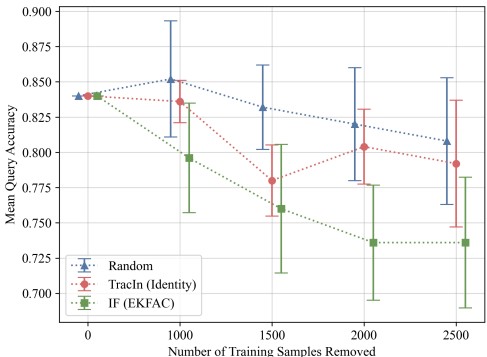

(a) Counterfactual retraining experiments on reading comprehension questions. We finetuned Cohere Command 2 (7B) on a subset of the DROP training set (8k examples) and use three different methods to remove training examples from the training set: randomly, TracIn, and Influence Functions (IF). For each number of samples removed we finetune the base model five times with different training data ordering, the variance over these runs is represented by the error bars. Each point in the plot is the average accuracy achieved by the five models after fine-tuning on the augmented dataset. We find that influence functions can find examples that impact the accuracy significantly more than baselines, although only slightly more than TracIn.

(b) Counterfactual retraining experiments on multiple-choice reasoning data. We finetuned Cohere Command 2 (7B) on a subset of the RACE training set (10k examples) and use three different methods to remove training examples from the training set: randomly, TracIn, and Influence Functions (IF). For each number of samples removed we finetune the base model five times with different training data ordering, the variance over these runs is represented by the error bars. Each point in the plot is the average accuracy achieved by the five models after fine-tuning on the augmented dataset. We find that influence functions can find examples that impact the accuracy significantly more than baselines, although there is some variance in the results.

Figure 5: Counterfactual retraining experiments on reading comprehension benchmark DROP (a) and the multiple-choice reasoning dataset RACE (b).

causal effect on accuracy over other methods is a good signal, but the absence of a significant effect does not necessarily mean EK-FAC influence functions do not properly do what they are designed to do.

Table 6: Counterfactual re-training accuracies on RACE (multiple-choice). We use three different methods (random, TracIn, influence functions) to remove $k$ datapoints, and re-train a model on the resulting dataset. Each number is the mean over five re-training runs with different data ordering. $\star$ indicates significantly lower than random with a p-value below 0.1 and $\star\star$ with a p-value below 0.05. The underlined means are the lowest.

| k $\rightarrow$ | 1000 | 1500 | 2000 | 2500 |
|---|---|---|---|---|
| Random | $0.85 \pm 0.04$ | $0.83 \pm 0.03$ | $0.82 \pm 0.04$ | $0.81 \pm 0.04$ |
| TracIn | $0.84 \pm 0.01$ | $0.78 \pm 0.03\star\star$ | $0.80 \pm 0.03$ | $0.79 \pm 0.04$ |
| IF (ours) | $\underline{0.80} \pm 0.04\star$ | $\underline{0.76} \pm 0.05\star\star$ | $\underline{0.74} \pm 0.04\star\star$ | $\underline{0.74} \pm 0.05\star$ |

## A.2 EK-FAC INFLUENCE FUNCTIONS

The code we use for EK-FAC influence functions at scale is a part of larger internal infrastructure, and hence cannot be released publicly. However, we base our code on the public GitHub repository `https://github.com/pomonam/kronfluence`. We implement estimation of the Hessian in the same way as in that codebase, except for a few changes to make it tractable, which we discuss in more detail below. Further, we compare the results produced by our implementation with the results using the public implementation. We do this by fine-tuning GPT-2 (124M) on Wikitext-2 using internal infrastructure, and calculating influence scores with both code bases. We find that the results correlate very strongly (with a Pearson's R of more than 0.99, see A.2.2 below for more details). Here, we provide details of the design choices and hyperparameters used in our implementation, as well as the additional approximations to make EK-FAC estimation and influence calculation tractable at scale.

**Query-batching and approximation** As mentioned in the main text, we approximate query gradients using approximate SVD (Halko et al., 2011). We use the default parameters for this algorithm, which can be found in the Dask documentation (Dask Development Team, 2016).

**Sampling from the Pretraining Data**. It is intractable to calculate influence for the entire pretraining data, so we sample a set of 5 million documents. To this end, we loop over the training data as seen by the models in order, and randomly sample 6 examples from each batch. This ensures that the pretraining sample we use is both similar to the pretraining distribution in terms of what kind of data the model sees, as well as when it has encountered the data during pretraining.

**Estimating EK-FAC**. To estimate the EK-FAC matrices, we sample $100\,000$ documents from pretraining in the same manner as described above. We use the same samples to estimate the EK-FAC for the 7B as for the 35B. For both models, we use a damping factor of 0.1 (see Grosse et al. (2023) for details on what the damping factor is). Further, part of estimating the EK-FAC is an eigendecomposition on the EK-FAC matrices. We use the same approximation as empirically motivated in (Grosse et al., 2023), namely block-diagonal approximation. For the 7B, we use 2 blocks, and for the 35B, we use 4. The block-diagonal approximation is not part of the public codebase, but simply amounts to dividing the matrices in $n$ blocks (where $n$ is 2 and 4 in our case), zero-ing out the remaining entries, and taking the eigendecomposition of each block individually. After, these blocks are patched back together again into the original size matrix, which will be further processed as in the public codebase.

### A.2.1 JUSTIFYING APPROXIMATIONS

In this section, we justify the additional approximations we do on top of those mentioned in Grosse et al. (2023) by reporting the correlation with the full implementation for a smaller model (124M parameters). Applying EK-FAC influence functions to models with billions of parameters requires estimating a multiple of the model parameters. E.g., for the 7B model we estimate around 70B EK-FAC parameters, and for the 35B model we estimate around 320B parameters. Further, to calculate the influence scores for a set of 5 million documents we have to calculate the gradient for 100 queries $\times$ 5 million documents, each of which has the same size as all feed-forward layers in the model itself. We can only afford to loop over the 5 million documents and calculate their gradients once, so we need to batch the query gradients in memory. This is impossible for the full gradients and we use SVD to store low-rank approximations instead, like in Grosse et al. (2023).

**Details on the experiment.** To compare results of using EK-FAC influence functions with different approximations, we use the same fine-tuned model from Section A.1 to calculate influence scores for the 4656 training examples (i.e. documents) on the first 32 validation examples (i.e. queries) of the Wikitext-2 dataset. We repeat this with different types of approximations applied; full SVD on the query gradients, approximate SVD (Dask Development Team, 2016) on the query gradients, and a block-diagonal approximation of the EK-FAC matrices before the eigendecomposition (described in Appendix A of Grosse et al. (2023)) with 2 and 4 blocks. For each level of approximation applied, this gives us 32 vectors with 4656 scores (one for each query-document pair), and we compare these to the full implementation without SVD and block diagonal approximations using Pearson's R correlation. The correlations reported are the average over all 32 queries, but in the supplement we provide the correlations for each query for all experiments done below.

In Table 7 we report the correlations of increasingly more approximations w.r.t. a full implementation. Note that the full implementation also uses approximations, but those are all justified in Grosse et al. (2023). Here, for completeness, we additionally justify the approximations we use that are different, namely approximate SVD instead of full SVD, and a block-diagonal approximation with 4 blocks instead of 2. From Table 7, we can see that the approximate SVD algorithm has a neglible effect on the scores, whereas the block-diagonal approximation has a small effect on the scores.

| Approximations | Pearson R |
|---|---|
| SVD | $0.96 \pm 0.01$ |
| Approximate SVD | $0.96 \pm 0.01$ |
| Approximate SVD + block diagonal EK-FAC (2 blocks) | $0.95 \pm 0.00$ |
| Approximate SVD + block diagonal EK-FAC (4 blocks) | $0.93 \pm 0.00$ |

Table 7: Score correlations of using increasingly more approximations with a full implementation.

### A.2.2 FULL IMPLEMENTATION

We also compare the full implementation scores of our own influence functions implementation with the scores calculated for the same model and dataset with the public implementation at `https://github.com/pomonam/kronfluence`, and confirm the average score correlation between queries is 0.993 ($\pm$ 0.003). We add a direct score comparison of both methods for the top 3 documents for each of the 32 queries to the supplemental material. Specifically, for each query we log the top 3 documents as determined by our internal implementation as well as the external implementation, showing that they are almost always the same documents, and logging the score given to that document by each implementation (the supplemental file also contains the score correlation for each query separately). The average number of documents that appear in both top 50's determined by the internal and external implementation is 46.7. The reason for using an internal implementation nonetheless is that the public implementation is not optimised for usage on large-scale models, and cannot be used for models above about 1B parameters. We used the internal pretraining library for implementing influence functions, because part of the infrastructure used for pretraining large models could be re-used.

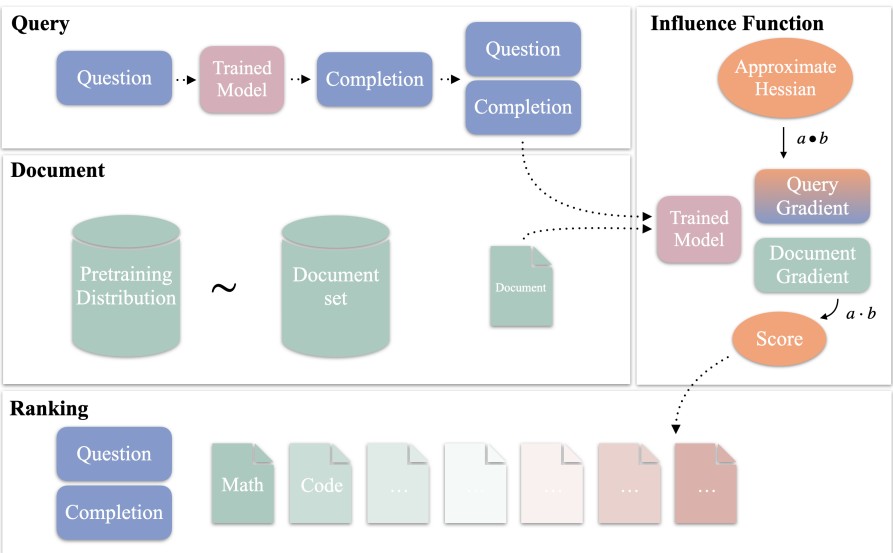

Figure 6: The pipeline for creating rankings of the most influential pretraining documents for a question-completion pair (*query*) using influence functions. The documents at the top of the ranking influence the likelihood of the completion positively, and the bottom negatively. We create rankings for a set of 40 reasoning, 40 factual, and 20 control queries over 5 million pretraining documents (2.5B tokens) for two models of different sizes (Cohere's Command R series, 7B and 35B).

A.3   QUERY SETS

**Reasoning query sets.** We show an example of the other two types of reasoning present in the reasoning query sets in Table 8 and 9. The former requires calculating the slope of a line going through two given points (used for both the 7B and 35B model) and the latter is about solving for $x$ in a linear equation (only used for the 35B model).

**Control query sets.** We design two control sets with 10 questions each for both the 7B and 35B model. These query sets resemble reasoning and factual questions, but do not require actual reasoning or factual retrieval. The reason for constructing such query sets is to compare the results for the factual / reasoning query sets to the corresponding control sets, and comment on whether the patterns we are seeing also show up for queries that do not require reasoning / retrieval, which further helps interpretations of the results.

Two factual control examples can be seen in Table 10 and 11, for the 7B and 35B model respectively. They are designed to match the first factual query superficially (e.g. the same numbers), but they are instead reading comprehension questions about fictional entities. This means all the model needs to do to answer the question is retrieve from context (not parameters).

Two of the reasoning control examples for the 7B and 35B respectively can be seen in Table 12 and 13. Again, these resemble reasoning queries in number and style, but do not require any actual reasoning as the answer is given in the context.

Table 8: Example from the reasoning set that involves calculating the slope of a line going through two points. We have 20 examples like these on different points. The completion is by the 7B model, but we use the same questions for the 35B model.

| **Reasoning query set (slopes)** |
| --- |
| *Prompt* |
| What is the slope of the line passing through the points (93, 28) and (74, 47)? |
| Think step-by-step. |
| *Completion by 7B* |
| To find the slope of the line passing through the points (93, 28) and (74, 47), we can use the following steps: |
| 1. Identify the coordinates of the two points:
- Point A: (93, 28)
- Point B: (74, 47) |
| 2. Calculate the difference in x-coordinates between the two points:
- Difference in x-coordinates: x2 - x1 = 74 - 93 = -19 |
| 3. Calculate the difference in y-coordinates between the two points:
- Difference in y-coordinates: y2 - y1 = 47 - 28 = 19 |
| 4. Determine the ratio of the difference in y-coordinates to the difference in x-coordinates:
- Ratio: 19 / -19 = 1 / -1 |
| 5. Multiply the ratio by -1 to reverse the direction of the line:
- Slope: -1 / 1 = -1 |
| Therefore, the slope of the line passing through the points (93, 28) and (74, 47) is -1. |

Table 9: Example from the reasoning set that requires solving for $x$ in a linear equation. We have twenty other examples like these on different numbers, only for the 35B model. The completion is by the 35B model.

| **Reasoning query set (linear)** |
| --- |
| *Prompt* |
| Solve for x: 5x + 21 = 91 |
| |
| Think step-by-step. |
| *Completion by 35B* |
| Here are the steps to solve for x:
1. Subtract 21 from both sides of the equation:
5x + 21 - 21 = 91 - 21
5x = 70

2. Divide both sides of the equation by 5 to solve for x:
(5x) / 5 = 70 / 5
x = 14

Therefore, the solution to the equation 5x + 21 = 91 is x = 14. |

Table 10: Example from the factual control set that resembles the question about the tallest mountain in the world from the 7B factual query set (Table 2), but does not require any factual recall. The completion is by the 7B model.

| **Factual control query set** |
| --- |
| *Prompt* |
| The planet Zog has multiple mountains, of which the tallest one is called Wirtu which is 29,029 feet tall. What is the largest mountain on the planet Zog called and how tall is it? |
| *Completion by 7B* |
| The largest mountain on the planet Zog is called Wirtu and it is 29,029 feet tall. |

Table 11: Example from the factual control set that resembles the first factual query for the 35B model superficially, but does not require any factual recall. The completion is by the 35B model.

| **Factual control query set** |
| --- |
| *Prompt* |
| In a fictional planetary system, there is a supernova remnant and pulsar wind nebula in the centre of a constellation of stars called Minotaur called the Goat Nebula.
What is the supernova remnant and pulsar wind nebula in the constellation Minotaur called? |
| *Completion by 35B* |
| In the fictional planetary system, the supernova remnant and pulsar wind nebula in the centre of the constellation Minotaur is called the "Goat Nebula". |

Table 12: Example from the reasoning control set that resembles the slope queries superficially, but does not require any reasoning. The completion is by the 7B model.

| Reasoning control query set |
| --- |
| *Prompt* |
| The slope of the line is -22. What is the slope of the line?

Think step-by-step. |
| *Completion by 7B* |
| Given that the slope of the line is -22.

To find the slope of the line, we need to find the change in y-coordinates divided by the change in x-coordinates.

The slope of the line is -22.

Therefore, the slope of the line is -22. |

Table 13: Example from the reasoning control set that resembles the slope queries superficially, but does not require any reasoning. The completion is by the 35B model.

| Reasoning control query set |
| --- |
| *Prompt* |
| This line has a slope 6. What is the slope of the aforementioned line?

Think step-by-step. |
| *Completion by 35B* |
| The slope of a line is a measure of the steepness of the line.
It is defined as the ratio of the change in the y-coordinate to the change in the x-coordinate.

In this case, the line has a slope of 6. This means that for every 1 unit increase in the x-coordinate, the y-coordinate will increase by 6 units.

It's important to note that the slope of a line is not limited to positive values and can be negative or zero depending on the orientation of the line. |

## A.4 QUERY KEYWORDS FOR FINDING THE ANSWER

We provide all keywords used to search for the answers for both the 7B and 35B in the supplementary material to this manuscript, and describe here the procedure for generating keywords, as well as giving some examples.

For the factual questions, we generated keywords manually for each query, and these can be found in the supplemental material. For example, for the question *"What is the world's smallest mammal by body length?"* (answer: bumblebee bat), we have the following keywords: bumblebee bat; bumblebee; bumble; bee; bat; smallest mammal; body length; mammal; smallest; small. This results in many false positives, e.g. if only the word 'small' occurs, which we all check manually for the answer.

Based on the type of reasoning question, we programatically create keywords for each question. For example, for the question in Table 9, the keywords are:

```
['14', 'x = 14', '5x + 21', '91', '5x + 21 = 91', '21', '5',
'91 - 21', '91 - 21 = 70', '(91 - 21) / 5', '70 / 5',
'70 / 5 = 14', '70', 'x=14', '5x+21', '5x+21=91', '91-21',
'91-21=70', '(91-21)/5', '70/5', '70/5=14',
'(91 - 21) divided by 5', '(91-21) divided by 5',
'(91 minus 21) divided by 5', '(91 min 21) divided by 5',
'70 divided by 5', '70 divided by 5 = 14',
'70 divided by 5 is 14', '70 / 5 is 14', '70/5 is 14',
'91 - 21 is 70', '91-21 is 70', '91 minus 21 is 70',
'91 min 21 is 70', '70 divided by 5 equals 14',
'70 / 5 equals 14', '70/5 equals 14', '91 - 21 equals 70',
'91-21 equals 70', '91 minus 21 equals 70', '91 min 21 equals 70',
'5x plus 21', '5x plus 21 = 91', '5x plus 21 is 91', '5x + 21 is 91',
'91 minus 21', '91 min 21', '91 minus 21 = 70', '91 min 21 = 70',
'(91 minus 21) / 5', '(91 min 21) / 5']
```

Note that, because the individual numbers '14', '5', '91', and '70' are part of the keywords, each document that contains one of these numbers becomes a hit, and we go over all hits manually.

## A.5 Prompts given to Command R+ for finding the answer

We use multiple prompts for each different type of reasoning question to allow Command R+ to find the answer in the top 500 influential documents; prompts to find the answer to the intermediate reasoning steps, and a prompt for finding the answer to the full question. We provide an example of each below.

Preamble:

> *You are a brilliant AI assistant that is excellent at arithmetic designed to help users with data analysis. You will be given an arithmetic query and a document, and your task is to determine whether the answer to the question is in the document.*

---

### Prompt for the first step to a two-step arithmetic question

Question: 4 + 2
Answer: 4 + 2 = 6

What also counts as an answer:

- The calculation is written out in words, or part of a story.
- The order of operations are changed. E.g. 2 + 4 = 6.
- Different symbol used for sum/subtract sign. E.g. plus/minus.
- The calculation is part of another larger calculation. E.g. (4 + 2) * 9 = 6 * 9 or (4 + 2)/12 = 6/12.
- Different formatting. E.g. (4) + (2) = (6).
- The calculation is a part of an algebraic formulation. E.g. 4X + 2X = 6X.

What does not count as an answer:
- Other numbers are being summed/subtracted. E.g. 5 + 2.
- Numbers are taken to the other side of the equals sign. E.g. 6 - 2 = 4.

Document:

$<document>$

Is the answer given in the document? Answer with yes or no. If you answer with yes, indicate where the answer is by copying the part of the document in which the answer occurs, ending with an explanation of why that passage contains the answer. Think step-by-step and carefully consider all the different ways in which such an answer might be given.

---

### Prompt for the second step to a two-step arithmetic question

Question: 6 * 15
Answer: 90

What also counts as an answer:

- The calculation is written out in words, or part of a story.
- The order of operations are changed. E.g. 15 * 6 = 90.
- Different symbol used for the multiplier sign. E.g. x or times.
- The calculation is part of another larger calculation. E.g. (6 * 15) * 9 = 90 * 9 or (6 * 15)/12 = 90/12.
- Different formatting. E.g. (6) * (15) = (90).
- The calculation is a part of an algebraic formulation. E.g. 6X * 15X = 90X.

What does not count as an answer:
- Other numbers are being multiplied. E.g. 7 * 15.
- Numbers are taken to the other side of the equals sign. E.g. 6 = 90/15.

Document:

$<document>$

Is the answer given in the document? Answer with yes or no. If you answer with yes, indicate where the answer is by copying the part of the document in which the answer occurs, ending with an explanation of why that passage contains the answer. Think step-by-step and carefully consider all the different ways in which such an answer might be given.

---

### Prompt for step 1 (and 2 is similar) to answer a slope question

Question: 74 - 73
Answer: 74 - 73 = 1

What also counts as an answer:
- The calculation is written out in words, or part of a story.
- The calculation is written in terms of a difference or change. E.g. the difference (or change) between 73 and 74 is 1.
- The order of operations are changed. E.g. 73 - 74 = -1.
- Different symbol used for the minus sign. E.g. subtracted from.
- The calculation is part of another larger calculation. E.g. (74 - 73) * 9 = 1 * 9 or (74 - 73)/12 = 1/12.
- Different formatting. E.g. (74) - (73) = (1).
- The calculation is a part of an algebraic formulation. E.g. 74X - 73X = 1X.

What does not count as an answer:
- Other numbers are being subtracted. E.g. 75 - 73.
- Numbers are taken to the other side of the equals sign. E.g. 74 = 1 + 73.

Document:

$<document>$

Is the answer given in the document? Answer with yes or no. If you answer with yes, indicate where the answer is by copying the part of the document in which the answer occurs, ending with an explanation of why that passage contains the answer. Think step-by-step and carefully consider all the different ways in which such an answer might be given.

### Prompt for step 3 to answer a slope question

Question: 74 / 1
Answer: 74 / 1 = 74

What also counts as an answer:

- The calculation is written out in words, or part of a story.
- The signs on the LHS are flipped. E.g. -74 / -1 = 74.
- Different symbol used for the division sign. E.g. divided by.
- The calculation is part of another larger calculation. E.g. (74 / 1) * 9 = 74 * 9 or (74 / 1)/12 = 74/12.
- Different formatting. E.g. (74) / (1) = (74).
- The calculation is a part of an algebraic formulation. E.g. 74X / 1 = 74X.

What does not count as an answer:
- Other numbers are being divided. E.g. 75 / 1.
- Numbers are taken to the other side of the equals sign. E.g. 74 = 74 * 1.

Document:

$<document>$

Is the answer given in the document? Answer with yes or no. If you answer with yes, indicate where the answer is by copying the part of the document in which the answer occurs, ending with an explanation of why that passage contains the answer. Think step-by-step and carefully consider all the different ways in which such an answer might be given.

## Prompt for step 1 to answer a linear question

Question: 32 - 16
Answer: 16

What also counts as an answer:
- The calculation is written out in words, or part of a story.
- The calculation is written in terms of a difference or change. E.g. the difference (or change) between 32 and 16 is 16.
- The order of operations are changed. E.g. -16 + 32 = 16.
- Different representation used for the minus sign. E.g. 'subtracted from'.
- The calculation is part of another larger calculation. E.g. (32 - 16) * 9 = 16 * 9 or (32 - 16)/12 = 16/12.
- Different formatting. E.g. (32) - (16) = (16).
- The calculation is a part of an algebraic formulation. E.g. 32X - 16X = 16X.

What does not count as an answer:
- Other numbers are being subtracted. E.g. 33 - 16.
- Numbers are taken to the other side of the equals sign. E.g. 32 = 16 + 16.

Document:

*< document >*

Is the answer given in the document? Answer with yes or no. If you answer with yes, indicate where the answer is by copying the part of the document in which the answer occurs, ending with an explanation of why that passage contains the answer. Think step-by-step and carefully consider all the different ways in which such an answer might be given.

## Prompt for step 2 to answer a linear question

Question: 16 / 8
Answer: 16 / 8 = 2

What also counts as an answer:
- The calculation is written out in words, or part of a story.
- The calculation is written in terms of a ratio. E.g. the ratio between 16 and 8 is 2.
- Different representation used for the division sign. E.g. 'divided by'.
- The calculation is part of another larger calculation. E.g. (16 / 8) * 9 = 2 * 9 or (16 / 8)/12 = 2/12.
- Different formatting. E.g. (16) / (8) = (2).
- The calculation is a part of an algebraic formulation. E.g. 32X / 16X = 2X.

What does not count as an answer:
- Other numbers are being divided. E.g. 17 / 8.
- Numbers are taken to the other side of the equals sign. E.g. 16 = 2 * 16.

Document:

*< document >*

Is the answer given in the document? Answer with yes or no. If you answer with yes, indicate where the answer is by copying the part of the document in which the answer occurs, ending with an explanation of why that passage contains the answer. Think step-by-step and carefully consider all the different ways in which such an answer might be given.

## Prompt for the full answer to a linear question

Question: 8x + 16 = 32
Answer: 2

What also counts as an answer:
- The calculation is written out in words, or part of a story.
- The calculation is written in terms of a ratio. E.g. the ratio between 16 and 8 is 2.
- Different representation used for the plus sign or the equals sign. E.g. 'added to' and 'equals'.
- A different variable than X is used. E.g. 't': 8t + 16 = 32'.
- The calculation is part of another larger calculation. E.g. (8x + 16 = 32) * 9 = 2 * 9 or (8x + 16 = 32)/12 = 2/12.
- The solution is written out in steps below each other. E.g.:
8x + 16 = 32
8x = 2
x = 0.

- The calculation is a part of an algebraic formulation. E.g.:
5 * (8x + 16) = 5 * 32
5 * x = 5 * 2.

What does not count as an answer:
- Other numbers are being used. E.g. 9x + 16 = 32.

Document:

<document>

Is the answer given in the document? Answer with yes or no. If you answer with yes, indicate where the answer is by copying the part of the document in which the answer occurs, ending with an explanation of why that passage contains the answer. Think step-by-step and carefully consider all the different ways in which such an answer might be given.

## A.6  PROMPTS GIVEN TO COMMAND R+ FOR CHARACTERISING THE RELATIONSHIP BETWEEN THE QUERY AND THE DOCUMENT

We combine all reasoning queries in pairs with their top 500 most influential documents, and prompt Command R+ to characterise the relationship. For all types of reasoning, we use the same preamble:

> *You are a brilliant AI assistant that is excellent at arithmetic designed to help users with data analysis. You will be given an arithmetic query and a document, and your task is to characterise the document by choosing keywords from a given set that best describe how the document relates to the question.*

For each type of reasoning, we craft a prompt that allows Command R+ to choose multiple keywords for each query-document pair in the top 500 documents. We provide each below.

---

**Prompt for arithmetic questions**

Start of Query:

*<query>*

End of Query

Start of Document

*<document>*

End of Document

How is the document related to the query?

Choose from the following keywords:

Similar arithmetic operations on similar numbers (e.g. the numbers are similar in magnitude or the numbers are the same)
Similar arithmetic operations (on other types of numbers, e.g. much larger or smaller)
Reasoning traces (multiple reasoning steps are explicitly given in the document explaining how one gets to an answer)
Other types of maths
Code that contains arithmetic
Code that concerns other types of math
Code that concerns no math/arithmetic
Text about math/arithmetic (no other relation to the query than that the text is about math, text does not perform math/arithmetic)
Superficial similarities (there is no real relation, but loosely related topics occur, like the text contains words related to other parts of math, like algebra)
Similar formatting (question/answer pair about other topics than math)
Similar formatting (other)
Other (pick own keyword)

Explain your answer for each keyword by quoting from the query and document and describing why they are similar. Keep in mind that the document might be in another language than English. If you pick any of the code keywords, add the programming languages in brackets (e.g. 'Code that contains arithmetic (Python, LaTeX)'). If the relation between the query and the document is not described by any of the given keywords, choose 'other' and pick your own keyword that describes the document. Otherwise, if the query is not related to the document, state 'no relation' and describe why. Give your answer in the form of a semicolon-separated list of keywords, and add an explanation below separated by newlines Give your answer in the form of a semicolon-separated list of keywords, and add an explanation below separated by newlines (e.g. 'keyword 1; keyword 2; keyword 3 (Python) [explanation]').

---

## Prompt for slope questions

Start of Query:

*<query>*

End of Query

Start of Document

*<document>*

End of Document

How is the document related to the query?

Choose from the following keywords:

Similar arithmetic operations on similar numbers (e.g. the numbers are similar in magnitude or the numbers are the same)
Similar arithmetic operations (on other types of numbers, e.g. much larger or smaller)
Reasoning traces (multiple reasoning steps are explicitly given in the document explaining how one gets to an answer)
Other types of maths
Code that contains arithmetic
Code that calculates the slope between two numbers
Math that calculates the slope between two numbers
Code that calculates the slope of an equation
Math that calculates the slope of an equation
Code that concerns other types of math
Code that concerns no math/arithmetic
Text about math/arithmetic (no other relation to the query than that the text is about math, text does not perform math/arithmetic)
Superficial similarities (there is no real relation, but loosely related topics occur, like the text contains words related to other parts of math, like algebra)
Similar formatting (question/answer pair about other topics than math)
Similar formatting (other)
Other (pick own keyword)

Explain your answer for each keyword by quoting from the query and document and describing why they are similar. Keep in mind that the document might be in another language than English. If you pick any of the code keywords, add the programming languages in brackets (e.g. 'Code that contains arithmetic (Python, LaTeX)'). If the relation between the query and the document is not described by any of the given keywords, choose 'other' and pick your own keyword that describes the document. Otherwise, if the query is not related to the document, state 'no relation' and describe why. Give your answer in the form of a semicolon-separated list of keywords, and add an explanation below separated by newlines (e.g. 'keyword 1; keyword 2; keyword 3 (Python) [explanation]').

## Prompt for linear questions

Start of Query:

*<query>*

End of Query

Start of Document

*<document>*

End of Document

How is the document related to the query?

Choose from the following keywords:

Code that solves a linear equation for a variable (of the form ax + b = c or ax - b = c)
Code that solves a linear equation with multiple variables for one or both variables (e.g. ax + by = c)
Code that solves a linear equation of another form than ax + b = c or ax - b = c
Math that solves a linear equation for a variable (of the form ax + b = c or ax - b = c)
Math that solves an equation with multiple variables for one or both variables (e.g. ax + by = c)
Math that contains linear equations of another form than ax + b = c or ax - b = c
Math that contains linear equations but they are not solved (of the form ax + b = c or ax - b = c)
Math that contains linear equations but they are not solved (of another form than ax + b = c or ax - b = c)
Similar algebraic operations on similar numbers (e.g. the numbers are similar in magnitude or the numbers are the same)
Similar algebraic operations (on other types of numbers, e.g. much larger or smaller)
Other forms of algebra
Arithmetic operations
Other types of maths
Code that contains arithmetic
Code that concerns other types of math
Code that concerns no math/algebra
Text about math/algebra (no other relation to the query than that the text is about math, text does not perform math/algebra)
Reasoning traces (multiple reasoning steps are explicitly given in the document explaining how one gets to an answer)
Superficial similarities (there is no real relation, but loosely related topics occur, like the text contains words related to other parts of math, like arithmetic)
Similar formatting (question/answer pair about other topics than math)
Similar formatting (other)
Other (pick own keyword)

Explain your answer for each keyword by quoting from the query and document and describing why they are similar. Keep in mind that the document might be in another language than English. If you pick any of the code keywords, add the programming languages in brackets (e.g. 'Code that contains arithmetic (Python, LaTeX)') If the relation between the query and the document is not described by any of the given keywords, choose 'other' and pick your own keyword that describes the document. Otherwise, if the query is not related to the document, state 'no relation' and describe why. Give your answer in the form of a semicolon-separated list of keywords, and add an explanation below separated by newlines (e.g. 'keyword 1; keyword 2; keyword 3 (Python) [explanation]'). If you pick a keyword about solving a linear equation, add the linear equation in the explanation.

### A.7 FURTHER DISCUSSION OF LIMITATIONS

More broadly, our work suffers from the same limitations any work does that uses EK-FAC influence functions; we do many approximations to estimate the counterfactual and only take into account MLP parameters. This latter decision is because EK-FAC influence functions are not properly defined for the attention layers (Grosse et al., 2023), although we do look at the dense layers used within them. We list the assumptions and approximations here:

- First-order Taylor approximation to the PBRF.
- Assume different layers of MLPs are independent, making the Gauss-Newton Hessian block-diagonal.
- Assume activations are independent of pre-activation pseudo-gradients.
- Estimate the approximation to the Fisher Information Matrix or equivalently the Gauss-Newton Hessian by sampling from the empirical data distribution / model output distribution, because it's an expectation over that distribution (MC estimation).
- Block-diagonal approximation of the eigenvector matrices within each layer.
- Low-rank approximation of query gradients.
- Assume EK-FAC for SFT stage is identity (Bae et al., 2024).

All these approximations are verified and justified in Grosse et al. (2023) and (Bae et al., 2024), and the reader is referred there for a more in-depth analysis.

Our empirical results showing that nonetheless influence functions surface documents that are causally related to accuracy in Appendix A.1 should alleviate some of these concerns, but not all.

### A.8 ADDITIONAL RESULTS FOR THE QUALITATIVE ANALYSIS

#### A.8.1 DETAILS ON ANSWERS TO QUESTIONS IN PRETRAINING DATA

In the main text, we find the answer to factual questions relatively often compared to the answer to reasoning questions. In this section, we comment on the possibility that the answer to reasoning questions are simply not part of the pretraining sample of 5 million documents we look at, as well as present examples of documents with answers to queries. Recall that all reasoning tasks require multiple steps, and the model outputs reasoning traces to get to the final answer. This means that if the model is retrieving the answers, it should retrieve answers to all the reasoning steps. On top of the search in the main paper in Section 5.2, we search for answers to the reasoning steps and factual questions in a random subset of the 5M pretraining documents. For the 7B reasoning questions, we find 43 documents containing answers to reasoning steps, of which only 9 show up in the top 0.02% of the data. Of these 9, 4 documents together contain the 2 answers found for the 7B arithmetic queries in the main text. The remaining 5 are answers to single reasoning steps that do not combine to a full answer. By contrast, we find the full answer to factual questions in 73 documents, of which 35 show up in the top 0.02% of the data. For the 35B, we find 7 documents with answers to reasoning steps, of which 4 show up in the top 0.02% (none combining to a full answer). For the factual questions, we find 17 documents with answers, of which 15 show up in the top 0.02%. In terms of full answers showing up in the top 0.02%, we find one additional full answer on top of the ones we found in the main text for the 7B reasoning questions, spread over two documents with rank 896542 and 4997351 of 5 million respectively (i.e. highly un- or negatively influential). For the 35B we do not find full answers to reasoning queries at all. We provide many documents with answers to factual and reasoning queries found in the top 0.02% in the supplemental material as well as one example per reasoning step answer we find (e.g. if we find the answer to $6 - 4 = 2$ four times, we show one example in the supplement). We highlight here some examples from the larger pretraining sample for illustration.

**Examples of pretraining data with answers.**
For factual questions, it happens relatively frequently that the answer to the question shows up as highly influential in multiple documents of the top 10 documents. For example, for the factual question in Table 2 (*"What is the tallest mountain in the world and how tall is it?"*), the answer shows up at ranks 1, 4, 6, and 7. The document at rank 1 (the most positively influential document), is the following, which has the same question and the answer (question 5 below, underlined):

**Factual query**

What is the tallest mountain in the world and how tall is it?

**Completion by 7B**

The tallest mountain in the world is Mount Everest, which is located between Nepal and Tibet. The height of Mount Everest is 8,849 meters (29,029 feet) above sea level.

**Rank 1 document**

Write some general trivia questions and answers. Here are some general trivia questions and answers:

1. What is the longest word in the English language?

Answer: The longest word in the English language is "pneumonoultramicroscopic-silicovolcanoconiosis", which is a type of lung disease.

2. How many bones are in the human body?

Answer: There are 206 bones in the human body.

3. What is the largest ocean in the world?

Answer: The largest ocean in the world is the Pacific Ocean, which covers approximately one-third of the Earth's surface.

4. What is the longest river in the world?

Answer: The longest river in the world is the Nile River, which is 6,650 kilometers (4,130 miles) long.

5. What is the highest mountain in the world?

Answer: The highest mountain in the world is Mount Everest, which is 8,848 meters (29,029 feet) high.

6. What is the most common element in the human body?

Answer: The most common element in the human body is oxygen, which makes up approximately 65% of the human body's mass.

*... redacted*

Another document has the answer to part of an arithmetic query for the 7B ("Calculate the answer: (5 - 3) * 12. Think step-by-step."), namely 5 - 3 = 2 (underlined below, note that one needs to understand rules to writing arithmetic to figure the answer out):

---

**Reasoning query (arithmetic)**

Calculate the answer: (5 - 3) * 12

Think step-by-step.

---

**Completion by 7B**

First, let's evaluate the expression inside the parentheses:
5 - 3 = 2

Now, let's multiply the result by 12:
2 * 12 = 24

Therefore, the answer is 24.

---

**Rank 2 document**

shown in numbers 7, 9 and 10.
PEMDAS Rules Practice 1 Solutions
1.
Solution:
Multiply: 2 x 3 + 4 x 6 = 6 + 24
Add: 6 + 24 = 30
Answer: 30
2.
Solution:
Multiply:
Divide:
Add:
Answer: 0
3.
Solution:
Parenthesis: $4 + (6 - 2)2 + 1 = 4 + 42 + 1$
Exponent: $4 + 42 + 1 = 4 + 16 + 1$
Add: 4 + 16 + 1 = 21
Answer: 21
4.
Solution:
Parenthesis: $8(6 - 2) \div 2\underline{(5 - 3)} = 8(4) \div 2\underline{(2)}$
Multiply: $8(4) \div 2(2) = 32 \div 2(2)$*
Divide: $32 \div 2(2) = 16(2)$
Multiply: 16(2) = 32
Answer: 32
*This is the case mentioned in the PEMDAS Rules that when multiplication and division are performed consecutively (without any other operations or grouping symbols in between), the perform the operations from the left hand side to the right hand side.
... *redacted*

---

Interestingly, this document shows up in the top 10 documents for 11 of 20 arithmetic queries. By contrast, the factual answer document shown before shows up in the top 10 for 4 of 40 queries (we

have another query that asks for the largest ocean in the world, for which this document also has the answer).

To show that answers to more "niche" questions also show up, consider this document that contains the answer to the question *"What is the common name for the larva of a housefly?"* (answer: maggot, underlined below):

---

**Factual query**

What is the common name for the larva of a housefly?

**Completion by 35B**

The common name for the larva of a housefly is a maggot.

**Rank 6 document**

Common House Fly
Flies are easily recognized and are universally annoying. To say they are unappealing is an understatement. Flies enjoy spending time on decaying organic filth, including rotting meat and feces. They then manage to enter your home and land on your kitchen counters and any uncovered food. House flies do more than just buzz around and steal your food. They can transfer over 65 diseases including cholera, dysentery, poliomyelitis, and tuberculosis. Flies accumulate nasty pathogens when they rest on sewers, trash, feces, and any other moist or decaying material.
Flies lay their eggs on food (fruit in your fruit bowl), other animals, and even rotting flesh. When the eggs hatch, the tiny rice-like larvae are called maggots. Few other bugs provoke nausea and an overall disgust like tiny, squirming maggots. There are a number of ways to eliminate flies and their larvae from your home. The most common practices include using sticky fly traps, ultra-violet lights, and baited traps. Fly swatters are effective when trying to get rid of the lone fly. A timely smack from a dish towel can also be employed.
There are simple daily practices that will help keep flies from taking over your house. Store food in sealed containers, wipe crumbs off the counters, and don't leave food out. Grass clippings, garbage, and decaying matter will also attract flies, so get rid of these items immediately. Keep your trash cans tightly covered and clean them on a man-made surface where the run-off can't seep into the soil. Whenever possible, use door and window screens. This will help keep flies from entering your home. However, flies can find holes and gaps so remember to check your screens occasionally and ensure they are impenetrable. Flies can also find openings on pipes or conduits that lead inside. As a means of prevention, caulk those areas that channel into your home.
At ZipZap Termite & Pest Control, we know how pesky the persistent buzzing of the house fly can be. With kids running in and out, it is near impossible to keep flies from getting in. Backyard parties are hard to enjoy when you've got house flies buzzing around and landing on your food. If you feel you have an overwhelming number of flies

---

This document has rank 6 for the relevant query, and never shows up in the top 10 for other queries.

Below, we show a document containing the answer to the reasoning step 5 + 4 = 9, required for one of the arithmetic queries for the 7B model ("Calculate the answer: (5 + 4) * 2. Think step-by-step."), which does not show up in the top 0.02%.

Reasoning query (arithmetic)

Calculate the answer: (5 + 4) * 2

Think step-by-step.

Completion by 7B

First, let's evaluate the expression inside the parentheses:
5 + 4 = 9

Now, let's move on to the next step:
9 * 2 = 18

Therefore, the answer to the given expression is 18.

Rank 2140 document

For example:
n=1 =  1+2=3
n=2 =  1+4=5, 2x3=6
n=3 =  4+5=9, 1+7=8, 2x3=6

The question is, does a solution exist for every n?
I tried writing a basic program and it becomes too slow after n = 14. Here are the solutions I have so far:
1 ['1+2=3']
2 ['2*3=6', '1+4=5']
3 ['4+5=9', '1+7=8', '2*3=6']
4 ['3+6=9', '1+10=11', '4+8=12', '2+5=7']
5 ['2+8=10', '3+6=9', '1+13=14', '5+7=12', '11+4=15']
6 ['3*5=15', '2+8=10', '4+14=18', '6+11=17', '7+9=16', '1+12=13']
7 ['6+12=18', '3*5=15', '7+10=17', '1+20=21', '4+9=13', '2+14=16', '8+11=19']
8  ['8+14=22',  '6+12=18',  '7+10=17',  '2+19=21',  '1+15=16',  '11+13=24', '4+5=9', '3+20=23']
9  ['6+19=25',  '8+14=22',  '4+13=17',  '2+18=20',  '1+26=27',  '3+7=10', '9+15=24', '5+16=21', '11+12=23']
10 ['6+19=25', '

This document has rank 2140 for the relevant query.

### A.8.2 CROSS-LINGUAL TRANSFER

***Additional finding: The answer to the factual question sometimes shows up in non-English languages***.
Interestingly, we observe some crosslingual transfer for the factual questions. For example, for the question about the tallest mountain in the world (Table 2), the answer shows up in Portuguese:

> *A americana Samantha Larson, de 19 anos, se tornou nesta sexta-feira a mulher estrangeira mais jovem a conquistar o Monte Everest, segundo nota oficial divulgada pelo Ministério de Turismo do Nepal. A montanha, de 8.848m, é a mais alta do mundo e se encontra na fronteira entre o Nepal e Tibet.*

Which translates to:

> *American Samantha Larson, 19, became the youngest foreign woman to conquer Mount Everest on Friday, according to an official statement released by Nepal's Ministry of Tourism. The 8,848m mountain is the highest in the world and is located on the border between Nepal and Tibet.*

We observe more crosslingual transfer for questions, for example for the question *"What is the capital of Belgium?"* the answer shows in up in French and Spanish. We show the French document here:

> *Le Premier ministre belge Yves Leterme a assuré ce mercredi qu'il resterait en place et mènerait à bien la réforme institutionnelle entre les régions, malgré les profondes divisions entre Flamands et Wallons qui menacent l'unité du pays.*
> *...*
> *Les francophones redoutent pour leur part une réduction des budgets accordés à la Wallonie, région la plus pauvre du pays, et à la capitale bilingue, Bruxelles. Ils estiment également que les régions se sont vu transférer depuis les années 1980 assez de compétences fédérales, et soupçonnent les néerlandophones de chercher à faire sécession de la Belgique afin de pouvoir déclarer l'indépendance de la Flandre.*

Which translates to:

> *Belgian Prime Minister Yves Leterme assured on Wednesday that he would stay in office and carry out the institutional reform between the regions, despite the deep divisions between Flemish and Walloons that threaten the unity of the country.*
> *...*
> *The French speakers, for their part, fear a reduction in the budgets granted to Wallonia, the poorest region of the country, and to the bilingual capital, Brussels. They also believe that the regions have been transferred enough federal powers since the 1980s, and suspect that the Dutch-speaking countries are seeking to secede from Belgium in order to be able to declare the independence of Flanders.*

Note that both these quotes are snippets from otherwise larger documents. We did not translate all documents and hence only found cases of crosslingual transfer if there happened to be keyword overlap. We show a few here, but have found the answer to factual questions through keyword overlap with non-English documents 8 times for the 7B model and 4 times for the 35B model. Note that because this is only based on circumstantial keyword overlap, we likely missed most cases of cross-lingual transfer, and therefore cannot assign any meaning to the fact that it happened less for the 35B than the 7B. It would be interesting to focus on cross-lingual transfer in future work.

### A.8.3 CHARACTERISE RELATION TOP DOCUMENTS TO QUERY

**Finding 4: why documents are influential for reasoning**. We prompt Command R+ to characterise the relationship between the top 500 documents and each query (see prompts in Appendix A.6). We add 'reasoning traces' as a potential keyword in the prompt, but after inspecting the results we find the model uses that keyword for almost any document, and we remove those results. We report the raw counts of each keyword occurring in the tables below.

| Arithmetic (7B) | Count |
|---|---|
| Other types of maths | 5765 |
| Similar arithmetic operations on other numbers (e.g. much larger/smaller) | 4691 |
| Code that contains arithmetic | 4038 |
| Text about math/arithmetic | 3202 |
| Code that concerns other types of math | 2554 |
| Similar arithmetic operations on similar numbers | 2246 |
| Similar formatting | 2223 |
| Superficial similarities | 1391 |
| Code that concerns no math/arithmetic | 277 |

Table 14: Raw counts of the amount of times Command R+ assigns a certain keyword to a query-document pair to characterise its relation, for the arithmetic (7B) queries.

| Slopes (7B) | Count |
|---|---|
| Other types of maths | 10787 |
| Similar arithmetic operations on similar numbers | 7312 |
| Code that contains arithmetic | 5035 |
| Similar formatting | 4675 |
| Text that explains in words how to calculate the slope of an equation | 3911 |
| Code that concerns other types of math | 3577 |
| Text about math/arithmetic | 3323 |
| Text that explains in words how to calculate the slope between two numbers | 2959 |
| Math that calculates the slope of an equation | 2921 |
| Math that calculates the slope between two numbers | 2490 |
| Superficial similarities | 2222 |
| Text that mentions the slope but does not explain how to calculate it | 1677 |
| Code that calculates the slope between two numbers | 1633 |
| Code that calculates the slope of an equation | 1110 |
| Code that concerns no math/arithmetic | 263 |
| Other | 15 |

Table 15: Raw counts of the amount of times Command R+ assigns a certain keyword to a query-document pair to characterise its relation, for the slopes (7B) queries.

| Slopes (35B) | Count |
|---|---|
| Other types of maths | 11104 |
| Similar arithmetic operations on similar numbers | 8340 |
| Code that contains arithmetic | 4617 |
| Similar formatting | 4141 |
| Text that explains in words how to calculate the slope of an equation | 3869 |
| Text about math/arithmetic | 3845 |
| Math that calculates the slope of an equation | 3745 |
| Math that calculates the slope between two numbers | 3533 |
| Code that concerns other types of math | 3192 |
| Text that explains in words how to calculate the slope between two numbers | 2747 |
| Superficial similarities | 2291 |
| Text that mentions the slope but does not explain how to calculate it | 1936 |
| Code that calculates the slope between two numbers | 1150 |
| Code that calculates the slope of an equation | 865 |
| Code that concerns no math/arithmetic | 121 |
| Other | 12 |
| Similar arithmetic operations on other numbers (e.g. much larger/smaller) | 1 |

Table 16: Raw counts of the amount of times Command R+ assigns a certain keyword to a query-document pair to characterise its relation, for the slopes (35B) queries.

| Linear (35B) | Count |
|---|---|
| Math that contains linear equations but they are not solved | 13434 |
| Similar algebraic operations on similar numbers | 10717 |
| Similar formatting | 5533 |
| Math that solves a linear equation for a variable | 2415 |
| Other forms of algebra | 2234 |
| Arithmetic operations | 2057 |
| Code that contains arithmetic | 1417 |
| Other types of maths | 1390 |
| Text about math/algebra | 1146 |
| Code that solves a linear equation of another form than $ax + b = c$ or $ax - b = c$ | 1109 |
| Superficial similarities | 1105 |
| Code that concerns other types of math | 949 |
| Code that concerns no math/algebra | 560 |
| Code that solves a linear equation for a variable | 475 |
| Math that solves an equation with multiple variables for one or both variables | 172 |
| Math that contains linear equations of another form than $ax + b = c$ or $ax - b = c$ | 156 |
| Code that solves a linear equation with multiple variables for one or both variables | 110 |
| Other | 1 |

Table 17: Raw counts of the amount of times Command R+ assigns a certain keyword to a query-document pair to characterise its relation, for the linear (35B) queries.

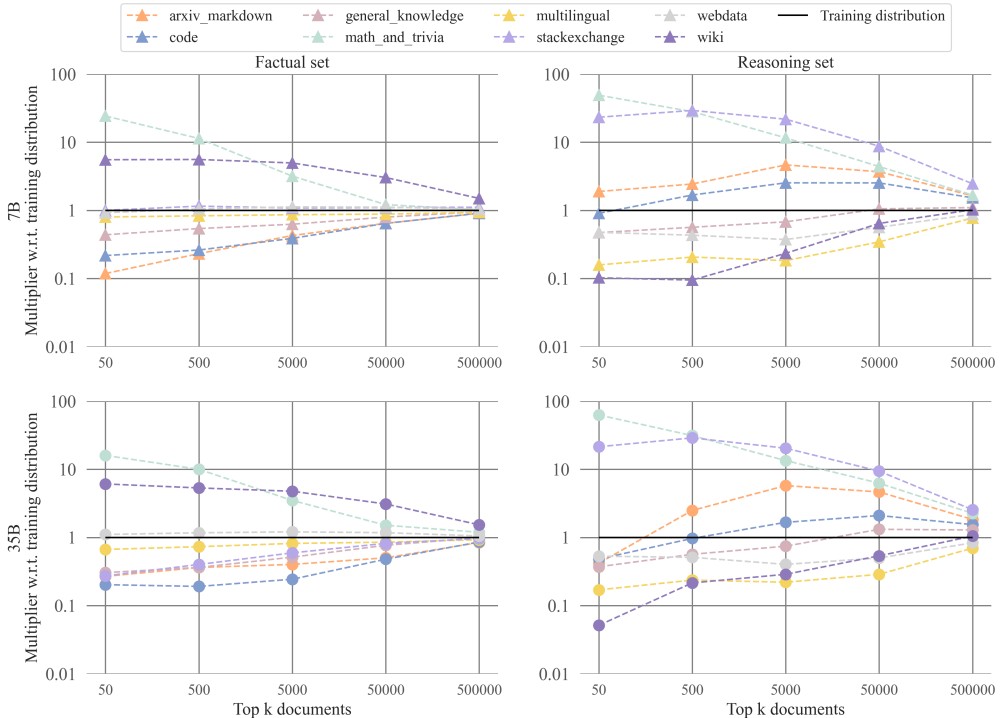

Figure 7: For the *reasoning and factual sets*, we compare the amount of documents from a certain source dataset that show up in the *top* portions of the rankings to the amount you would expect to show up if you randomly sample from the pretraining distribution (indicated by 'Training distribution' in the figure). The top two plots are for the 7B, and the bottom for the 35B. We find that data from Wikipedia and Math & Trivia are important for the factual questions for both models, for the reasoning questions Math & Trivia, StackExchange, Code, and ArXiv data is important. In all cases, the multipliers tend to the training distribution for higher $k$.

### A.8.4 SOURCE DATASET ANALYSIS

**Finding 5: code is heavily overrepresened for reasoning both for the top and bottom portions of the ranking.**

For each source dataset, we report the multiplier w.r.t. the training distribution. This means that if the top $k$ documents are randomly sampled from pretraining, the multipliers will be one, whereas if they are above or below one, that source dataset is either over- or underrepresented in the most influential documents. The full results are presented in Figure 7, and we discuss the most interesting deviations from the pretraining distribution here. For the factual questions, the most overrepresented source datasets for both the 7B and 35B are *Math & Trivia* (multiplier of 27 and 16 for $k = 50$ respectively) and *Wikipedia* (multipliers of 5 and 6 respectively). For the reasoning questions, the most overrepresented datasets are *StackExchange* and *Math & Trivia* (with 50 and 24 als multipliers for the 7B, and 62 and 21 for the 35B). Interestingly, for both the 7B and the 35B, code data is important for the influential documents. Besides *StackExchange*, for the medium-influential portion of the rankings (between $k = 5000$ and $k = 50000$), more code data becomes influential (with multipliers around 2, compared to 0.5 for the factual questions at that same part of the ranking). This is conventional wisdom among practitioners (most LLMs designers use some percentage of code data in pretraining now, e.g. Touvron et al. (2023)), and recent work has empirically found code to be important for reasoning performance (Aryabumi et al., 2024). However, the question of why code data is important for reasoning is still open. Below, in Appendix A.8.5, we further confirm that code is important for reasoning by not only relying on the fact that these documents come from a code dataset, but actually classifying their contents. In Figure 8 we present the same plot for the bottom portion of the ranking, showing the findings are similar. Further, in Figure 9 and 10 we respectively show the same results for the top and bottom portion of the rankings for the

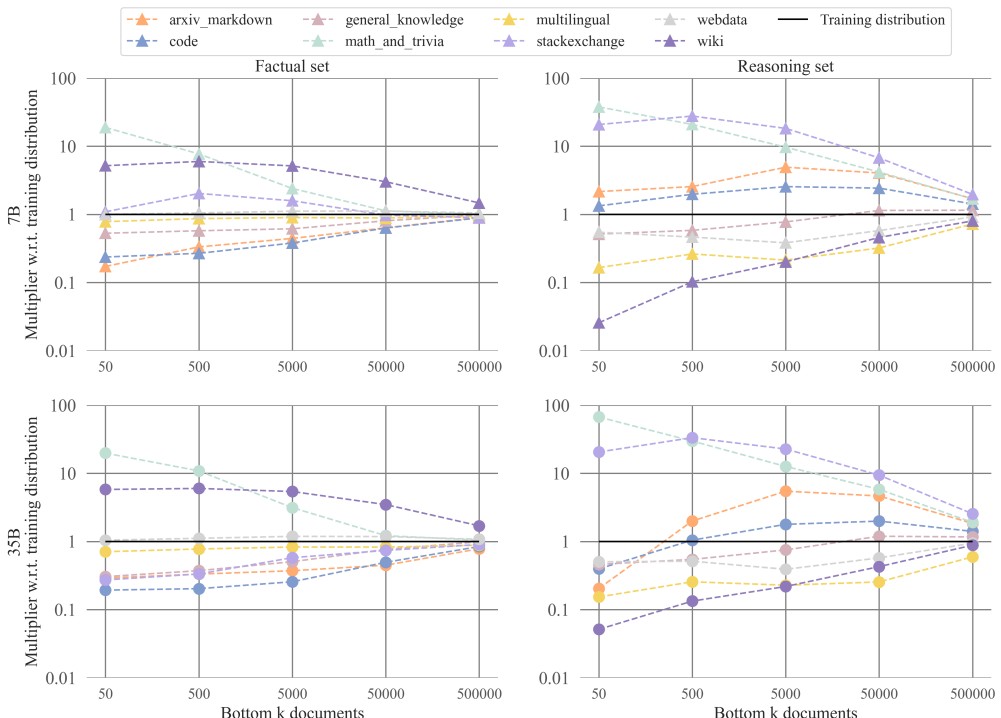

Figure 8: For the *reasoning and factual sets*, We compare the amount of documents from a certain source dataset that show up in the *bottom* portions of the rankings to the amount you would expect to show up if you randomly sample from the pretraining distribution (indicated by 'Training distribution' in the figure). The top two plots are for the 7B, and the bottom for the 35B. We find the patterns are almost identical to those shown for the top portions of the ranking: data from Wikipedia and Math & Trivia are important for the factual questions for both models, for the reasoning questions Math & Trivia, StackExchange, Code, and ArXiv data is important. In all cases, the multipliers tend to the training distribution for higher $k$.

control queries. Again, the results look similar (code and StackExchange is also overrepresented for the reasoning control queries), but arXiv is less overrepresented for reasoning control and wiki is less overrepresented for factual control answering.

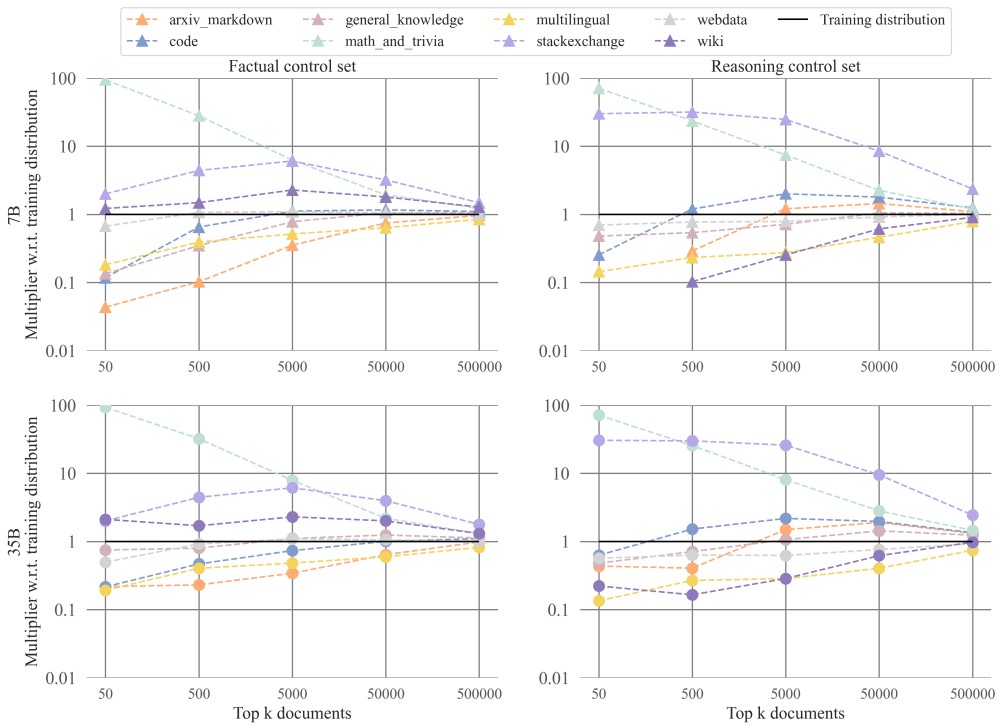

Figure 9: For the query *control sets*, we also compare the amount of documents from a certain source dataset that show up in the *top* portions of the rankings to the amount you would expect to show up if you randomly sample from the pretraining distribution (indicated by 'Training distribution' in the figure). The top two plots are for the 7B, and the bottom for the 35B. We find that code is still overrepresented, but arXiv as source is less overrepresented for the top portions of the reasoning control set than for the reasoning set.

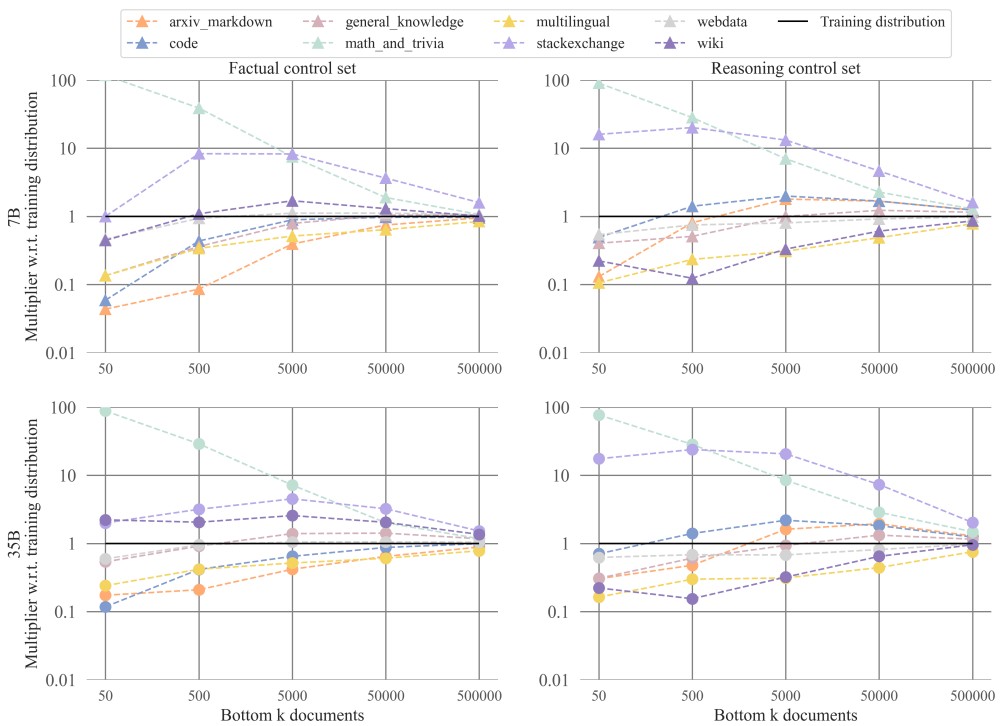

Figure 10: For the query *control* sets, we also compare the amount of documents from a certain source dataset that show up in the *bottom* portions of the rankings to the amount you would expect to show up if you randomly sample from the pretraining distribution (indicated by 'Training distribution' in the figure). The top two plots are for the 7B, and the bottom for the 35B. We find that it again looks similar to the source distribution for the top of the rankings for the query control sets.

A.8.5 CONTENT ANALYSIS OF RELEVANT DOCUMENTS

We provide further insights into the characteristics of influential documents on reasoning queries. To do so, we compute capability categories of the $n = 500$ most frequently occurring documents among the $k = 5000$ most (top) or least (bottom) influential documents for the reasoning queries (for the 7B model), and compare these to a randomly sampled set of 500 documents (we repeat the sampling process three times and provide mean and standard deviation scores on the detected capabilities). Results are shown in Figure 11. We can see that the "code" category represents the vast majority of most and least influential documents, whereas for the random subsets the fraction of code-related documents is relatively small. This provides further evidence that code-related documents strongly influence model performance on reasoning tasks.

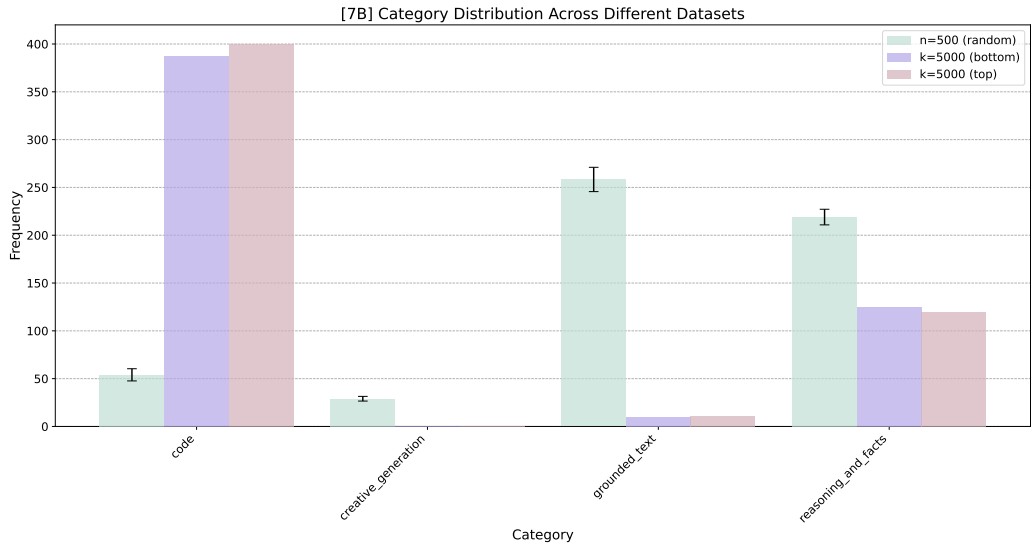

Figure 11: Comparison of capability categories identified for the most and least influential documents for the reasoning queries, as well as for a random subset of sampled documents. We repeat the random sampling three times and report mean scores with standard deviations indicated.

## A.9  ADDITIONAL RESULTS FOR THE QUANTITATIVE ANALYSIS

### A.9.1  CORRELATION ANALYSIS

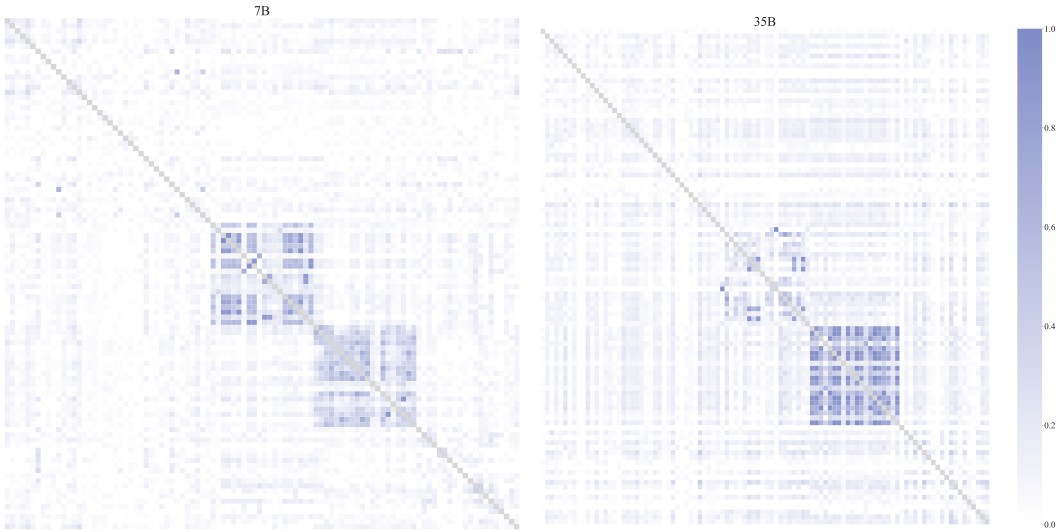

Figure 12: The correlation between the influence scores of all 5 million documents for pairs of queries. All queries are on the x- and y-axis, with the first 40 belonging to the factual set, the next 40 to the reasoning set (arithmetic and slopes for the 7B, and linear and slopes for the 35B), the following 10 to the factual control set, and the last 10 to the reasoning control set. The take-away is that there is only a signficant correlation between queries of the same reasoning type, most strongly so for the 35B slopes queries.

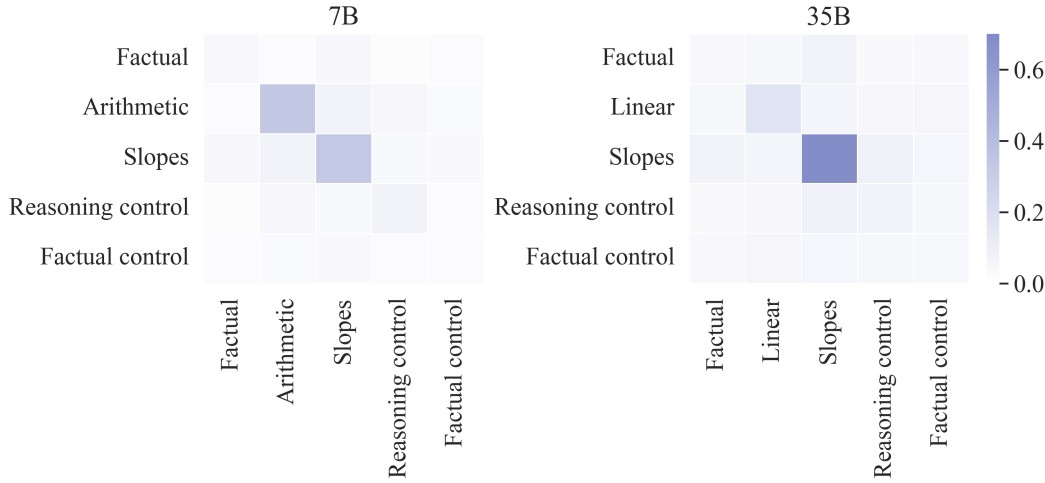

Figure 13: The average correlations between the influences of all documents for queries of a specific type grouped. We leave out any query combinations where the correlation is not significant and any combination where the query on the x- and y-axis is the same query. We again observe that there is only a correlation of influence for queries of the same reasoning type.

**Additional results finding 1 (correlation between reasoning queries of the same type).**
In the main text, we find that there is a correlation between the influence scores for the documents for different queries that underlie the same type of reasoning question (e.g. questions that all require calculating the slope but for different numbers). One other explanation for this result could be the

fact that all these queries are superficially more similar to each other than the factual questions, and that this is the reason the influence correlates. To test this hypothesis, we use the 10 control queries for both the factual questions and the reasoning questions that are superficially similar, but do not require factual retrieval or reasoning to get to the solution (see Appendix A.3 for examples from each set). Figure 12 shows all query-query correlations for the 100 queries we look at, ordered as follows: 40 factual queries, 40 reasoning (of which 20 are arithmetic and 20 slopes questions for the 7B, and 20 linear equation and 20 slopes questions for the 35B), 10 factual control, and 10 reasoning control queries. We see that there is only a significant correlation between queries of the same reasoning type, which we summarise in Figure 13 by grouping queries of the same type and averaging the correlations.

To get a sense of what the control questions look like, we show a side-by-side comparison of a slopes query (top) and the control query that is meant to resemble it (bottom) below. The influences

Table 18: Two query examples, of which one is from the reasoning set (slopes) and the other from the reasoning control set that resembles the slope queries superficially, but does not require any reasoning. The completions are by the 7B model.

| **Reasoning and Reasoning Control Query Set (7B)** |
| --- |
| *Slopes (original prompt)* |
| What is the slope of the line passing through the points (79, 47) and (80, 25)? 

 Think step-by-step. |
| *Completion* |
| To find the slope of the line passing through the points (79, 47) and (80, 25), we can use the following steps: 

 1. Identify the coordinates of the two points: 
 - Point A: (79, 47) 
 - Point B: (80, 25) 

 2. Calculate the difference in x-coordinates between the two points: 
 - Point A x-coordinate: 79 
 - Point B x-coordinate: 80 ... *redacted completion*. |
| *Slopes (control prompt)* |
| The slope of the line is -22. What is the slope of the line? 

 Think step-by-step. |
| *Completion* |
| Given that the slope of the line is -22. 

 To find the slope of the line, we need to find the change in y-coordinates divided by the change in x-coordinates. 

 The slope of the line is -22. 

 Therefore, the slope of the line is -22. |

for this query correlate with the influences for the slope questions on average with a Pearson's R of 0.05, which is much smaller than the 0.32 average correlation between the influences found for the different slope reasoning questions by the 7B model. Below, we perform a more detailed qualitative analysis of the query combinations and what drives their correlations, but first we discuss the quantitative result.

As mentioned, we have 10 factual and 10 reasoning control questions for both models, and show the full correlation matrices below in Figure 12 (per query) and Figure 13 (averaged per group). We observe that the correlations between queries from the control sets and other query sets for the 35B is always between 0.05 and 0.10, which indicates that there can be a score correlation of at least 0.10 for other things than genuine reasoning (e.g. formatting, or topic). Further, the within-group correlations of the reasoning control set sometimes go as high as 0.38 (although the average

is 0.06 for the 7B and 0.10 for the 35B). For comparison, the average linear-linear score correlation for the 35B is 0.16, and not many of the correlations that make up this average are higher than the correlations in the reasoning control sets. To get a sense of how different the correlations are in magnitude between the reasoning questions and the control questions, we calculate the highest correlation of a query from a specific reasoning type with any other query that does not concern reasoning, and count the amount of reasoning query-query combinations for which the correlation is higher. For example, the maximum correlation we find between any slope question for the 35B and any other query that is not a slope question is 0.30 Pearson's R. If we discard all slope query combinations that are below 0.30 we are left with 138 of 190 significant combinations that are higher, ranging up to 0.96 Pearson's R (note that each reasoning group has 20 queries, and all combinations are $20 * 19/2 = 190$). For the linear equation queries by contrast, there are only 34 of 190 query-query combinations within this group that have a correlation higher than the highest correlation with the control queries, ranging up to 0.95 Pearson's R. For the 7B, 84 of 190 arithmetic query combinations have a higher correlation than the control correlations, ranging up to 0.96 Pearson's R, and 120 of 190 slopes query combinations, ranging up to 0.88. We therefore conclude that the correlations between the queries for the linear equations can mainly be explained by other, more superficial things than procedural knowledge, and connect this finding to the fact that the model is less robustly able to solve linear equations. The within-group correlations of the factual set are much lower, and for the 7B we only find 5 of 780 correlations that are higher than the maximum correlation of a factual query with another query group, ranging to 0.63 Pearson's R (we show the queries with the highest correlation below). For the 35B, we find no correlations for factual queries higher than the maximum correlation with another group.

We release all $100 * 100$ query-query correlations in the supplemental material (on which Figure 12 is based), and we highlight a few examples here to get a sense of what is driving higher or lower correlations. We mostly do not find a correlation between the influence scores for different factual questions (on average the correlation is 0.06 for the 7B and 0.03 for the 35B), but we show the highest correlation found between two factual queries below in Table 19. For this particular example, the correlation seems driven by the fact that they are both "what"-questions with very short completions using quotation marks. By contrast, a similar style query-completion pair for the 35B shows almost no correlation (Table 20). For the factual control set (the average correlation is 0.04 for the 7B and 0.08 for the 35B, but we also show a factual control correlation below for completeness. Table 11 shows a factual query and a matched factual control example for the 35B model. The correlation between this control query and the factual query it matches is -0.03.

Table 19: Two query examples for which the Pearson's R correlation of their document influences is 0.63, both from the factual set of the 7B model. We observe that the main similarity between these queries is that they both concern short "what" questions, and the completions both are short answers with quotes used.

| **Factual Query Set (7B)** |
| --- |
| *Factual prompt 1* |
| What is the symbol for gold on the periodic table? |
| *Completion to factual prompt 1* |
| The symbol for gold on the periodic table is "Au". |
| *Factual prompt 2* |
| What is the common name for the larva of a housefly? |
| *Completion to factual prompt 2* |
| The common name for the larva of a housefly is "maggot" |

See Table 22 below for a representative high correlation example for the 35B slopes questions (Pearson's R of 0.89):

We see that the numbers for the questions are different, but the steps to get to the answer are the same and formatted similarly. We generally observe that a combination of different numbers used in the same procedure and similarly formatted reasoning steps results in higher correlation. Another

Table 20: Two query examples for which the Pearson's R correlation of their document influences is 0.08, both from the factual set of the 35B model. We observe that the main similarity between these queries is that they both concern short "what" questions, and the completions both are relatively short answers with quotes used.

| Factual Query Set (35B) |
| --- |
| *Factual prompt 1* |
| What is the name of the main antagonist in the Shakespeare play "Othello"? |
| *Completion to factual prompt 1* |
| The main antagonist in Shakespeare's play "Othello" is Iago, Othello's standard-bearer. Iago is a manipulative and deceitful character who orchestrates the tragedy in the play. |
| *Factual prompt 2* |
| What is the name of the smallest member of the camel family? |
| *Completion to factual prompt 2* |
| The smallest member of the camel family is called a "dromedary". Dromedaries are easily recognizable by their single hump. They are native to North Africa and the Middle East. |

Table 21: Two query examples for which the Pearson's R correlation of their document influences is -0.03, one from the factual set (top), and one from the factual control set (bottom). We observe that although the formatting is similar, and there are some overlapping concepts, there is no correlation between the document influences.

| Factual and Control Query Set (35B) |
| --- |
| *Factual prompt* |
| What is the supernova remnant and pulsar wind nebula in the constellation of Taurus called? |
| *Completion to factual prompt* |
| The supernova remnant and pulsar wind nebula in the constellation Taurus is called the Crab Nebula. |
| *Factual control prompt* |
| In a fictional planetary system, there is a supernova remnant and pulsar wind nebula in the centre of a constellation of stars called Minotaur called the Goat Nebula. What is the supernova remnant and pulsar wind nebula in the constellation Minotaur called? |
| *Completion to factual control prompt* |
| In the fictional planetary system, the supernova remnant and pulsar wind nebula in the centre of the constellation Minotaur is called the "Goat Nebula". |

example is shown in Table 23 below for a query combination that gets a lower correlation (0.55), but has the same answer (-22). The relatively lower correlation here seems driven by the different formatting of the completion (less new lines). This indicates that part of the correlations are driven by superficial similarities such as completion formatting.

For comparison, Table 24 shows a pair of control queries for which the prompt and completion are also very similarly formatted, and the correlation is 0.38 (which is the highest correlation we find using a control query).

However, superficial formatting cannot be the full picture, as we show in Table 25, where the correlation is 0.35 but the completions are very dissimilar. For this particular query, it seems that it's not the positive scores that explain the correlation (Pearson's R of about 0.2), but rather the negative scores (Pearson's R of 0.44). Perhaps some of the correlation can be explained by documents that are generally negatively influential for the types of completions the models generate here.

Finally, in Table 26 and 27 we show two relatively high correlation examples for the arithmetic queries, again indicating that both underlying operations as well as formatting drive correlations.

Table 22: Two query examples for which the Pearson's R correlation of their document influences is 0.89, both from the reasoning query set concerning calculating slopes. We observe that, although the reasoning is applied to different numbers, the steps follow an identical pattern. In fact, the only difference between the two queries, including completions, is the numbers used.

| **Reasoning Query Set (35B)** |
| --- |
| *Slopes (prompt 1)* |
| What is the slope of the line passing through the points (94, 62) and (95, 19)? |
| Think step-by-step. |
| *Completion to prompt 1* |
| The slope of a line passing through two points is calculated using the formula: |
| Slope = (change in y) / (change in x) |
| In this case, we have the points (94, 62) and (95, 19). |
| Substituting the values into the formula, we have: |
| Slope = (19 - 62) / (95 - 94) |
| Slope = (-43) / (1) |
| Slope = -43 |
| Therefore, the slope of the line passing through the points (94, 62) and (95, 19) is -43. |
| *Slopes (prompt 2)* |
| What is the slope of the line passing through the points (90, 20) and (94, 64)? |
| Think step-by-step. |
| *Completion to prompt 2* |
| The slope of a line passing through two points is calculated using the formula: |
| Slope = (change in y) / (change in x) |
| In this case, we have the points (90, 20) and (94, 64). |
| Substituting the values into the formula, we have: |
| Slope = (64 - 20) / (94 - 90) |
| Slope = (44 / 4) |
| Slope = 11 |
| Therefore, the slope of the line passing through the points (90, 20) and (94, 64) is 11 |

Taken together, it seems like correlations can be driven by underlying procedures, formatting of the completion, and other more general things (like "what"-questions in Table 19 and 25). We generally find the highest correlations when procedures and formatting of completions coincide (of which two examples are given in Table 22 and 27). The magnitude of these correlations indicate that almost all of the influence of the 5 million documents in similar for such queries. One interesting possibility is that the query information surrounding the actual numbers generated (which do not seem to drive correlation much at all) is determined by the attention layers (which, besides the dense parameters contained in them, we ignore in this work), connecting potentially to literature attributing reasoning operations to attention heads. An interesting avenue for future work would be investigating this further.

Table 23: Two query examples for which the Pearson's R correlation of their document influences is 0.55, both from the reasoning query set concerning calculating the slope. We observe that the second completion is structured very similarly to the first, except it has less new lines.

| Reasoning Query Set (7B) |
| --- |
| *Slopes (prompt 1)* |
| What is the slope of the line passing through the points (62, 31) and (61, 53)? |
| Think step-by-step. |
| *Completion* |
| |
| The slope of a line passing through two points is calculated using the formula:

Slope = (change in y) / (change in x)

In this case, we have the points (62, 31) and (61, 53).

Substituting the values into the formula, we have:

Slope = (31 - 53) / (62 - 61)

Slope = (-22) / (1)

Slope = -22

Therefore, the slope of the line passing through the points (62, 31) and (61, 53) is -22 |
| *Slopes (prompt 2)* |
| What is the slope of the line passing through the points (79, 47) and (80, 25)? |
| Think step-by-step. |
| *Completion* The slope of a line passing through two points is calculated using the formula:
Slope = (change in y) / (change in x)

In this case, we have the points (79, 47) and (80, 25).

Substituting the values into the formula, we have:
Slope = (25 - 47) / (80 - 79)
Slope = (-22) / (1)
Slope = -22

Therefore, the slope of the line passing through the points (79, 47) and (80, 25) is -22. |

**7B vs 35B**

An additional finding that is not central to the research question in this work, but is nonetheless interesting, is that there is almost no correlation between the influence scores of the two different models. We have 36 queries that share the same prompt for the 7B and 35B (16 factual questions, and 20 slopes reasoning questions) and we can calculate the Pearson's R of the queries with matched prompts (i.e. 36 combinations). The average correlation of influence scores is 0.02 Pearson's R (if we only look at the slopes questions the average correlation is 0.03). The maximum correlation we find is 0.19, for the question *"What is the capital of Belgium?"*, which we know from above is not a comparatively high score correlation. Interestingly, for this query, both models produced the exact same completion, and still the correlation is comparatively low. All other query combinations correlate with a Pearson's R below 0.11. This connects to a finding from Grosse et al. (2023) (larger models rely on data that is more abstractly related to the prompt): the 35B model relies on very different pretraining data than the 7B, and the same pretraining documents influence completions for the same prompt very differently.

Table 24: Two query examples for which the Pearson's R correlation of their document influences is 0.38, both from the reasoning control set for the 7B model. We observe that the formatting is very similar, but the correlation is still lower than for the reasoning queries above.

| **Reasoning Control Query Set (7B)** |
| --- |
| *Reasoning Control (prompt 1)* |
| Find the answer: There are 21 kids in the bus. How many kids are in the bus? |
| Think step-by-step. |
| *Completion* |
| There are 21 kids in the bus. |
| *Reasoning Control (prompt 2)* |
| Find the answer: There are 132 marbles in a jar. How many marbles are in the jar? |
| Think step-by-step. |
| *Completion* |
| There are 132 marbles in the jar. |

Table 25: Two query examples for which the Pearson's R correlation of their document influences is 0.35, both from the reasoning control set for the 35B model. We observe that their completions are very different, indicating that it can not only be superficial similarities that drive correlations.

| **Reasoning Control Query Set (35B)** |
| --- |
| *Reasoning control (prompt 1)* |
| This line has a slope 6. What is the slope of the aforementioned line? |
| Think step-by-step. |
| *Completion* |
| The slope of a line is a measure of the steepness of the line.
It is defined as the ratio of the change in the y-coordinate to the change in the x-coordinate.

In this case, the line has a slope of 6. This means that for every 1 unit increase in the x-coordinate, the y-coordinate will increase by 6 units.

It's important to note that the slope of a line is not limited to positive values and can be negative or zero depending on the orientation of the line |
| *Reasoning control (prompt 2)* |
| Find the answer: there are two bananas on a tree, how many bananas are on the tree? |
| Think step-by-step. |
| *Completion* |
| There are two bananas on the tree. |

Table 26: Two query examples for which the Pearson's R correlation of their document influences is 0.49, both from the reasoning query set concerning two-step arithmetic. We observe that one of the operations required is overlapping (+ 6), and the completion is somewhat similarly formatted but not fully.

| **Reasoning Query Set (7B)** |
| --- |
| *Arithmetic (prompt 1)* |
| Calculate the answer: (5 + 6) * 10

Think step-by-step. |
| *Completion* |
| First, let's evaluate the expression inside the parentheses:
5 + 6 = 11

Now, let's multiply the result by 10:
11 * 10 = 110

Therefore, the answer is 110 |
| *Arithmetic (prompt 2)* |
| Calculate the answer: (6 + 10) * 6

Think step-by-step. |
| *Completion* |
| First, let's evaluate the expression inside the parentheses:
6 + 10 = 16

Now, let's calculate the result of multiplying the expression inside the parentheses by 6:
16 * 6 = 96

Therefore, the answer to the expression is 96 |

Table 27: Two query examples for which the Pearson's R correlation of their document influences is 0.94, both from the reasoning query set concerning two-step arithmetic. We observe that not only is the completion very similarly formatted, but one of the operations overlaps (multiplying something by 2).

| **Reasoning Query Set (7B)** |
| --- |
| *Arithmetic (prompt 1)* |
| Calculate the answer: (5 - 3) * 12 |
| Think step-by-step. |
| *Completion* |
| First, let's evaluate the expression inside the parentheses: 5 - 3 = 2 |
| Now, let's multiply the result by 12: 2 * 12 = 24 |
| Therefore, the answer is 24 |
| *Arithmetic (prompt 2)* |
| Calculate the answer: (4 - 2) * 15 |
| Think step-by-step. |
| *Completion* |
| First, let's evaluate the expression inside the parentheses: 4 - 2 = 2 |
| Now, let's multiply the result by 15: 2 * 15 = 30 |
| Therefore, the answer is 30 |

### A.9.2 MAGNITUDE OF INFLUENCE

**Additional results finding 2 (magnitude of influence is much lower and less volatile for reasoning questions).**

In the main paper, we find that the influence of documents at the same rank for factual questions is much more volatile than for reasoning questions. We mention that one explanation for this might be that the queries for the 35B model are much more niche, and therefore the relevant documents much more infrequent. To test this hypothesis, we plot the same results for only the overlapping queries (those that are part of both query sets for the 7B and 35B) in Figure 14. We find that the magnitude and variance is still larger for the 35B model than for the 7B model, indicating that the influence of influential documents for the factual and reasoning questions by the 35B can be much larger than for the 7B model. Further, in Figure 15 we show that the results look similar for the negative portions of the ranking (where we flip the influence scores from negative to positive).

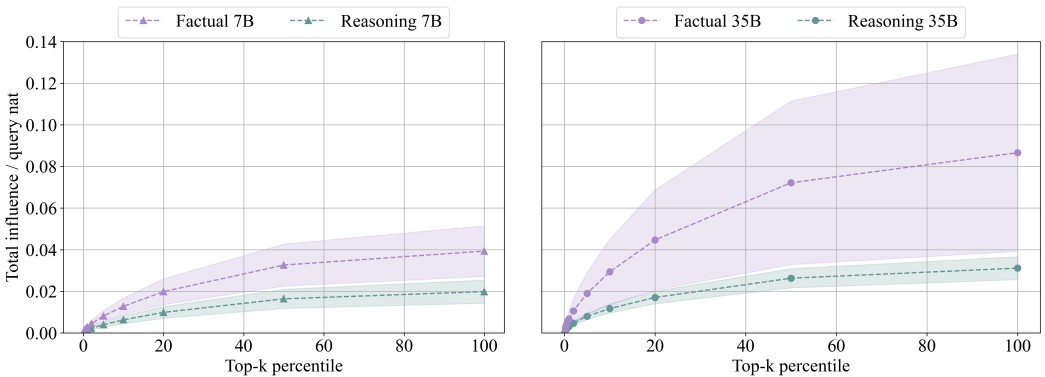

Figure 14: The total influence per nat of query completion information for different portions of the *positive* ranking over documents, left for the 7B model, right for the 35B. In this case, we only plot queries that are present in the query sets for both models. This means the prompt is the same, but the completion is be different. The pattern is very similar as the observed pattern for the top of the ranking.

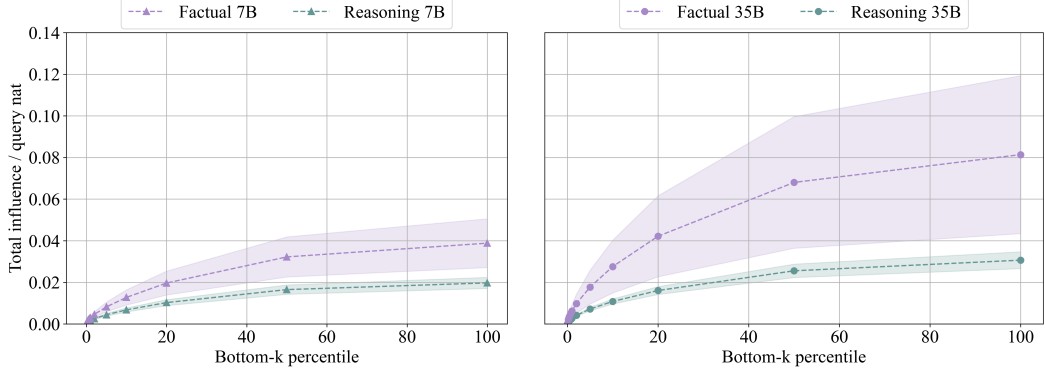

Figure 15: The total influence per nat of query completion information for different portions of the *negative* ranking over documents, left for the 7B model, right for the 35B. We again only plot queries that are present in the query sets for both models. In this case, the $k$-th percentile contains the top $k$ % of most negatively influential documents. The pattern is very similar as the observed pattern for the top of the ranking.

Finally, in Figure 16 and Figure 17 we plot the same metric for all queries for the top and bottom parts of the rankings respectively, now including the 10 control set queries of the factual and

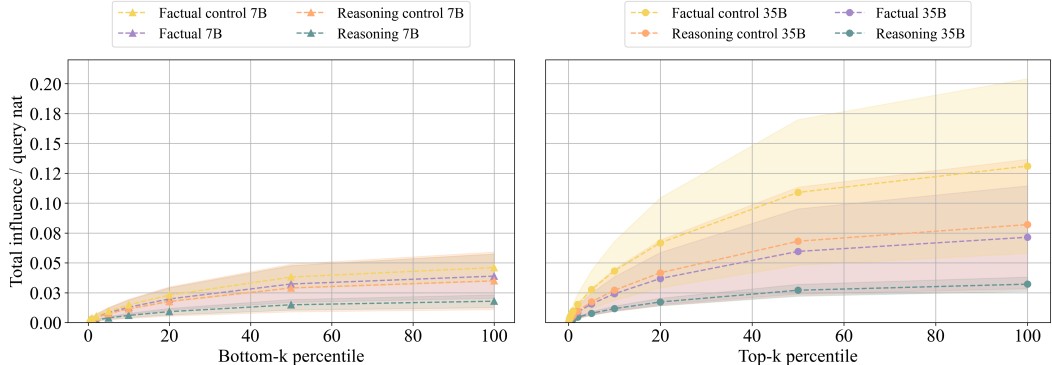

Figure 16: The total influence per nat of query completion information for different portions of the *positive* ranking over documents, left for the 7B model, right for the 35B. We plot all queries, including the query control sets for both factual and reasoning, which contain 10 queries each.

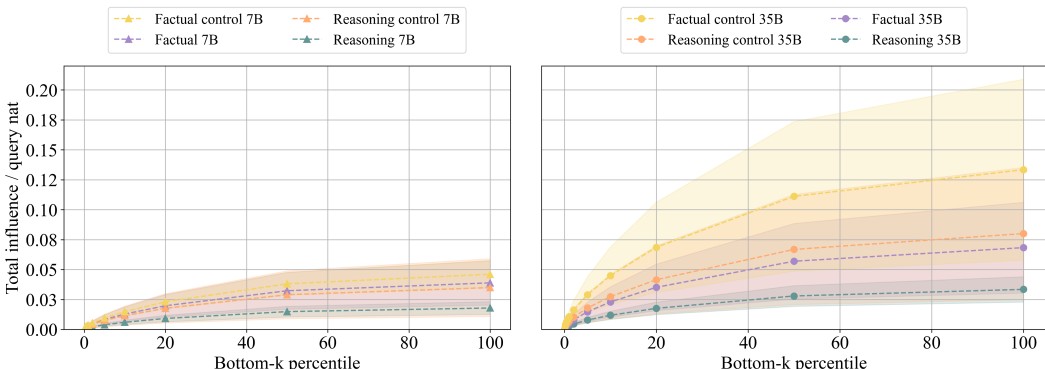

Figure 17: The total influence per nat of query completion information for different portions of the *negative* ranking over documents, left for the 7B model, right for the 35B. We plot all queries, including the query control sets for both factual and reasoning, which contain 10 queries each.

reasoning control set. As shown in Appendix A.3, we use 10 control queries for each set to investigate whether results hold similarly for queries that superficially look similar as the factual/reasoning questions, but that do not require factual retrieval or reasoning respectively. We observe that the control sets both show much higher variance and magnitude than the reasoning queries as well, for the positive and negative portions of the ranking. For completeness, we show the same result with the number of documents on the x-axis instead of percentiles in Figure 18 and Figure 19, to show that the results are similar if we take into account that the 20-th percentile of documents for each query contains a different amount of documents $k$.

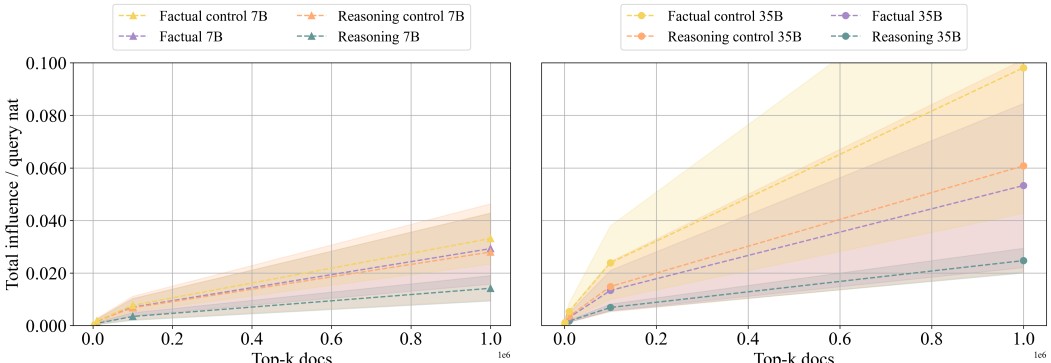

Figure 18: The total influence per nat of query completion information for different number of documents $k$ of the *positive* ranking, left for the 7B model, right for the 35B. We plot all queries, including the query control sets for both factual and reasoning, which contain 10 queries each.

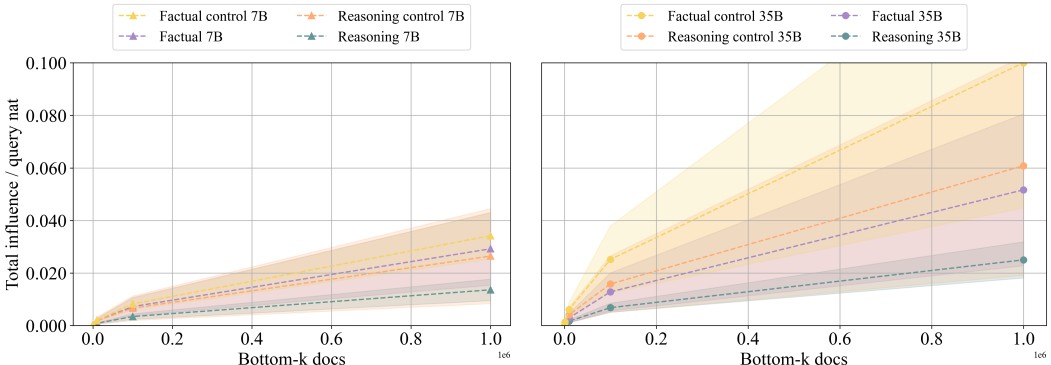

Figure 19: The total influence per nat of query completion information for different number of documents $k$ of the *negative* ranking, left for the 7B model, right for the 35B. We plot all queries, including the query control sets for both factual and reasoning, which contain 10 queries each.

### A.9.3 INFLUENCE SPREAD: POWER LAWS

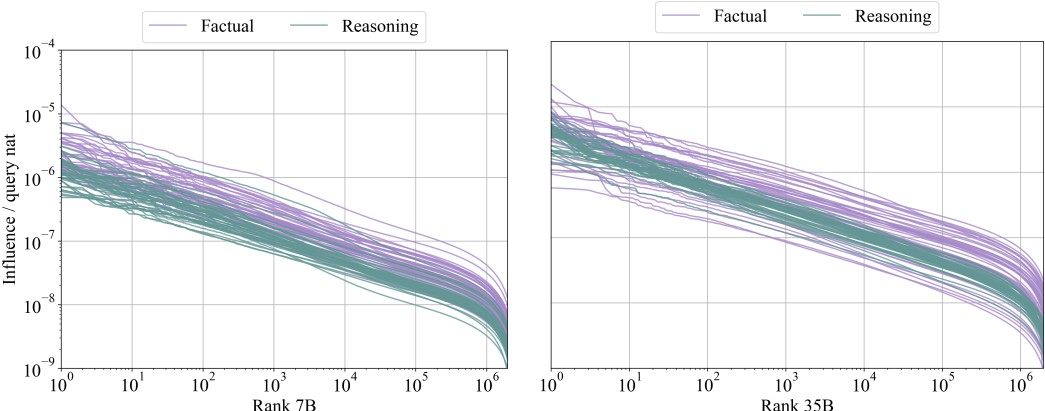

Figure 20: The ranked influence scores per query nat for each query shown separately in log-log space. We observe; the results follow power laws (linear in log-log space), everything is shifted up for the 35B model (right), generally the scores for the reasoning documents are lower for the 7B model, and for the 35B model there is less variance in magnitude of influence for reasoning queries than for factual queries, and more often than not the influence scores are lower than for factual questions.

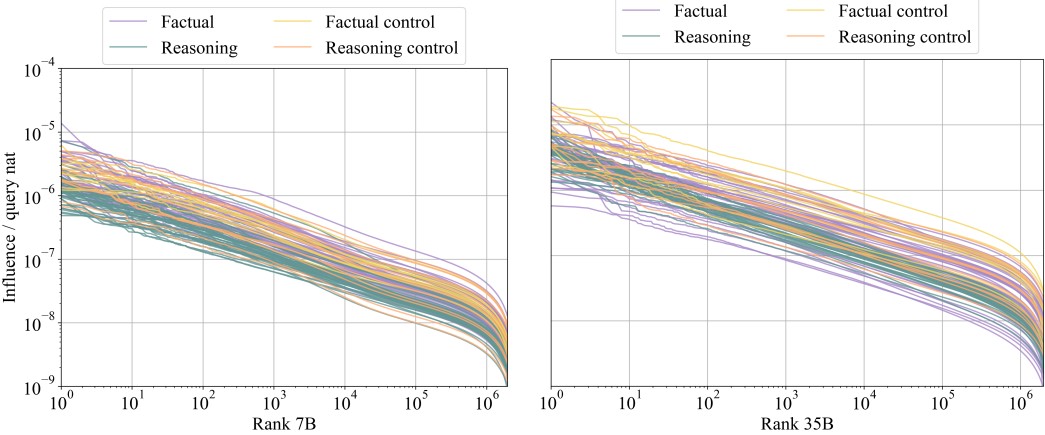

Figure 21: The ranked influence scores per query nat for each query shown separately in log-log space again, but now also showing the control queries. We observe that also for the control queries the influence is much more volatile than for reasoning questions, and on average the magnitude is higher.

In this section, we look at the power laws induced by the top portions of the rankings. We can fit linear functions to the rankings in log-log space, and analyse the slopes to comment on the sparsity of the rankings (i.e. how many documents do models rely on for a completion). Specifically, we perform linear regression on the log-log top 500 rankings of each query, and report the slopes in Table 28.

After qualitatively inspecting the queries for the 35B model with the steepest slope, we believe an explanation for this result may be 'noise' in the influence scores. For example, the query with the steepest slope ($\alpha = -0.45$) has as the most influential document a document that is seemingly entirely unrelated to the query. Namely, the query asks the question *"What is the slope of the line passing through the points (41, 23) and (18, 92)? Think step-by-step."*, and the top influential

Table 28: Slopes of the fitted functions to the top 500 documents in the influence rankings in log-log space, separated by query set and whether the model gets the question right or wrong. $\star$ indicates the significance of an independent T-test performed between the slopes of the factual vs. reasoning queries, where $\star$ indicates a p-value below $0.1$ and $\star\star$ below $0.05$.

| | 7B (Correct) | 7B (Incorrect) | 35B (Correct) | 35B (Incorrect) |
|---|---|---|---|---|
| Reasoning ($\alpha$) | $-0.36 \pm 0.03^\star$ | $-0.33 \pm 0.02$ | $-0.36 \pm 0.04^{\star\star}$ | $-0.38 \pm 0.04^\star$ |
| Factual ($\alpha$) | $-0.34 \pm 0.03$ | $-0.34 \pm 0.04$ | $-0.32 \pm 0.05$ | $-0.34 \pm 0.04$ |

document is a snippet about the lunar eclipses and when and where they can be viewed which does not have high N-gram overlap with the query either:

> *December 8, 1946 — Total Lunar Eclipse — Rawaki, Phoenix Islands, Kiribati*
> *Max view in Rawaki*
> *Sunday, December 8, 1946 at 5:01 AM*
> *Global Type: Total Lunar Eclipse*
> *Rawaki: Partial Lunar Eclipse*
> *Began: Sun, Dec 8, 1946 at 3:13 AM*
> *Maximum: Sun, Dec 8, 1946 at 5:01 AM*
> *Ended: Sun, Dec 8, 1946 at 8:22 AM*
> *Duration: 5 hours, 10 minutes*
> *December 8, 1946 — Total Lunar Eclipse — Rawaki*
> *You are using an outdated browser, to view the animation please update or switch to a modern browser. Alternatively you can view the old animation by clicking here.*
> *Animation: How the Partial Lunar Eclipse Looked*
> *The total phase of this lunar eclipse was not visible in Rawaki, but it could be observed there as a partial lunar eclipse.*
> *More about the December 8, 1946 — Total Lunar Eclipse*
> *Phases and local times of this eclipse*
> *Eclipses visible in Rawaki*
> *All eclipses worldwide, from 1900 to 2100*

This is the only query for which we observe an unrelated top 1 document, but for the 35B model we qualitatively observed seemingly irrelevant documents in the rankings more often (in the 7B we did not observe this). This connects to a finding from literature that for large models influence functions sometimes surface documents with high gradient norms that are unrelated to the query (Barshan et al., 2020; Grosse et al., 2023; Choe et al., 2024). As Grosse et al. (2023) note, it is currently unclear whether this is true noise, or whether these are genuinely influential for the completions. Regardless, it seems like noise cannot easily explain the difference between the factual and slopes queries, as one would expect noise to show up equally everywhere.

Another way to visualise this result is to plot the percentage of total influence contained in different parts of the top ranking, which we do in Figure 22 below. The results in this plot show that for the top-k percentile of most positively influential documents, the total percentage of positive influence is much higher than $k$ (e.g. $20\%$ of the total positive influence is contained in the top $5\%$ of documents). Here, it is clear that on average, for the 35B model the total amount of influence contained in the top-$k$ percentile increases faster for reasoning questions than for factual questions, indicating that a larger portion of the total positive influence is contained in the top portions of the rankings. In Figure 23 we show the same result holds if we include the control queries. As Grosse et al. (2023), it is not clear whether this is a sensible result to show because for each query we are dividing the total influence at each $k$ by the sum of positive influence for that query (perhaps a large part of the positive influence gets cancelled out by negative influence), but we show the result here nonetheless for completeness. We know from the absolute results of the total influence at different portions of the ranking that each percentage of total influence at the top-$k$ percentile a much lower value in absolute terms for reasoning than for the factual questions. If the relative result does not turn out to be noise, it is the case that of the total influence, a higher percentage is contained in the top portions

of the rankings for reasoning questions than for factual questions. Taken together with the fact that the absolute influence is often much higher for factual questions, this indicates that the model relies on more highly influential documents for factual retrieval than for reasoning. This could indicate that there are more highly relevant factual documents further down the ranking, which makes sense given the fact that the pretraining distribution is dominated by websources and news, which are more likely to contain relevant information for factual question answering than for reasoning. Further, it connects to the finding from literature that models need to see examples often before text gets memorised (Chowdhery et al., 2022).

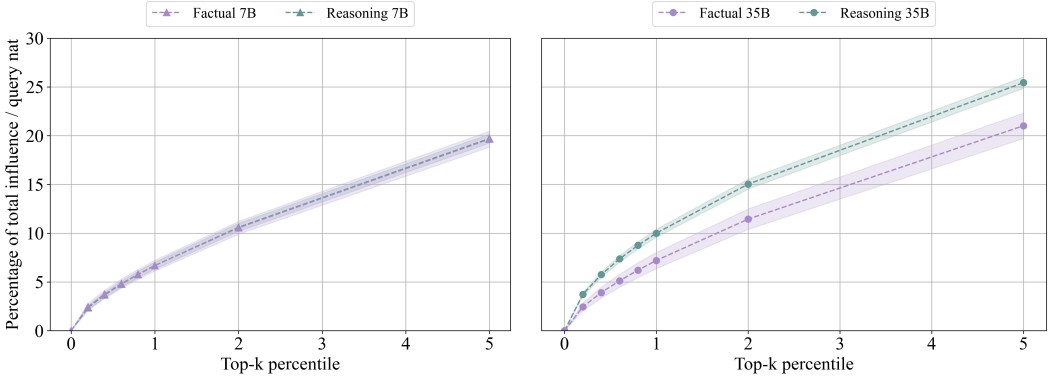

Figure 22: The percentage of total influence per nat of query completion information for different portions of the *positive* ranking over documents, left for the 7B model, right for the 35B. We plot only non-control queries.

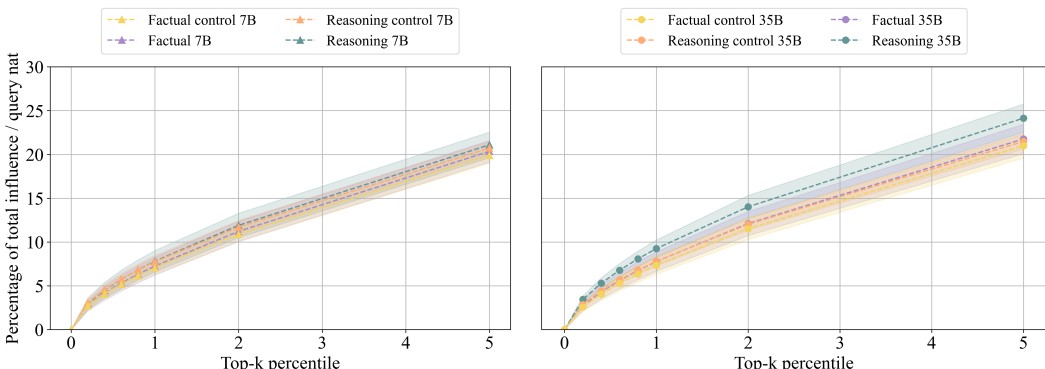

Figure 23: The percentage of total influence per nat of query completion information for different portions of the *positive* ranking over documents, left for the 7B model, right for the 35B. We plot all queries, including the query control sets for both factual and reasoning, which contain 10 queries each.

Again, the picture looks similar for the negative portions of the ranking, shown for completeness below in Figure 24 and 25.

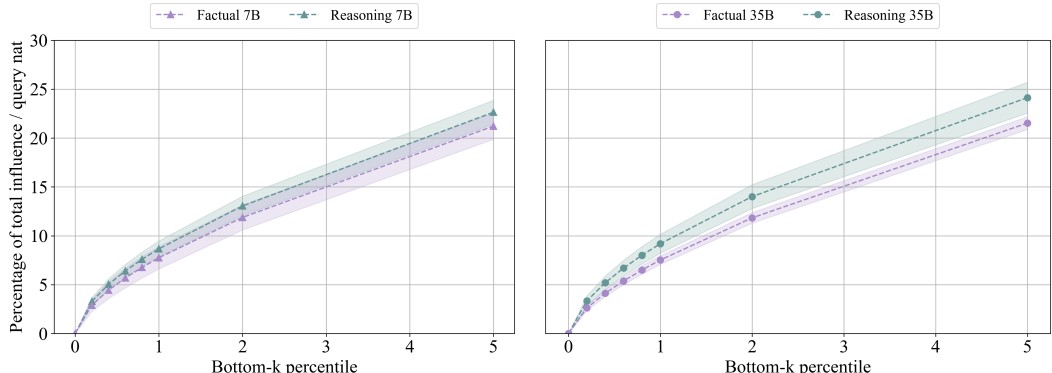

Figure 24: The percentage of total influence per nat of query completion information for different portions of the *negative* ranking over documents, left for the 7B model, right for the 35B. We plot only non-control queries.

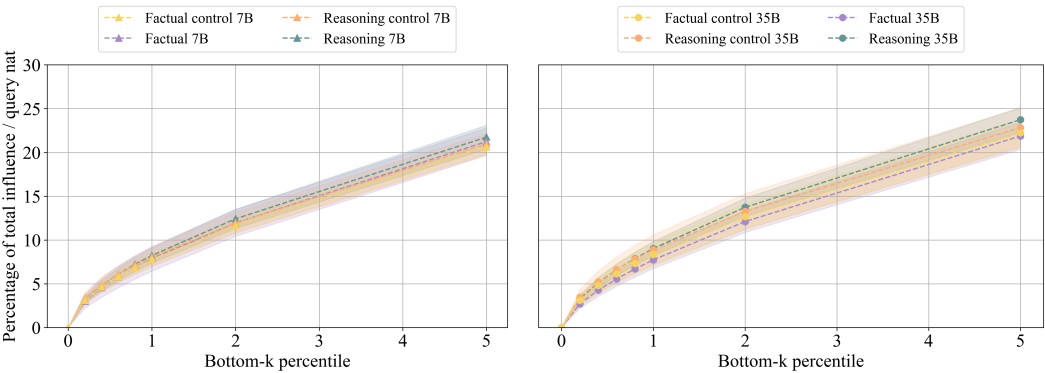

Figure 25: The percentage of total influence per nat of query completion information for different portions of the *negative* ranking over documents, left for the 7B model, right for the 35B. We plot all queries, including the query control sets for both factual and reasoning, which contain 10 queries each.

