# OpenReview forum: "Procedural Knowledge in Pretraining Drives Reasoning in Large Language Models"
_ICLR.cc/2025/Conference — ICLR 2025 Poster_

### Official Review · Reviewer_KXBG · 2024-10-30

**Soundness:** 3
**Presentation:** 3
**Contribution:** 2
**Rating:** 5
**Confidence:** 4

**Summary:**

This paper explores the influence of specific pretraining data on the reasoning abilities of large language models (LLMs), focusing on how models rely on different types of documents when responding to reasoning versus factual queries. The paper applies influence functions to identify pretraining documents that impact performance on simple reasoning tasks. Results show that factual questions often depend on a smaller set of documents containing the answer, whereas reasoning questions are more influenced by documents with procedural knowledge.

**Strengths:**

1. The proposed method is straightforward, well-explained, and includes sufficient detail, making it easily reproducible.
2. This research addresses a crucial question: identifying what training data impacts LLM reasoning abilities, an area closely tied to model generalization and interpretability. It contributes to our understanding of LLMs.
3. The paper presents intriguing findings, highlighting distinctions in how LLMs handle factual versus reasoning tasks. For instance, factual questions frequently retrieve specific information, while reasoning tasks benefit from procedural knowledge.

**Weaknesses:**

1. The experimental setup is limited, potentially compromising the reliability of conclusions. Specifically: (1) only 80 queries were used for analysis, (2) the study included only three types of reasoning tasks, potentially limiting representation to other reasoning tasks, (3) there was no exploration of how different prompt formulations of the same query affect results, and (4) keyword-based methods for determining whether documents contain answers may be insufficiently accurate.
2. The analysis may lack granularity, as it considers only each document’s influence on the overall completion without examining its impact on individual reasoning steps. This might affect the conclusions.
3. While Appendix A.1 reports that influence scores are higher for certain documents, their similarity to random selections raises questions about whether influence functions reliably indicate actual influence.

**Questions:**

1. Why were these two specific LLMs chosen, instead of more widely used and capable models?
2. Using both fine-tuned and base models in the same experiment could lead to unreliable results due to differences in parameter initialization, potentially affecting influence calculations.
3. Since LLMs rely on embedded representations, even if keyword matching fails to find an answer, does it conclusively mean the document is not similar to the answer?
4. Could examples of retrieved documents for reasoning tasks be provided to offer insights into how they influence the model's approach to reasoning?

---

> ### Author Response · Authors · 2024-11-18
> **Author response - part 1/3**
>
> We thank the reviewer for their review, saying our *“method is straightforward, well-explained, and includes sufficient detail, making it easily reproducible”*, it *“addresses a crucial question”*, the *“paper presents intriguing findings.”*, and it *“contributes to our understanding of LLMs”*. We respond to each weakness mentioned below separately, and answer all questions.
>
> **Weakness 1 - (1) and (2)**: *“The experimental setup is limited, potentially compromising the reliability of conclusions.”*
>
> We would like to kindly refer the reviewer to [the general comment above](https://openreview.net/forum?id=1hQKHHUsMx&noteId=ZZ8uSWvlC0) for a response to this point, as multiple reviewers have raised this. To summarise here, although we agree the setup in terms of tasks and models is limited (which is due to a hard compute constraint), we respectfully disagree that this will compromise the reliability of the conclusions (explained in the general comment). As reviewer **6knH** also points out, despite the narrow scope of the experiments, we are careful to qualify any claims to make sure they are well supported. Further, in the general comment we highlight that the scope is very large w.r.t. most research.
>
> **Weakness 1 - (3)**: *“there was no exploration of how different prompt formulations of the same query affect results”*
>
> This is a valuable suggestion and aligns closely with considerations we have thought about (e.g. how do the rankings change for the same reasoning question with different zero-shot prompts). However, we believe it falls outside the scope of the current work, as it would not change conclusions. To illustrate why we believe this; we might find different results for different prompt formulations (e.g. a retrieval-like strategy for reasoning). This would fit with prior work on dependence of models to prompt formulation, but would still mean models can in principle learn a generalisable strategy for reasoning with the right prompt. Alternatively, we do not find different results with different prompt formulations, which would be interesting as well. To highlight a snippet from the submission related to this: *“[...] we do not claim to say [...] that LLM reasoning is not brittle. All we showed is that in principle it seems to be possible for LLMs to produce reasoning traces using a generalisation strategy that combines information from many abstractly related documents, as opposed to doing a form of retrieval”*
>
> **Weakness 1 - (4) and question (3)**: *“keyword-based methods for determining whether documents contain answers may be insufficiently accurate.”*
>
> These are good points, and we agree with the reviewer. However, we would like to point out that we also use methods independent of keyword overlap. We both manually look over keyword hits, and give all query-doc pairs of the top 500 documents for each query to Command R+ (a 100B model) to identify documents with the answer independently of keyword overlap. We confirmed that this method found all documents we found manually and more that eluded the keyword search. We made this clearer in the revision (we use the colour purple to highlight revisions in response to your review, and this particular revision can be found in Section 5.2, Finding 3, L406-407).
>
> **Weakness 2**: *“The analysis may lack granularity, as it considers only each document’s influence on the overall completion without examining its impact on individual reasoning steps. This might affect the conclusions.”*
>
> Calculating influence on the individual reasoning steps is an intriguing suggestion, but it is unclear to us how this would change the conclusions in the paper. The influence scores for the full completion reflect influence on all reasoning steps (which are highly correlated because they are generated linearly taking into account all previous context), and given the correlation observed for influence scores of queries of the same reasoning type, we expect the rankings for the individual reasoning steps to be very similar to the ones we find now. Although this is an interesting suggestion for a more fine-grained analysis, we would be grateful if the reviewer could further clarify how its results could affect our conclusions.

---

> > ### Author Response · Authors · 2024-11-18
> > **Author response - part 2/3**
> >
> > **Weakness 3**: *"While Appendix A.1 reports that influence scores are higher for certain documents, their similarity to random selections raises questions about whether influence functions reliably indicate actual influence.”*.
> >
> > Thanks for raising this point, as it allows us to clarify an important nuance in the interpretation of influence functions that was not clear enough in the submission. That influence functions reliably estimate actual influence is a well-documented area of research (e.g. section 5.1 in [1] for similar architectures like the ones we look at). The claims in our paper do not rely on influence functions empirically estimating a causal effect on accuracy as well, but it helps interpret results and was previously unknown. To contextualise Appendix A.1; it was a priori unclear that the experiments were sensible, because influence functions estimate the effect of removing a single document. However, because accuracy is a discrete metric, it is unclear how many documents one needs to remove from the training set in order to affect the model parameters just enough to flip the accuracy. We need to remove multiple documents at once, but that might have unexpected interactional effects that influence functions do not account for. Therefore, any empirical experiment to test this is going to be a crude measure, because random removal as a baseline will also have an effect on accuracy. Considering all this, it’s an important encouraging signal that accuracy is still significantly more impacted by taking out documents with influence functions. If we would’ve found the same effects on accuracy as randomly taking out documents, we couldn’t have claimed influence functions estimate no effect on accuracy for the above reasons. We tried to make the motivation of A.1 clearer in the revision, on L200-202 (colour-coded purple). We also rewrote part of A.1 to make the nuance clearer (L797-799 and L948-956 in the Appendix).
> >
> > **Question 1**: *“Why were these two specific LLMs chosen, instead of more widely used and capable models?”*
> >
> > For our experiments, we need access to the pretraining distribution. None of the widely used models publish their pretraining data, and further many openly available models that do publish the pretraining distribution (such as Pythia), are not able to generate zero-shot reasoning traces for mathematical tasks such as the ones we investigate.
> >
> > **Question 2**: *“Using both fine-tuned and base models in the same experiment could lead to unreliable results due to differences in parameter initialization, potentially affecting influence calculations.”*
> >
> > Using different models for calculating the influence scores is a method called SOURCE [1], and effectively we are assuming that the fine-tuning stage second order information is the identity. This means we are ignoring the second-order impact on the completions of the fine-tuning stage. We argue that this is unlikely to impact conclusions, because prior work has shown that SFT serves primarily to enhance existing model capabilities as opposed to endowing them with new ones [2], [3], [4]. Further, the fine-tuning stage consisted of a couple thousand supervised instruction-tuning steps on top of the base model we use, which is negligible compared to the pretraining stage. Nonetheless, we believe an interesting direction for future work would be to apply the same method used here to the fine-tuning stage. We hypothesise that this might surface documents that are similar in formatting to the queries, as opposed to documents that are similar in content. We dedicated a few lines to this question in the new discussion in the revision (L513-518, colour-coded orange), and copy here: *“Another limitation is that we do not look at the supervised fine-tuning stage. The reason we only look at the pretraining data is because the fine-tuning stage is targeted at making the models more aligned and ‘instructable’, as opposed to teaching the model any new capabilities. Prior work has shown that SFT serves primarily to enhance existing model capabilities (Jain et al., 2024; Kotha et al., 2024; Prakash et al., 2024). Nonetheless, an interesting direction for future work is applying the same method used here to the fine-tuning data.”*
> >
> > [1] Training Data Attribution via Approximate Unrolled Differentiation; Bae et al. 2024
> > [2] Mechanistically analyzing the effects of fine-tuning on procedurally defined tasks, Jain et al. 2024
> > [3] Understanding catastrophic forgetting in language models via implicit inference. Kotha et al., 2024
> > [4] Fine-tuning enhances existing mechanisms: A case study on entity tracking, Prakash et al. 2024

---

> > > ### Author Response · Authors · 2024-11-18
> > > **Author response - part 3/3**
> > >
> > > **Question 4**: *“Could examples of retrieved documents for reasoning tasks be provided to offer insights into how they influence the model's approach to reasoning?”*
> > >
> > > Yes! We are working on releasing the top and bottom 20 documents for each query, all documents with answers to questions, and all documents with the procedures to calculate the slope as mentioned in one of the qualitative findings. Together, this will cover thousands of document examples, and until we are ready to release all of them (which requires internal approval), we already uploaded about 80 documents with answers to questions and slope procedures to the supplement to show that we are working on it. We hope to upload all by the end of the discussion period.
> > >
> > > We thank the reviewer again for their time and their review. We are happy to discuss remaining weaknesses if they are not addressed by the above, and hope the reviewer would consider raising their score if weaknesses are addressed.

---

> > > > ### Comment · Reviewer_KXBG · 2024-11-27
> > > >
> > > > Thank you for your effort in addressing my concerns. I appreciate the clarifications provided, which have resolved some of the issues. However, I still find the experimental settings to be somewhat limited. However, I understand the inherent challenges in investigating pretraining data. Therefore, I do not oppose its acceptance if other reviewers believe it meets the necessary standards for ICLR. For now, I will maintain my score.

---

> > > > > ### Author Response · Authors · 2024-11-27
> > > > >
> > > > > Dear reviewer,
> > > > >
> > > > > Thanks a lot for the follow-up response and recognising the efforts we have made to address your concerns. We appreciate your acknowledgement that some issues have been resolved.
> > > > >
> > > > > We would be grateful if you could elaborate on your outstanding concerns and what a satisfactory and reasonable resolution might look like to you, to ensure that—if they are the byproduct of some outstanding misunderstanding, we address them in the paper, and if not, that we address them in follow-on experiments. Either way, we are eager to ensure these points are thoughtfully addressed, regardless of the outcome of this paper’s acceptance.
> > > > >
> > > > > For context, we'd like to note that our investigation goes beyond prior research in scale. Grosse et al. (2023), who are the first ones to apply EK-FAC influence functions at a similar scale, investigated 29 queries, where we look at 100 queries (which, at a lower-bound, took ~424448 TPU chip-hours). Moreover, we have control sets that highlight the findings for the reasoning queries are not spurious, and the results are statistically *highly* significant, for example for the correlation results with p-values below $4e^{-8}$. It would be very helpful if you could clarify how you believe the experimental setup might still limit the conclusions and in what ways it might affect our findings.
> > > > >
> > > > > Finally, we would like to kindly point out that all other reviewers indicated the work meets the publishing standards for ICLR (8, 8, 6). Given that your comment suggest you are open to its acceptance if the other reviewers believe it meets the necessary standards, we hope you might consider revisiting your score to reflect this position.
> > > > >
> > > > > Thank you again for your time and your engagement with our work.

---

> > > > > > ### Author Response · Authors · 2024-11-30
> > > > > >
> > > > > > Dear reviewer KXBG,
> > > > > >
> > > > > > Given that the discussion period ends soon, we wanted to check in if our provided responses address your concerns, and see if there are any further questions that we can help address.
> > > > > >
> > > > > > Thanks again for reviewing!

---

> ### Comment · Area_Chair_B6yu · 2024-11-25
> **[Reminder] Response to Authors**
>
> Dear Reviewer,
>
> As the rebuttal period is drawing to a close, I would appreciate your response to the authors' rebuttal at your earliest convenience.
>
> Best Regards,
>
> Area Chair

---

### Official Review · Reviewer_6knH · 2024-11-02

**Soundness:** 4
**Presentation:** 4
**Contribution:** 3
**Rating:** 8
**Confidence:** 4

**Summary:**

This paper applies the EK-FAC influence function to LLMs in an investigation of which documents, from a representative sample of LLM pretraining data, are used by a given model to answer basic mathematical reasoning questions. The EK-FAC influence function is used to produce a score for a given triple (prompt, completion, document) and these scores provide a basis to rank documents as more or less useful in generating the completion for the given prompt. Due to intense computational requirements, this technique is applied on a small sample of 80 prompt/completion pairs, but in great detail, examining several hundred documents at the top of the ranking for each pair. Several key findings emerge, including that models employ documents for reasoning responses in a different manner than for factual responses, and that such mathematical reasoning responses often rely on documents describing verbal procedures or code.

**Strengths:**

1. This paper presents a series of interesting and novel investigations into the influence of documents from pretraining in model responses. Most research in model interpretability is done by examining or modulating model parameters and activations, since it is usually computationally intractable to trace model responses back to pretraining samples; this is frontier research, and I was excited to read it.

2. The paper presents insights into which documents are used to answer mathematical reasoning questions, and crucially provides comparisons between two models within the same family, and also to a secondary task in factual question answering. The latter comparison was especially useful and cleanly conveyed the points made: specifically, that factual responses often rely on a specific document, but evidence is shown that reasoning responses may draw on a breadth of documents, possibly aggregating heterogeneous information into one response.

3. The experiments were extremely narrowly defined, but the authors caveat this early and often throughout the paper. Additionally, even in this narrowly scoped setting approximations must be made in order to be computationally tractable, and the authors honestly qualify discussions with reasonable alternate hypotheses and give sub-experiments to explore what is the most likely hypothesis. This kind of writing is very thoughtful and I appreciated that the authors made reasonable decisions and honestly qualified the claims, which were well supported.

**Weaknesses:**

1. As mentioned above, the experiments were very narrowly scoped. Only 80 questions were analyzed in total, and this 80 was further broken down into smaller sub-groups. Moreover, the questions were very simple mathematical problems using small numbers, requiring only short reasoning hops, and not resulting in fractional-valued answers. The experiments were performed only on two models within one model family, and one model is not available publicly. The authors do note all of these things, and some (not all) of these decisions seem to be made due to computational constraints, which is understandable. However, it would have been nice if these experiments were at least reproduced on fully public models such as Llama.

2. The description of EK-FAC was brief and not as clearly described as the later experiments and results, which were very clear. It would be nice to have a little more motivation about the individual components in the given formulas, since this methodology underlies all of the later experiments. Further, the discussion section at the end of the paper (sec 5) was very dense and a bit confusing. Maybe this could be restructured? The alternating hypotheses in the paragraph starting on L490 were particularly hard to follow.

3. (This is a minor point) Some of the mystique surrounding "reasoning" in LLMs may be because as a field we have conflated many types of problems into one, in the fervor of "AGI". Though this paper often discusses general reasoning, it looks specifically at mathematical reasoning, and it could be made more clear that these studies are distinct from linguistic reasoning, logical reasoning, spatial, etc etc. Analyzing linguistic reasoning provenance would be fascinating using this method, but would require different experiments.

**Questions:**

I have no serious questions for the authors, but if they have time:
1. Can this methodology be applied to model false-positives? It would be interesting to explore how pretraining documents may relate to hallucinations in generative responses, given prior research which points to cases of memorization.

---

> ### Author Response · Authors · 2024-11-18
> **Author response - part 1/2**
>
> We thank the reviewer for a positive review. We were very happy to read that: *“this is frontier research, and I was excited to read it”*, and that the review recognises evaluating factual question answering *“was especially useful and cleanly conveyed the points made”*. In the below, we address your raised weaknesses and questions.
>
> **Weakness 1**: *“As mentioned above, the experiments were very narrowly scoped.”*
> We respond to the first weakness in [the general response to all authors above](https://openreview.net/forum?id=1hQKHHUsMx&noteId=ZZ8uSWvlC0). To add to specific comments by your review here, we agree that it would be great to see these experiments reproduced on other model families. However, Llama specifically is not possible because the pretraining data is not published. On the short reasoning hops not resulting in fractional-valued answers; the reason we did this is two-fold; it is less likely that answers to reasoning steps are in our sample of 5M documents if they contain fractional values, and in many cases expecting an LLM to output fractional values is less reasonable if it does not have access to tools.
>
> **Weakness 2 - part 1**: *“The description of EK-FAC was brief and not as clearly described as the later experiments and results”*
> This is understandable, and it’s useful for us to know that more motivation for using EK-FAC is required. To address this, we added the following line in the main paper: *“In the same experiments, we motivate the use of EK-FAC estimations of the Hessian, by showing it significantly improves over a method using only first-order information.”* (referring to Appendix A.1, see red-coded revisions L210-211). Given the limited space we have in the revision, and because this is background material, we decided to further address your point in the appendix. To summarise here; EK-FAC estimation of the Hessian is a much better estimate of the counterfactual question that we are interested in (*“how do the trained model parameters change if a datapoint is included in the pretraining set and the model is retrained”*) than methods using only first-order gradient information. This is especially true in a regime where many gradient steps are taken such as for LLMs, because second order information becomes even more important. Beyond the motivation of using EK-FAC over first-order methods, we expanded section A.2 of the appendix with two subsections that should address this point, and referred to it in the main paper (see L235, colour-coded red). In A.2.1, we ran additional experiments to motivate each approximation we do. To estimate the Hessian from Equation 1 with EK-FAC tractably for LLM-scale models we use a block-diagonal approximation of the Hessian. We estimate the effect this has on influence scores compared to a full implementation by calculating the correlations in an experiment on Wikitext with GPT-2. We find the scores correlate highly with the full implementation scores (Pearson’s R of 0.96). In the second section we added (A.2.2), we further compare our EK-FAC implementation to a publicly available implementation of EK-FAC influence functions (that correlates with our implementation with 0.996 Pearson’s R), and we share the detailed results of this experiment in the supplement. This provides a reference implementation that can further help with understanding the EK-FAC estimations.
>
> **Weakness 2 - part 2**: *“the discussion section at the end of the paper (sec 5) was very dense and a bit confusing.”*
> This is very valuable feedback, thank you. We have restructured the discussion to add detail and improve clarity. Please refer to the second and third paragraph in the discussion in the uploaded revision (L486-512, colour-coded red). To summarise the changes; we separated the two alternative hypotheses, rewrote them to be clearer, and reframed the second half of the paragraph starting on L490 originally (now on L496) in terms of limitations.
>
> **Weakness 3**: *“it could be made more clear that these studies are distinct from linguistic reasoning”*
> We agree with the point about the field conflating many different forms of “reasoning”, without being too clear about what reasoning is. This is in part why we chose very simple mathematical reasoning, with clear-defined steps that build on each other. We tried to be clear about this, by making a point about saying we look at simple mathematical reasoning tasks in the abstract, and specifying the types in the introduction (right before the summary of the findings). To emphasise again at the end of the paper that this does not mean that our findings would generalise to other forms of reasoning, we added the following line in the discussion: *“Finally, in this work we look at mathematical reasoning, which is very different from other types of reasoning, especially if they are inductive. Future would should verify whether similar results hold for more types of reasoning”* (colour-coded orange, L525-528).

---

> > ### Author Response · Authors · 2024-11-18
> > **Author response - part 2/2**
> >
> > **Question 1**: *“Can this methodology be applied to model false-positives? It would be interesting to explore how pretraining documents may relate to hallucinations in generative responses, given prior research which points to cases of memorization.”*
> > That’s a very interesting suggestion, and yes this should be possible with this methodology. An experimental setup that comes to mind that is even possible with the results we already have is taking completions for factual questions the model gets right and the ones it gets wrong (which are essentially hallucinations as the model makes up an answer) and try to find patterns in the difference between the rankings. Probably a better setup though would be to look at more interesting forms of hallucinations, where the model more freely hallucinates (or indeed false-positives, e.g. where the model identifies something in text that is not there), as opposed to failures of retrieval in response to a factual question. The most interesting would be to get a broad set of hallucinations in completions that otherwise don’t have much to do with each other, and try to find patterns in the most influential data.
> >
> > We were very happy to read your review and excellent summary of the paper, and that you believe the claims are honestly qualified and well-supported. We hope the revisions made in response to your review as well as the explanation of the limited scope address your weaknesses and are happy to discuss further where required. We believe that the improvements made following the feedback have considerably strengthened the positioning of our work. Thank you!

---

> > > ### Author Response · Authors · 2024-11-25
> > > **Author response - request for support**
> > >
> > > Dear reviewer,
> > >
> > > Given that the discussion period is coming to a close tomorrow, we were wondering if you have had the time to look at our responses. We believe we have significantly improved the paper in response to your review. Most notably, we added experimental results motivating the EK-FAC estimation of the Hessian, we comment on the scope of the experiments, and we significantly rewrote the discussion in response to your points. We sincerely hope you can find the time to look over our response and let us know your thoughts, and if your points of weakness have been addressed, whether you would consider further strengthening the support for our submission.
> > >
> > > Thanks again!
> > >
> > > The Authors of Submission 7193

---

> > > > ### Comment · Reviewer_6knH · 2024-11-26
> > > > **Thank you!**
> > > >
> > > > Thank you very much for taking the time to write such a thorough response (even to an already positive review). The changes made in the new revision are clarifying, and themselves quite interesting.
> > > >
> > > > I especially appreciate the additions made in appendix A. It might not be a bad idea to mention in the main text that these parallel experiments in finetuning GPT-2 agreed with the main findings, just for the benefit of readers like me trying to assess reproducibility of the findings across models. I think adding finetuning data for this experiment is a nice answer to the issue of Llama, whose pretraining data is not available.
> > > >
> > > > Overall, I believe this paper would be a strong contribution to ICLR should it be accepted.

---

### Official Review · Reviewer_RvSn · 2024-11-04

**Soundness:** 3
**Presentation:** 3
**Contribution:** 3
**Rating:** 8
**Confidence:** 3

**Summary:**

The paper investigates the generalization strategies employed by LLMs when performing reasoning tasks compared to factual recall. The authors examine the influence of pretraining data on two LLMs of different sizes (7B and 35B parameters) by using influence functions to rank documents based on their impact on the likelihood of model outputs for reasoning and factual questions. They find that for reasoning tasks, LLMs do not rely heavily on direct retrieval of answers from pretraining data but instead use a broader set of documents that contain procedural knowledge relevant to the task. This suggests that LLMs generalize by learning how to perform reasoning steps rather than memorizing specific solutions. In contrast, for factual questions, the influential documents often directly contain the answers. The authors also note the overrepresentation of code in influential documents for reasoning, indicating its importance in teaching procedural knowledge to the models.

**Strengths:**

- The paper provides an important insight of LLMs, namely how models generalize beyond their training data, which is crucial for advancing reasoning capabilities of LLMs.
- The use of influence functions to study generalization in LLMs offers a good perspective on how models might learn to reason.
- The experiments are well-executed, and the analysis and explanation for drawing the findings are reasonable.

**Weaknesses:**

- The study only looks at a subset of the pretraining data, which might not capture less frequent but highly influential documents.
- Findings are based on two models from the same organization, potentially limiting the generalizability across different architectures or training regimes.
- There's no cross-validation with other methods of understanding model behavior which could corroborate the findings.

**Questions:**

- Could you elaborate more on how you define "procedural knowledge" in the context of your findings? How does this relate to the concept of learning algorithms or routines within the training data?
- Given the high influence of code documents, how might this skew the model's reasoning capabilities, especially in non-coding contexts?
- With these insights, what are the potential adjustments or enhancements in training strategies for LLMs to improve their reasoning generalization?

---

> ### Author Response · Authors · 2024-11-18
> **Author response - part 1/2**
>
> We thank the reviewer for such a supportive review. We are very excited to read that the reviewer thinks our *“paper provides an important insight of LLMs”* which is *“crucial for advancing reasoning capabilities of LLMs”*, *the experiments are well-executed, and the analysis and explanation for drawing the findings are reasonable*. In the following, we aim to address the weaknesses mentioned and answer any questions.
>
> **Weakness 1**: *“The study only looks at a subset of the pretraining data, which might not capture less frequent but highly influential documents.”* and *“Findings are based on two models from the same organization, potentially limiting the generalizability across different architectures or training regimes.”*
> We respond to these points in detail in [the general comment to all reviewers above](https://openreview.net/forum?id=1hQKHHUsMx&noteId=ZZ8uSWvlC0). To summarise here, we agree with the reviewer that our results leave open questions about generalisation to other architectures and training regimes, but we believe this does not undermine our conclusions. Further, we believe 5 million documents that are similarly distributed to the pretraining data are sufficient to make the conclusions we have in the paper.
>
> **Weakness 2**: *“There's no cross-validation with other methods of understanding model behavior which could corroborate the findings.”*
> Before choosing EK-FAC influence functions to explain model completions, we thought about using other methods (predominantly less expensive ones, such as representational similarity or first-order gradient-based methods such as TracIn). However, we found in preliminary experiments that these do not estimate the counterfactual we are interested in well (i.e *“how the trained model parameters (or any function thereof, such as the likelihood of completions) change if a datapoint is included in the pretraining set and the model is retrained”*). We summarised these experiments in Appendix A.1, where we show EK-FAC influence functions estimate the counterfactual better than TracIn (based on first-order gradient information). We did not use representational similarity in these experiments because in preliminary experiments this worked even less well than TracIn, and we believe it has little explanatory power for LLM behaviour. Therefore, we expect other methods of explaining model completions to work less well than EK-FAC influence functions, which estimate the counterfactual question about why a model produced a completion best.

---

> > ### Author Response · Authors · 2024-11-18
> > **Author response - part 2/2**
> >
> > **Question 1**: *“Could you elaborate more on how you define "procedural knowledge" in the context of your findings? How does this relate to the concept of learning algorithms or routines within the training data?”*
> > We define procedural knowledge as knowledge that contains information that is applicable to questions which underlie the same tasks (or, procedures). We contrast this to knowledge that is only applicable to specific instantiations of questions or tasks (like we find for the factual questions). We believe learning algorithms or routines described in the pretraining data would fall under this category. Please let us know if this answers the question, and if not we are happy to discuss more.
> >
> > **Question 2**: *“Given the high influence of code documents, how might this skew the model's reasoning capabilities, especially in non-coding contexts?”*.
> > It seems like evidence from multiple sources is converging on code improving LLM reasoning in non-coding context (e.g. [1]), and evidence from our paper adds to this by showing it can also impact reasoning negatively, which hints at the possibility of better code-data filtering for reasoning. An interesting paper recently came out that investigates your question for natural language reasoning [1], and they find that even just training on code models can learn to do natural language reasoning tasks better than random. Some hypotheses around how this is possible are that there’s a lot of natural language data in code data as well, in the form of comments, instructions, Jupyter notebooks with text between code, etc. However, how and why exactly the model’s reasoning capabilities in non-coding context get skewed due to training on code is an interesting open question where there is still a lot to learn.
> >
> > [1] “To Code, or Not To Code? Exploring Impact of Code in Pre-training”, Viraat Aryabumi et al., 2024
> >
> > **Question 3**: *“With these insights, what are the potential adjustments or enhancements in training strategies for LLMs to improve their reasoning generalization?”*
> > This is a good question, and we spent some additional space in the revision to discuss this in more detail. We believe the main takeaway is that pretraining data selection methods can focus on high-quality descriptions and applications of procedures, covering diverse reasoning tasks. Further, the finding that code can be both positively and negatively influential for reasoning highlights there is a possibility here to filter out bad code data. The revisions relevant to this question can be found in the following: the last paragraph of the introduction (L141-150, colour-coded orange) as well as the revision near the end of the discussion (colour-coded orange, L520-522).
> >
> > We believe the revision of the paper constitutes a substantial improvement over the previous one, and we hope the above points can address the weaknesses mentioned in the review. We look forward to discussing further where necessary.

---

> > > ### Comment · Reviewer_RvSn · 2024-11-22
> > >
> > > Thank you for the response. I maintain my positive score.

---

### Official Review · Reviewer_cjVF · 2024-11-05

**Soundness:** 3
**Presentation:** 3
**Contribution:** 3
**Rating:** 6
**Confidence:** 2

**Summary:**

This paper investigates the role of pretraining data in shaping large language models' (LLMs) abilities in reasoning tasks compared to factual question-answering. By analyzing two models of different sizes (7B and 35B parameters) across reasoning and factual queries, the authors aim to understand how LLMs generalize when tackling reasoning tasks and whether they rely on specific retrieval of information or broader procedural knowledge. The study applies influence functions to rank the most impactful pretraining documents for different queries, examining if reasoning draws from procedural patterns rather than specific facts.

Empirically, the study finds that reasoning tasks rely on a more distributed set of documents, often containing procedural content like code snippets or mathematical explanations, while factual questions frequently rely on specific documents containing direct answers. Code-based documents, in particular, emerge as influential for reasoning, likely due to their structured, step-by-step nature. Additionally, reasoning tasks across similar queries show correlated influence scores, suggesting a reliance on shared procedural knowledge. The larger 35B model also shows less variation in influence across documents, hinting at improved data efficiency. Together, these findings imply that LLMs approach reasoning by aggregating procedural knowledge rather than retrieving isolated factual data, shedding light on different generalization strategies in LLMs.

**Strengths:**

1. The paper tries to tackle an intellectually significant question: how do LLMs generalize reasoning abilities from pretraining data to solve completion questions? This exploration into the mechanics of LLM reasoning generalization is both timely and meaningful, given the increasing focus on interpretability and robustness in AI.
2. The findings provide intuitive insights, showing that LLMs draw on a broad range of abstractly related documents when solving reasoning questions, as opposed to the more targeted document reliance seen in factual questions. This highlights the importance of procedural knowledge and coding data for reasoning tasks, an observation that aligns with broader intuitions about reasoning and learning in LLMs.
3. A key technical strength lies in the revision and adaptation of EK-FAC influence functions. The authors refine this method to assess the influence on model accuracy, which is essential for examining how specific documents impact LLM performance in reasoning versus factual tasks.

**Weaknesses:**

1. The overall style resembles a blog post, presenting intriguing observations over a cohesive scientific narrative. For example, the conclusion/discussion section takes more than 1 page to explain everything again. The paper could either prioritize the revised EK-FAC function or convert the observations into some actionable strategies to improve LLMs. Additionally, reorganizing the paper to integrate findings more succinctly could create a more cohesive narrative.
2. Although the paper acknowledges computational constraints, the scale of data and task complexity could be expanded to strengthen the conclusions. The study’s focus on basic arithmetic and simple mathematical queries limits its generalizability to broader reasoning tasks that are common in real-world applications. Also, the study examines only a subset (5 million documents) of the total pretraining data, which may exclude influential documents crucial to understanding the LLMs’ full generalization strategy.
3. The paper predominantly examines positively influential documents, yet negatively influential documents could offer essential insights into reasoning limitations and biases. Understanding negative influences would allow the authors to identify pretraining data that hinders reasoning or introduces procedural noise, shedding light on inherent biases that might restrict generalization. Only focusing on the positively influential documents might bias our judgements towards cherry-picking conclusions.

**Questions:**

1. Could you further explain why calculating document gradients with the base model and the query gradients with the fine-tuned model? Could this discrepancy cause any potential problems?

---

> ### Author Response · Authors · 2024-11-18
> **Author response - part 1/2**
>
> We thank the reviewer for their thoughtful and positive review, stating that we tackle an *“an intellectually significant question”* that is *“both timely and meaningful”*, that *“the findings provide intuitive insight”*, and for recognising the technical difficulty of using EK-FAC influence functions. We significantly rewrote the revision in response to weakness number 1, and highlight in more detail below where these revisions can be found. We also dedicate [a common response](https://openreview.net/forum?id=1hQKHHUsMx&noteId=g1DwLjGUor) to the revision, to highlight the updates to the other reviewers. Further, we added additional analyses for the negative portions of the rankings in response to weakness 3. Please find details below.
>
> **Weakness 1**: *“The overall style resembles a blog post, presenting intriguing observations over a cohesive scientific narrative.”*
> This is very useful feedback, and we believe we have improved the submission in response. Most notably, we changed the title of the submission to *“procedural knowledge in pretraining drives LLM reasoning”* in order to start building a cohesive narrative early on. Relatedly, we changed Fig 1 to summarise key findings instead of the method. At the end of the introduction, we make recommendations for strategies to improve LLMs based on our findings (colour-coded orange, L141-150). We also rewrote the discussion, which now spends only 1 paragraph on summarising results, and the rest on discussion, limitations, and future work.
>
> **Weakness 2**: *“Although the paper acknowledges computational constraints, the scale of data and task complexity could be expanded to strengthen the conclusions.”*
> We would like to refer the reviewer to [the general comment on scope above](https://openreview.net/forum?id=1hQKHHUsMx&noteId=ZZ8uSWvlC0). The summary is that these design decisions were made due to hard compute constraints, and indeed our findings have no bearing on other forms of reasoning; it’s an open question whether similar conclusions will hold there. However, the scope is also large compared to prior work, and in our opinion broad enough to substantiate our claims, crucially relying on documents that are similarly distributed as the pretraining data.
>
> **Weakness 3**: *“The paper predominantly examines positively influential documents, yet negatively influential documents could offer essential insights into reasoning limitations and biases.”*
> Thanks for pointing this out; we agree that the negative influences are equally important, so this is useful feedback. Most of our quantitative analyses already incorporate negative influences (e.g. the correlations are computed using all 5M documents), but we were not clear enough about this in the manuscript, referring often only to *“positively influential”* sequences. We adjusted the manuscript to reflect more clearly that the quantitative findings all hold similarly for the negative portions of the ranking, which supports the claims made (see especially blue-coded text below finding 2 in section 5.1 in the revision, starting on L349, and Figure 24 and 25 in Appendix A.9.3, around L3367).
> For the qualitative analyses, looking at the negative influences is interesting in terms of suggestions for improving LLM reasoning, but it is difficult to make general recommendations based on them. We found few clear qualitative patterns in the negative influences. For example, for factual queries it seems like often topics are similar to the top portions of the rankings, but then do not give all the information (e.g. it discusses Mount Everest but mentions the height of another mountain), which is hard to quantify. Therefore, we believe future work is necessary to make recommendations based on these. We did find an important general pattern which was to the best of our knowledge previously unknown: that code data is equally positively as negatively influential for reasoning. In response to your review, we adjusted the main text to reflect that the code finding is about both the positive and negative portions of the ranking (see blue colour-coded 4th finding L137-139 in the introduction and Finding 5 L462-463), and we adjusted the discussion to more clearly present this insight as a potential future direction towards better LLM reasoning by filtering out bad code data (see discussion orange-coded text L520-522). Further, we are working on releasing the top and bottom 20 data points per query, which can provide further insights for practitioners.
>
> To summarise, our main finding that LLMs learn to produce reasoning traces from procedural knowledge in pretraining data is supported by the negative influences, and we believe it’s an interesting direction for future work to use this to filter negatively influential pretraining data for better reasoning.

---

> > ### Author Response · Authors · 2024-11-18
> > **Author response - part 2/2**
> >
> > **Question 1**: *“Could you further explain why calculating document gradients with the base model and the query gradients with the fine-tuned model? Could this discrepancy cause any potential problems?”*
> > Using different models for calculating the influence scores is a method called SOURCE [1], and we are assuming here that the fine-tuning stage second order information is the identity (meaning instead of using second order information for that stage we multiply the query gradients with the identity matrix, see Figure 6 around L1114 in the appendix of the revision, previously Figure 1). This means we are ignoring the second-order impact on the completions of the fine-tuning stage. We argue that this is unlikely to impact conclusions, because prior work has shown that SFT serves primarily to enhance existing model capabilities as opposed to endowing them with new ones [2], [3], [4]. Further, the fine-tuning stage consisted of a couple thousand supervised instruction-tuning steps, which is negligible compared to the pretraining stage. Nonetheless, we believe an interesting direction for future work would be to apply the same method used here to the fine-tuning stage. We hypothesise that this might surface documents that are similar in formatting to the queries, as opposed to documents that are similar in content. We dedicated a few lines to this question in the revision (L513-518, color-coded orange in the discussion), copied here: *“Another limitation is that we do not look at the supervised fine-tuning stage. The reason we only look at the pretraining data is because the fine-tuning stage is targeted at making the models more aligned and ‘instructable’, as opposed to teaching the model any new capabilities. Prior work has in fact shown that it does not teach the model new capabilities, but rather enhances existing ones (Jain et al., 2024; Kotha et al., 2024; Prakash et al., 2024). Nonetheless, an interesting direction for future work is applying the same method used here to the fine-tuning data.”*
> >
> > [1] Training Data Attribution via Approximate Unrolled Differentiation; Bae et al. 2024.
> > [2] Mechanistically analyzing the effects of fine-tuning on procedurally defined tasks, Jain et al. 2024.
> > [3] Understanding catastrophic forgetting in language models via implicit inference. Kotha et al., 2024.
> > [4] Fine-tuning enhances existing mechanisms: A case study on entity tracking, Prakash et al. 2024.
> >
> > We hope these points address the weaknesses and the question raised by the reviewer. We believe the revision presents a more cohesive narrative as a result of incorporating this feedback, and importantly we believe we were able to make stronger recommendations for future work on improved LLM reasoning because of your points 1 and 3. We are looking forward to an engaged discussion. If there are weaknesses still remaining that might prevent you from increasing your score, we would be grateful for the opportunity to discuss these further.

---

> ### Comment · Area_Chair_B6yu · 2024-11-25
> **[Reminder] Response to Authors**
>
> Dear Reviewer,
>
> As the rebuttal period is drawing to a close, I would appreciate your response to the authors' rebuttal at your earliest convenience.
>
> Best Regards,
>
> Area Chair

---

> > ### Comment · Reviewer_cjVF · 2024-11-28
> > **Thanks for the rebuttal**
> >
> > I thank the authors for the great rebuttal. Glad to see the review and rebuttal improve the paper significantly.
> >
> > I hold a positive score toward acceptance and increased soundness and presentation scores. Thanks.

---

> > > ### Author Response · Authors · 2024-11-28
> > >
> > > Dear reviewer,
> > >
> > > We are very glad to read you believe the paper is improved significantly, and that you now think the contribution, soundness, and presentation are all good. We would be grateful if you could update your rating to reflect this, or otherwise let us know what outstanding concerns are so we can address them carefully.
> > >
> > > Thanks again for your time reviewing and your engagement

---

> > > > ### Author Response · Authors · 2024-12-01
> > > >
> > > > Dear reviewer cjVF,
> > > >
> > > > Given that the discussion period ends soon, we wanted to check in on the above, and see if there are any outstanding concerns we can address.
> > > >
> > > > Thanks again for your time!

---

### Author Response · Authors · 2024-11-18
**Response to all reviewers: limited scope of experiments**

We agree with the reviewers that the scope of tasks and models we look at is narrow. On the other hand, we do 1B LLM-sized gradient dot products (100 queries * 2 models * 5M) for these experiments; in that sense the scope is very large compared to prior interpretability research. We view the task scope as a limitation that was necessary to answer our research question, and not a weakness. We highlight this in the submission: *“All we showed is that in principle it seems to be possible for LLMs to produce reasoning traces using a generalisation strategy that combines information from procedurally related documents, as opposed to doing a form of retrieval.”*. Reviewer **6knH** calls this out as a **strength**: *“The experiments were extremely narrowly defined, but the authors caveat this early and often throughout the paper [ ..] I appreciated that the authors made reasonable decisions and honestly qualified the claims, which were well supported”* . We pushed the compute we had to the limit, and made careful design decisions to answer our research question, which we will explain below.

**Compute and memory**
We used 379,392 TPU v5 chip-hours and 45,056 TPU v4 chip-hours (https://cloud.google.com/tpu/pricing#regional-pricing for reference only), which we parallelised to get it down to about ~3 months of consecutive computations. Further, fitting more than 100 35B query gradients on our largest TPU was impossible, and looping over the pretraining sample twice would almost double the compute required. For comparison, the entire Pythia pretraining suite of models required 544,280 A100 hours (see Appendix D in the Pythia paper).

**Tasks**
We chose mathematical reasoning for two reasons: it has well-defined answers to intermediate steps and we can easily generate questions that underlie the exact same procedure but that use different numbers. We wanted to look at at least 2 tasks per model, but could not fit more than about 100 query gradients on the largest TPU we have (any additional queries would require an entire new loop over the pretraining set, which would take another few months to run). Therefore, we used 40 factual and 40 reasoning questions (and the remaining 20 queries we used for control questions, see [other general common comment](https://openreview.net/forum?id=1hQKHHUsMx&noteId=g1DwLjGUor) for details on these new results). We effectively look at 200 factual, reasoning, and control queries (100 for the 7B, and 100 for the 35B, of which 36 share  prompt, but all different completions).

**Pretraining data**
This aspect is the bottleneck, and took >80% of the TPU chip hours. The important point about this subset of tokens is that it is identically distributed as the pretraining data. Our findings take into account that it is a sample, and we reason about how the conclusions might change if we would be able to look at the entirety of the data in the submission. Unfortunately, that is not tractable (no research has looked at the entire pretraining data in this way), so we have to draw conclusions based on the fact that we have an equally distributed sample. The highly qualitatively relevant data we find for all queries provides strong support that this sample is large enough to cover highly influential data. E.g., we find answers in the documents to 'niche' questions such as *"Who was the prime-minister of the Netherlands in 1970?"*.

**Models**
Our results can be seen as evidence that a common style decoder-only transformer can in principle learn a generalisable strategy from pretraining data, for a 7B and 35B model. Comparing to another model family is an interesting direction for future work, but it is not essential for our conclusion. However, it is prohibitive in terms of compute costs. Furthermore, it’s not immediately clear what other model we could look at, as our investigations require full access to the pretraining distribution. Llama, for example, is trained on proprietary data.

To summarise, we would like to reframe the scope of our experiments as a necessary limitation given the high cost of the experiments, and not a weakness. We are the first to look at the pretraining data in this way to understand how LLMs generalise when reasoning, and show that it is possible for LLMs to learn a generalisable strategy from procedural knowledge in pretraining. We agree with the reviewers that our results leave open the question of whether this holds for other models and forms of reasoning, like inductive reasoning. We added the a few lines in the revision to highlight this further (L525-528). We are excited about future work in this area. When influence functions become further tractable, findings can be confirmed on the entire pretraining set (which is an active area, e.g. [1], but these style of functions are currently less effective in estimating the counterfactual).

[1] *"What is Your Data Worth to GPT? LLM-Scale Data Valuation with Influence Functions"*, Sang Keun Choe et al., 2024

---

### Author Response · Authors · 2024-11-18
**To all reviewers: details on revision**

Dear reviewers,

We believe we have significantly improved our submission in response to your reviews, detailed in a separate comment to each reviewer below, and we want to thank you all for your thoughtful reviews. In this brief comment, we wanted to highlight two changes to the manuscript in response.

The first change is in response to reviewer **cjVF**’s first weakness saying we should present a more cohesive narrative. To this end, we change the title to *“Procedural Knowledge in Pretraining Drives Reasoning in Large Language Models”*. With this title we aim to introduce the main finding early. To address the same weakness, we change Figure 1, which now represents a summary of our findings instead of an image of the pipeline, which we moved to the appendix (Figure 6). We also rewrote the discussion to spend less time on summarising results and more on discussion.

The second major change is that we added experimental results based on a group of 20 control queries for each model (which we were able to fit together with the 80 queries for each model in the same loop over the pretraining data). Because it was not feasible to add an interesting amount of additional reasoning tasks (see [the other comment to all reviewers right below](https://openreview.net/forum?id=1hQKHHUsMx&noteId=ZZ8uSWvlC0), about scope), we believed a better use of these 20 extra queries was to test alternative hypotheses about the data. These queries are control queries in that they are similar to the factual and reasoning queries in style and wording, but do not require any factual retrieval or reasoning to be resolved. We believe these additional results help address raised points by reviewers about the experimental scope by confirming that similar quantitative results do not hold for a control group. For the change to the main paper, please refer to the revision at the end of Finding 1 in the quantitative findings section 5.1 (L314-319, orange colour-coded). Table 10-14 in the Appendix has examples of what the control queries look like, and they can also be found in the supplement.

More generally, we have colour-coded all revisions with colours specific to reviewers:

Orange: relevant to multiple reviewers.
Blue: relevant to **reviewer cjVF**.
Green: relevant to **reviewer RvSn**.
Red: relevant to **reviewer 6knH**.
Purple: relevant to **reviewer KXBG**.

We hope our revisions detailed below in response to your reviews address all your points and we are happy to discuss further wherever required. Thank you again for your time!

---

### Author Response · Authors · 2024-11-22
**Follow-up: request for engagement during discussion period**

Dear Reviewers,

With only a few days remaining in the discussion period, we would greatly appreciate your engagement to ensure a constructive dialogue. In our revision, we’ve worked hard to address your feedback, making significant improvements to the paper:

- The findings present a more coherent message, guided by reviewer cjVF's comments.
- We included additional experimental results and responses to shared reviewer points about the scope of the work.
- Detailed responses to reviewer-specific points in the each separate comment below.

We are eager to hear your thoughts on these updates and hope you’ll have a chance to review our responses. We value your time and effort in shaping this submission.

Thank you again for your thoughtful reviews and for considering our responses.

Best regards,

The Authors of Submission 7193

---

### Meta-Review · Area_Chair_B6yu · 2024-12-19

**Metareview:**

The paper investigates how LLMs utilize pre-training data differently when performing reasoning tasks versus factual question-answering. Using influence functions, the authors analyzed two models (7B and 35B parameters) to examine which pre-training documents had the greatest impact on model outputs. The experiments reveal several key insights: reasoning tasks draw from a broader, more distributed set of pre-training documents compared to factual queries; larger models show more uniform influence scores across documents, suggesting improved data efficiency; and documents containing procedural content (especially code snippets and mathematical explanations) are particularly influential for reasoning tasks. These findings advance our understanding of how LLMs may develop reasoning abilities through training, suggesting they acquire procedural knowledge rather than simply memorizing solutions.

The paper makes significant contributions to reasoning research for several reasons: (1) it addresses a fundamental question about how LLMs develop reasoning capabilities from pre-training data, representing frontier research in model interpretability by tracing responses back to training samples; (2) the technical execution features well-adapted EK-FAC influence functions and thoughtful comparative analysis between different model sizes and task types; and (3) the findings provide actionable insights about the importance of procedural knowledge and diverse training data in developing reasoning capabilities.

This paper also has several limitations: (1) the experimental scope is notably narrow, examining few mathematical questions and using just two models from the same family. While computational constraints explain some limitations, validation on public models like OLMo or StarCoder whose pre-training corpus is also tractable would strengthen the findings; (2) the study analyzes only a subset of pre-training data, potentially missing less frequent but influential documents; (3) additionally, while the paper discusses "reasoning" broadly, it specifically examines mathematical reasoning, and distinctions between different types of reasoning (logical, spatial) could be better explained.

Overall, the paper's approach and insights into how LLMs leverage training data for reasoning tasks make it a valuable contribution to the field. I believe it should be accepted.

**Additional Comments On Reviewer Discussion:**

In summary, all reviewers have acknowledged the authors' rebuttal, with three maintaining their original positive assessments and expressing appreciation for the authors' clarifications. While one reviewer's concerns remained unchanged after the rebuttal, their primary criticism regarding the lack of other models should be considered in historical context - when this research began, full open models are relatively rare (e.g, OLMo). Additionally, the computational intensity of running influence functions makes it impractical to expect more data points. Overall, the discussion supports moving forward with acceptance, as the paper's core contributions and insights outweigh these limitations.

---

### Decision · Program_Chairs · 2025-01-22

Accept (Poster)